# The human amygdala parametrically encodes the intensity of specific facial emotions and their categorical ambiguity

Shuo Wang[1,2], Rongjun Yu[3,4], J. Michael Tyszka[5], Shanshan Zhen[4], Christopher Kovach[6], Sai Sun[4], Yi Huang[4], Rene Hurlemann[7], Ian B. Ross[8], Jeffrey M. Chung[9], Adam N. Mamelak[9], Ralph Adolphs[1,2,5] & Ueli Rutishauser[5,9]

The human amygdala is a key structure for processing emotional facial expressions, but it remains unclear what aspects of emotion are processed. We investigated this question with three different approaches: behavioural analysis of 3 amygdala lesion patients, neuroimaging of 19 healthy adults, and single-neuron recordings in 9 neurosurgical patients. The lesion patients showed a shift in behavioural sensitivity to fear, and amygdala BOLD responses were modulated by both fear and emotion ambiguity (the uncertainty that a facial expression is categorized as fearful or happy). We found two populations of neurons, one whose response correlated with increasing degree of fear, or happiness, and a second whose response primarily decreased as a linear function of emotion ambiguity. Together, our results indicate that the human amygdala processes both the degree of emotion in facial expressions and the categorical ambiguity of the emotion shown and that these two aspects of amygdala processing can be most clearly distinguished at the level of single neurons.

[1] Computation and Neural Systems, California Institute of Technology, Pasadena, California 91125, USA. [2] Humanities and Social Sciences, California Institute of Technology, Pasadena, California 91125, USA. [3] Department of Psychology, National University of Singapore, Singapore 117570, Singapore. [4] School of Psychology, Center for Studies of Psychological Application, and Key Laboratory of Mental Health and Cognitive Science of Guangdong Province, South China Normal University, Guangzhou 510631, China. [5] Division of Biology and Biological Engineering, California Institute of Technology, Pasadena, California 91125, USA. [6] Department of Neurosurgery, University of Iowa, Iowa City, Iowa 52242, USA. [7] Division of Medical Psychology, University of Bonn, Bonn 53105, Germany. [8] Epilepsy and Brain Mapping Program, Huntington Memorial Hospital, Pasadena, California 91105, USA. [9] Departments of Neurosurgery and Neurology, Cedars-Sinai Medical Center, Los Angeles, California 90048, USA. Correspondence and requests for materials should be addressed to S.W. (email: wangshuo45@gmail.com) or to R.Y. (email: psyyr@nus.edu.sg) or to U.R. (email: ueli.rutishauser@cshs.org).

The human amygdala has long been associated with recognizing faces and facial emotions[1–3]. Subjects who lack a functional amygdala can have a selective impairment in recognizing fearful faces[4–6], and blood-oxygen-level dependent functional magnetic resonance imaging (BOLD-fMRI) shows activation within the amygdala that is often highest for fearful faces[7–9]. While the large majority of work has focused on fearful faces[1], the human amygdala is also responsive to neutral or happy faces measured using BOLD-fMRI[10] and single-neuron recordings[11–14]. Indeed, in some studies the amygdala responds to some extent to all facial expressions[15]. Similarly, amygdala neurons in non-human primates[16,17] respond to faces, face identities, facial expressions, gaze directions and eye contact[18–20]. Nonetheless, recent neuroimaging studies still argue for a disproportionate amygdala response to facial expressions related to threat (fear and anger[21]) and lesion studies provide strong support for a role in recognizing fear[2,4–6]. It is worth noting that the above-mentioned lesion studies encompass damage to both basolateral and centromedial nuclei. In contrast, patients with lesions involving only the basolateral, but sparing the centromedial nuclei, have revealed diverging results from those with such complete lesions, typically showing a hypersensitivity to fear[22,23]. In the three amygdala lesion patients we study here, most of the basolateral complex of the amygdala was lesioned but the centromedial nucleus was spared, and we thus hypothesized that these patients would show an increased sensitivity to fear in faces.

On the other hand, in addition to encoding facial emotions, an alternative hypothesized function of the amygdala is to identify ambiguous stimuli and modulate vigilance and attention as a function thereof[24–26]. Here we tested the hypothesis that the amygdala encodes aspects of perceptual ambiguity when making judgments about facial emotions. Throughout this study, by ambiguity we refer to the closeness to the perceptual boundary during categorical decision between two emotional facial expressions. Note that in studies of decision-making, the term 'ambiguity' usually refers to an absence of information about a stimulus above and beyond categorical uncertainty. In contrast, the term ambiguity here refers exclusively to categorical uncertainty, because all information about the stimulus was always available and the task was deterministic (see Discussion section for details). Previous neuroimaging work indicates that the amygdala can differentiate stimuli that vary in their perceptual ambiguity. For instance, the amygdala responds strongest to highly trustworthy and untrustworthy faces but less to faces of intermediate (ambiguous) trustworthiness[27–29], even if the faces are perceived unconsciously[27]. Furthermore, for both faces varying along a valence dimension and faces varying along an orthogonal non-valence dimension, the amygdala responds strongest to the anchor faces[30]. Consistent with this idea, it has been found that emotional stimuli of any type lead to greater amygdala activity compared to neutral stimuli, with comparable effect sizes for most negative and positive emotions[31]. Together, these findings argue that the amygdala plays a key role in processing categorical ambiguity of dimensions represented in faces.

To test these two theories of human amygdala function, we performed three separate studies to investigate this question at three levels of abstraction: (i) human epilepsy patients with single-neuron recordings in the amygdala, (ii) healthy subjects with fMRI, and (iii) subjects with well-defined lesions of the amygdala for behavioural analysis. All subjects performed the same task, in which we asked them to judge the emotion of faces that varied systematically as a function of both ambiguity and the degree of fear or happiness. The behavioural and neuroimaging results revealed a role of the amygdala in processing of both emotion degree and ambiguity. At the single-neuron level, in contrast, we found clear evidence for a segregation of these two functions: we identified two separate populations of neurons, one whose response correlated with the gradual change of fearfulness or happiness of a face and a second whose response primarily correlated with decreasing level of categorical ambiguity of the emotion. This separation of function was only visible at the level of individual neurons but not at the level of BOLD-fMRI. This highlights the utility of directly recording single neurons, which enabled us to, for the first time, reveal two separate human amygdala neuron populations who signal the degree of emotion and levels of emotion ambiguity during decision making about faces. Together, our work indicates that both signals are likely used for computations performed by the amygdala.

## Results

**Emotion judgments.** We asked subjects to judge emotional faces as fearful or happy. Faces were either unambiguously happy or unambiguously fearful ('anchors') or graded ambiguous morphs between these two emotions (Fig. 1a,b and Supplementary Fig. 1). Subjects were 3 patients with focal bilateral amygdala lesions (Supplementary Fig. 2), 9 neurosurgical patients (14 sessions; Supplementary Table 1) and 19 healthy subjects for the fMRI study, as well as another 15 healthy control subjects. In the three amygdala lesion patients, most of the basolateral complex (lateral, basal and accessory basal nuclei) was lesioned bilaterally but the central, medial and cortical nuclei of the amygdala were intact (see Supplementary Fig. 2 and Methods for details). This pattern of amygdala damage has been previously reported to result in possibly exaggerated ('hypervigilant') responses to fear stimuli[22,23,32] (see Discussion for details).

For each session, we quantified behaviour as the proportion of trials identified as fearful as a function of morph level (Fig. 1c and Supplementary Fig. 1b–e). We found a monotonically increasing relationship between the likelihood of identifying a face as fearful and the proportion of fear shown in the morphed face (Fig. 1c). We quantified each psychometric curve using two metrics derived from the logistic function: (i) $x_{half}$—the midpoint of the curve (in units of %fearful) at which subjects were equally likely to judge a face as fearful or happy, and (ii) $\alpha$—the steepness of the psychometric curve. Based on these two metrics, the behaviour of the neurosurgical patients was indistinguishable from the control subjects (Fig. 1d,e; $x_{half}$: unpaired two-tailed $t$-test: $t(27) = 1.10$, not significant (NS); $\alpha$: $t(27) = 1.98$, NS). In contrast, the amygdala lesion patients ($x_{half} = 44.2 \pm 1.88\%$) were more likely to judge faces as fearful, with $x_{half}$ significantly lower than neurosurgical patients (Fig. 1d; $x_{half} = 53.2 \pm 4.97\%$; $t(15) = 3.00$, $P = 0.0089$, effect size in Hedges' $g$: $g = 1.81$, permutation $P < 0.001$) and controls ($x_{half} = 51.1 \pm 5.16\%$; $t(16) = 2.23$, $P = 0.040$, $g = 1.34$, permutation $P = 0.058$) and $\alpha$ significantly steeper (Fig. 1e; $t(15) = 3.85$, $P = 0.0016$, $g = 2.33$, permutation $P = 0.002$). We also confirmed these behavioural results with a logistic mixed model (Supplementary Notes). Together, our results suggest that amygdala lesion patients had an abnormally low threshold for reporting fear, a finding consistent with prior reports[22,33] (see Discussion section).

**Confidence judgments.** We defined emotion ambiguity as the variability in judging the emotion of a given morphed face. The more variable the judgment, the more ambiguous is the face (Fig. 1b,c). After reporting a face as fearful or happy, we asked subjects to report their confidence in their decisions (Fig. 1a). Subjects reported significantly higher levels of confidence for anchor faces (no ambiguity) compared to the ambiguous faces (Fig. 1f,j,n; one-way repeated-measure analysis of variance

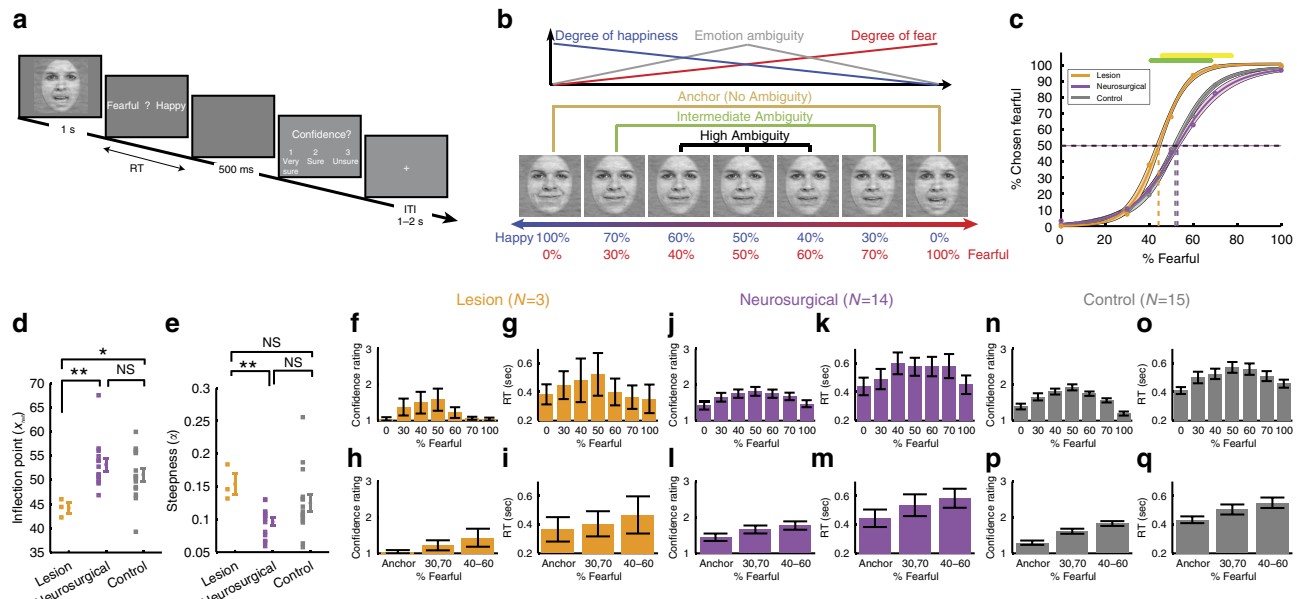

**Figure 1 | Behavioural results.** (**a**) Task. A face was presented for 1 s followed by a question asking subjects to identify the facial emotion (fearful or happy). After a blank screen of 500 ms, subjects were then asked to indicate their confidence in their decision ('1' for 'very sure', '2' for 'sure' or '3' for 'unsure'). Faces are not shown to scale. (**b**) Sample stimuli of one female identity ranging from 0% fear/100% happy to 100% fear/0% happy. (**c–q**) Behavioural results. (**c**) Group average of psychometric curves. The psychometric curves show the proportion of trials judged as fearful as a function of morph levels (ranging from 0% fearful (100% happy; on the left) to 100% fearful (0% happy; on the right)). Shaded area denotes ± s.e.m. across subjects/sessions (n = 3, 14, 15). The top bars illustrate the points with significant difference between amygdala lesion patients and neurosurgical patients (green; unpaired two-tailed t-test, P < 0.05, corrected by FDR for Q < 0.05) and between amygdala lesion patients and healthy controls (yellow). (**d**) Inflection point of the logistic function ($x_{half}$). (**e**) Steepness of the psychometric curve ($\alpha$). Individual values are shown on the left and average values are shown on the right. Error bars denote one s.e.m. across subjects/sessions. Asterisks indicate significant difference using unpaired two-tailed t-test. *P < 0.05, and **P < 0.01. NS: not significant (P > 0.05). (**f–q**) Confidence ratings for lesion (**f–i**), neurosurgical (**j–m**) and control (**n–q**) subjects. (**f,j,n**) Explicit confidence ratings showed highest confidence for anchor faces and lowest for the most ambiguous (50% fear/50% happy) faces. (**g,k,o**) The RT for the fear/happy decision can be considered as an implicit measure of confidence because it showed a similar pattern as the explicit ratings. For the neural analysis, we grouped the seven morph levels into three levels of ambiguity (anchor, 30%/70% morph, 40–60% morph). Both explicit (**h,l,p**) and implicit (**i,m,q**) confidence measures varied systematically as a function of ambiguity. The behavioural patterns of all three subject groups were comparable. Error bars denote one s.e.m. across subjects/sessions.

(ANOVA) of morph levels; lesion: F(6,12) = 3.22, P = 0.040, $\eta^2$ = 0.36; neurosurgical: F(6,72) = 16.6, P = 6.15 × 10$^{-12}$, $\eta^2$ = 0.11; control: F(6,84) = 27.2, P = 8.26 × 10$^{-18}$, $\eta^2$ = 0.40). Also, reaction times (RT) for the fearful/happy judgment were faster for anchor faces compared to ambiguous faces (Fig. 1g,k,o; lesion: F(6,12) = 2.20, NS, $\eta^2$ = 0.13; neurosurgical: F(6,78) = 9.09, P = 1.56 × 10$^{-7}$, $\eta^2$ = 0.059; control: F(6,84) = 12.3, P = 7.12 × 10$^{-10}$, $\eta^2$ = 0.14; Supplementary Fig. 3a; fMRI: F(6,108) = 8.45, P = 1.59 × 10$^{-7}$, $\eta^2$ = 0.044). For further analyses, we grouped all trials into three levels of ambiguity (Fig. 1b; anchor, intermediate (30%/70% morph), and high (40–60% morph)), which showed the expected systematic relationships with confidence (Fig. 1h,l,p) and RT (Fig. 1i,m,q and Supplementary Notes). Notably, this relationship was similar in all subject groups including the amygdala lesion patients, arguing that amygdala lesion patients were not impaired in judging ambiguity and confidence, even though they were atypical in how they judged the degree of fear.

**Functional neuroimaging.** We next conducted a fMRI study with 19 healthy subjects using the same morphed face stimuli and task (Fig. 1a). Each subject first performed a separate face localizer task to identify a functional region of interest (ROI) within the amygdala sensitive to faces (Supplementary Fig. 4a).

We first compared the response of voxels within the functional ROI as a function of emotion degree. We found that activation correlated with the degree of emotion (interestingly, increasing activation correlated positively with the degree of happiness, negatively with the degree of fear) specifically within the left amygdala (Fig. 2a; peak: Montreal Neurological Institute (MNI) coordinate: x = − 21, y = − 6, z = − 15, Z = 3.22, 6 voxels, family-wise error (FWE) P < 0.05, small volume corrected (SVC)). The average BOLD signal within the entire ROI in the left amygdala was significantly negatively correlated with increasing fear levels (Fig. 2b; Pearson correlation: r = − 0.79, P = 0.034; see Supplementary Fig. 4b and Supplementary Table 2 for whole-brain results).

Next, we investigated whether the amygdala BOLD signal correlated significantly with ambiguity. This time we found a significant increase of activity in the right, but not the left, amygdala with decreasing level of ambiguity (Fig. 2c; peak: x = 30, y = 0, z = − 21, Z = 3.17, 17 voxels, FWE P < 0.05, SVC) (see Supplementary Fig. 4c and Supplementary Table 2 for other areas). The time course of the BOLD signal in the right amygdala as a function of different ambiguity levels revealed that anchor faces elicited the strongest BOLD response while the most ambiguous faces elicited the weakest response (Fig. 2d,e; one-way repeated ANOVA of parameter estimate (beta values): F(2,36) = 7.10, P = 0.0025, $\eta^2$ = 0.062; average % BOLD change of TR 3 and 4: F(2,36) = 2.55, NS, $\eta^2$ = 0.051). The difference between anchor versus intermediate ambiguous faces was no greater than that between intermediate versus the most ambiguous faces (paired t-test on beta values: t(18) = 1.44, NS,

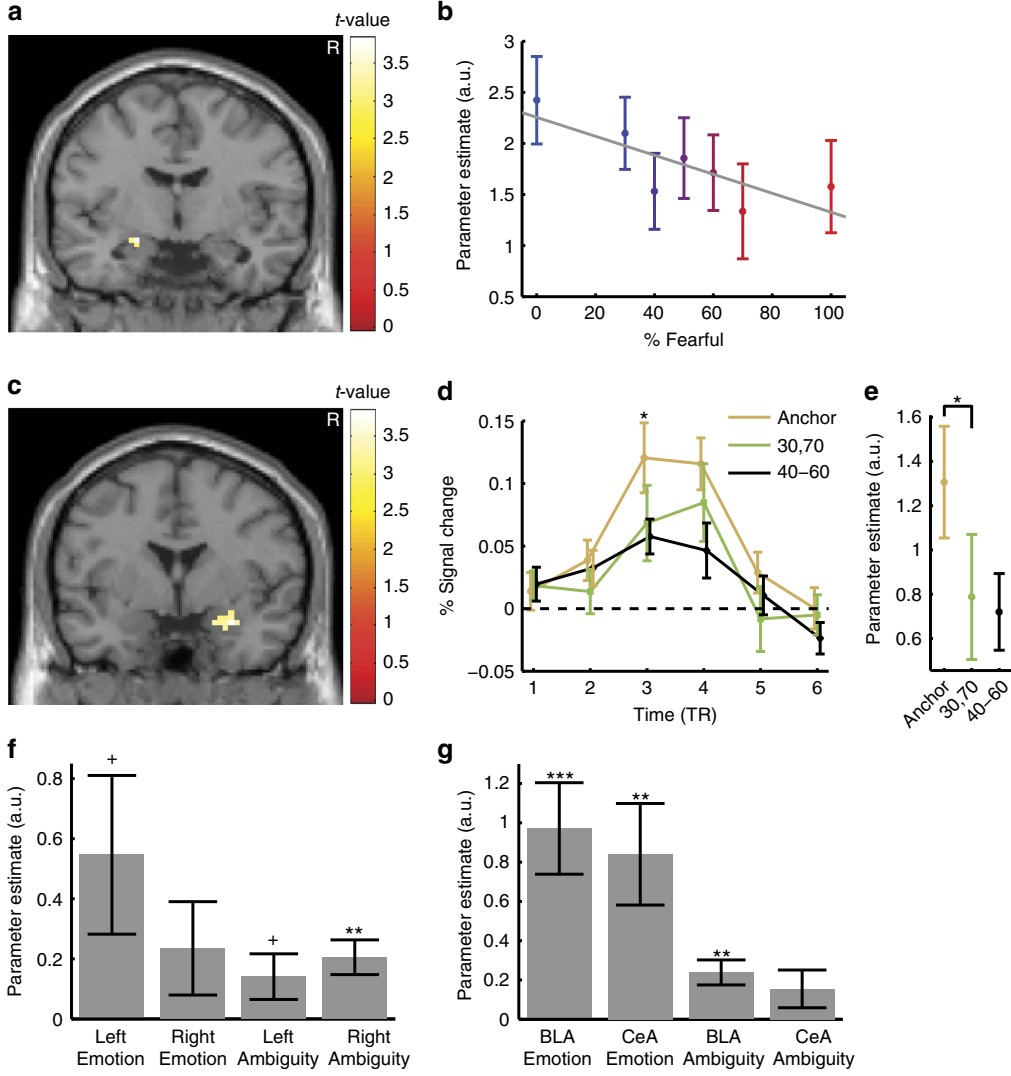

**Figure 2 | fMRI results.** (**a**) Fear levels were negatively correlated with the BOLD activity in the left amygdala. Here we used a functional amygdala ROI defined by the localizer task (see Methods section). The generated statistical parametric map was superimposed on anatomical sections of the standardized MNI T1-weighted brain template. Images are in neurological format with subject left on image left. R: right. (**b**) Parameter estimate (beta values) of the GLM for each fear level (Pearson correlation: $r = -0.79$, $P = 0.034$). The colour scale denotes increasing degree of fear (cf. Fig. 1b). Error bars denote one s.e.m. across 19 subjects. (**c**) Ambiguity levels were correlated with the BOLD activity in the right amygdala (functional ROI defined by localizer task). (**d**) Time course of the BOLD response in the right amygdala (averaged across all voxels in the cluster) in units of TR (TR = 2 s) relative to face onset. Error bars denote one s.e.m. across subjects. One-way repeated ANOVA at each TR: *$P < 0.05$. (**e**) Parameter estimate of the GLM for each ambiguity level (one-way repeated-measure ANOVA, $P = 0.0025$). Error bars denote one s.e.m. across subjects. Asterisks indicate significant difference between conditions using paired two-tailed $t$-test. *$P < 0.05$. (**f**) Mean GLM parameter estimate of all voxels within the functional ROI for each side of the amygdala and for each aspect of the emotion coding. Error bars denote one s.e.m. across subjects. $T$-test against 0: **$P < 0.01$, and +: $P < 0.1$. (**g**) Peak voxel activity within the basolateral nuclei (BLA) and central nuclei (CeA) for each aspect of the emotion coding.

$g = 0.58$), suggesting that different levels of ambiguity were encoded with similar strength. Note that when modelling with four ambiguity levels (anchor, 30%/70% morph, 40%/60% morph and 50% morph), and/or when adding RT as another orthogonalized parametric modulator, we derived essentially the same results (Supplementary Fig. 4f–n). Thus the observed correlation with ambiguity appeared to track the degree of ambiguity parametrically and could not be attributed simply to different RTs. Note that the above results remain qualitatively the same when using an anatomical ROI of the entire amygdala (Supplementary Fig. 4d,e).

Confirming the lateralized results described above, the average BOLD signal within the entire functional ROI of the amygdala showed a marginally significant interaction between the laterality

of activation and the aspect of emotion coding (Fig. 2f; repeated-measure ANOVA of laterality (left versus right) × emotion aspect (degree versus ambiguity): F(1,18) = 4.04, $P = 0.060$, $\eta^2 = 0.0018$). Finally, the fusiform face area also tracked the emotion degree and ambiguity (Supplementary Notes).

In conclusion, we found that both emotion degree and ambiguity modulated the BOLD signal of the amygdala, with the left amygdala primarily tracking the degree of happiness and the right amygdala primarily tracking ambiguity.

**Amygdala neurons encode the degree of fear or happiness.** Finally, we investigated the amygdala's role in emotion and ambiguity at the single-neuron level. We recorded from 234

single neurons ($>0.2$ Hz mean firing rate) in the amygdalae of 9 neurosurgical patients implanted with depth electrodes (Supplementary Table 1; see Supplementary Fig. 5 for spike sorting quality metrics) while they performed the same task (Fig. 1a). In total, we recorded 14 sessions (as is customary in analyses of human single-unit recordings, neurons from each individual recording session were considered independent even if they were from the same patient because it was not possible to reliably record the same neurons across sessions).

We quantified the response of each neuron based on the number of spikes observed after stimulus onset (1.5 s window, starting 250 ms after stimulus onset, as is customary given the long latencies of human amygdala neurons[34,35]). Ninety-six neurons (41.0%) were responsive to the onset of faces (response versus firing rate at baseline 1 s before stimulus onset; paired two-tailed $t$-test, $P<0.05$; 59 increased and 37 decreased activity compared to baseline; binomial test on the number of significant cells: $P<10^{-20}$). This substantial proportion of face-responsive neurons in the amygdala is similar to previous studies[13]. We next investigated whether neurons were modulated by emotion degree. We found that the response of 33 neurons (14.1%; binomial $P<10^{-7}$; 15 were also face responsive; see Supplementary Fig. 7e,f for chance levels and Supplementary Notes for control analysis) was correlated with morph levels (averaged separately for each morph level, linear regression with %fearful at $P<0.05$). In the following, we refer to this group of neurons as 'emotion-tracking'. There were two subtypes of such responses: 21/33 neurons increased their firing rate as a function of the degree of fear (Fig. 3a and Supplementary Fig. 7a,c), whereas 12/33 increased their firing rate as a function of the degree of happiness (Fig. 3b and Supplementary Fig. 7b,d).

The linear model we used revealed that a substantial proportion of variance was explained by a continuous response as a function of the degree of fear or happiness (see Fig. 3c for $R^2$),

with an average absolute slope of $0.80 \pm 0.75$ Hz/100%fear (Fig. 3d; $0.71 \pm 0.72$ for positive slope and $-0.95 \pm 0.80$ for negative slope). We also compared our linear model to more complex models but found that a linear relationship fitted our data better. In particular, our data were not best described by a step-like threshold model (Supplementary Notes).

Overall, these findings argue that some human amygdala neurons parametrically encoded gradual changes of facial emotions. This is a significantly more fine-grained representation relative to the binary discrimination between fearful and happy facial expressions we and others have previously reported in studies that did not explicitly test for a more continuous representation[7,35].

**Amygdala neurons encode emotion ambiguity.** We next investigated whether the responses of amygdala neurons might also be modulated by the level of categorical ambiguity of the emotion, regardless of the specific emotion. Comparing the firing rate between anchor faces (regardless of emotion) and morphed faces revealed a subset of 36 neurons (15.4%; binomial $P<10^{-9}$; unpaired two-tailed $t$-test at $P<0.05$), most of which (30/36) had a higher firing rate for the anchor compared to the morphed faces. The pattern of response of these 'ambiguity-coding' neurons suggests that they differentiated ambiguity levels but not individual facial emotions.

To directly investigate this hypothesis, we next used a linear regression to identify neurons whose firing rate correlated trial-by-trial with three levels of ambiguity (anchor, intermediate (30%/70% morph) and high (40–60% morph)). Thirty-two neurons showed a significant trial-by-trial correlation (13.7%; binomial $P<10^{-7}$; Fig. 4a,b and Supplementary Fig. 6; see Supplementary Fig. 7h,i for chance levels and Supplementary Notes for control analysis), most (29/32) of which had the maximal firing rate for the anchors (which have low ambiguity).

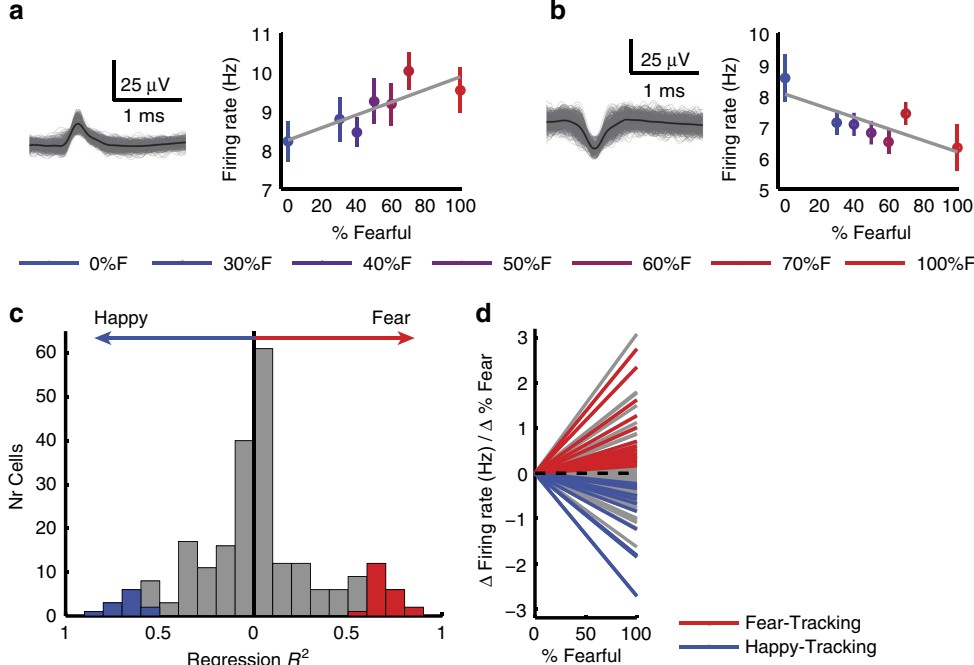

**Figure 3 | Emotion-tracking neurons. (a)** Example neuron that increased its firing rate as a function of %fear (linear regression, $P=0.024$). **(b)** Example neuron that increased its firing rate as a function of %happy ($P=0.032$). Right shows the average firing rate for each morph level 250- to 1,750-ms post-stimulus onset. Grey lines represent linear fit. Error bars denote $\pm$ s.e.m. across trials. Waveforms for each unit are shown at the left. **(c)** Histogram of regression $R^2$. Neurons that had a higher firing rate for fearful faces are shown on the right of 0, whereas neurons that had a higher firing rate for happy faces are shown on the left. Fear-tracking neurons are in red, happy-tracking neurons are in blue, whereas non-emotion-tracking neurons are in grey. **(d)** Slope of linear regression fit. Grey: non-emotion-tracking neurons. Red: fear-tracking neurons. Blue: happy-tracking neurons.

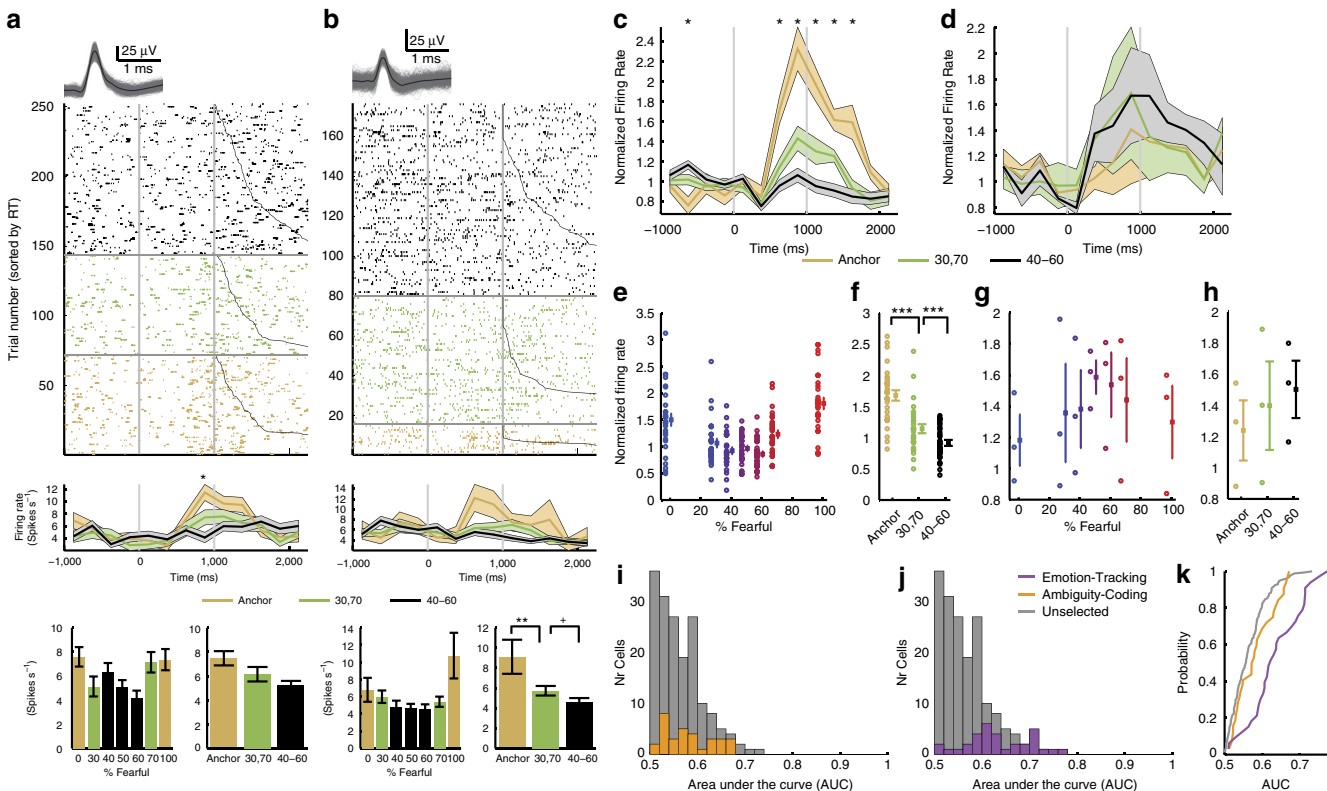

**Figure 4 | Ambiguity-coding neurons.** (**a,b**) Two example neurons that fire most to the anchors and least to the most ambiguous stimuli (linear regression: $P < 0.05$). Each raster (top), PSTH (middle) and average firing rate (bottom) is colour coded according to ambiguity levels as indicated. Trials are aligned to face stimulus onset (left grey bar, fixed 1 s duration) and sorted by RT (black line). PSTH bin size is 250 ms. Shaded area and error bars denote ± s.e.m. across trials. Asterisk indicates a significant difference between the conditions in that bin ($P < 0.05$, one-way ANOVA, Bonferroni-corrected). Bottom left shows the average firing rate for each morph level 250- to 1,750-ms post-stimulus-onset. Bottom right shows the average firing rate for each ambiguity level 250- to 1,750-ms post-stimulus onset. Asterisks indicate significant difference between levels of ambiguity using unpaired two-tailed t-test. $**P < 0.01$ and $+$: $P < 0.1$. Waveforms for each unit are shown at the top of the raster plot. (**c,d**) Average normalized firing rate of ambiguity-coding neurons that increased ($n = 29$) and decreased ($n = 3$) their firing rate for the least ambiguous faces, respectively. Asterisk indicates a significant difference between the conditions in that bin ($P < 0.05$, one-way ANOVA, Bonferroni-corrected). (**e,f**) Mean normalized firing rate at each morph level (**e**) and ambiguity level (**f**) for 29 units that increased their spike rate for less ambiguous faces. (**g,h**) Mean normalized firing rate at each morph level (**g**) and ambiguity level (**h**) for 3 units that increased their spike rate for more ambiguous faces. Normalized firing rate for each unit (left) and mean ± s.e.m. across units (right) are shown at each level. Asterisks indicate significant difference between conditions using paired two-tailed t-test. $***P < 0.001$. (**i**) Histogram of AUC values for ambiguity-coding neurons (orange) and unselected neurons that are neither ambiguity coding nor emotion tracking (grey). (**j**) Histogram of AUC values for emotion-tracking neurons (purple) and unselected neurons that are neither ambiguity coding nor emotion tracking (grey). (**k**) Cumulative distribution of the AUC values. (**i–k**) Ambiguity-coding neurons did not differentiate fearful versus happy emotions with anchor faces (similar AUC values as unselected neurons) but emotion-coding neurons did (greater AUC values than unselected neurons).

We refer to this group of cells as ambiguity-coding neurons. Neurons with higher firing rate for less ambiguous faces had a U-shaped response curve as a function of morph levels (Fig. 4c,e) and thus decreasing levels of activity as a function of ambiguity levels (Fig. 4f). The difference between anchor versus intermediate ambiguous faces (mean normalized firing rate: 0.53 ± 0.44; mean ± s.d.) was greater than that between intermediate versus the most ambiguous faces (0.23 ± 0.24; paired t-test: $t(28) = 2.96$, $P = 0.0062$, $g = 0.86$; Fig. 4c,f), indicating a sharper transition from anchor face to ambiguity. In contrast, neurons with higher firing rate for more ambiguous faces had inverted U-shaped response curves as a function of morph levels (Fig. 4d,g) and thus increasing the levels of activity as a function of ambiguity levels (Fig. 4h). Since most ambiguity-coding neurons responded least to high ambiguity but most to unambiguous anchor faces ($\chi^2$-test: $P < 10^{-10}$; Fig. 4e–h), the overall population response ($n = 32$) was, as expected, maximal for the least ambiguous faces.

Did ambiguity-coding neurons also carry information about the specific emotion of a face? We performed a single-neuron receiver-operating characteristic curve (ROC) analysis, considering only correctly identified anchor faces, to answer this question (Methods section). The area under the curve (AUC) of the ROC specifies the probability by which an ideal observer could predict the choice (fear or happy) of a subject by counting spikes in an individual trial. Ambiguity-coding neurons had an average AUC of 0.58 ± 0.052 (mean ± s.d.; Fig. 4i,k), significantly lower than emotion-tracking neurons (0.64 ± 0.069; Kolmogorov–Smirnov test: $P = 0.0052$) but similar to neurons that were neither selected as ambiguity coding nor emotion tracking (0.56 ± 0.047; NS versus AUC of 0.58 ± 0.052). This shows that ambiguity-coding neurons did not encode emotion degree. Note that chance performance here was >0.5 due to the symmetry of the response (see Methods section). Thus we used the unselected neurons to empirically estimate chance performance (which was 0.56). As expected, emotion-tracking neurons had significantly higher AUC values than unselected neurons ($P = 4.91 \times 10^{-8}$; Fig. 4j,k). Furthermore, only two ambiguity-coding neurons were also emotion-tracking neurons. A $\chi^2$-test of independence showed that emotion-tracking and ambiguity-coding neurons were two

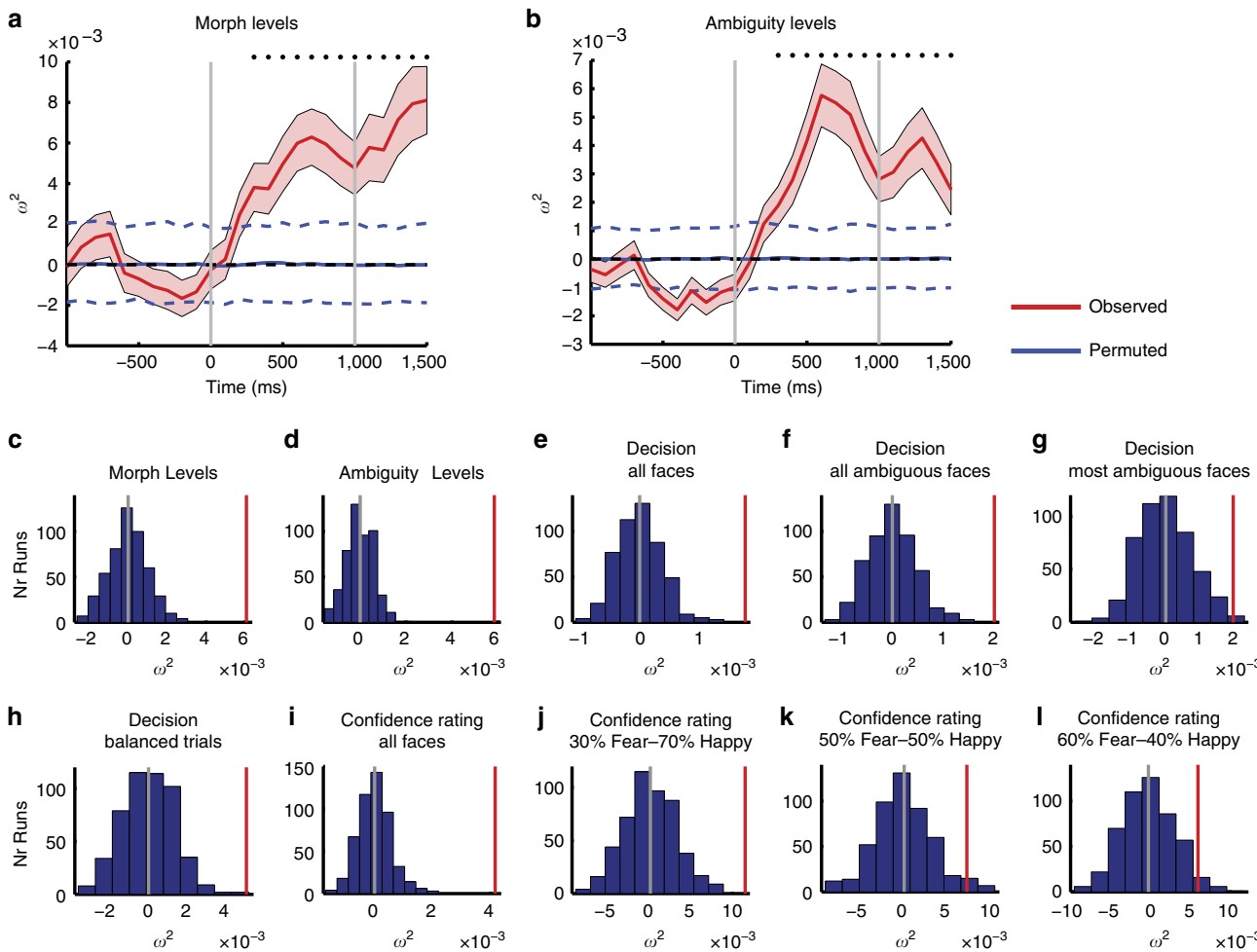

**Figure 5 | Population analysis of all recorded neurons ($n = 234$) using a regression model and effect size metric $\omega^2$. (a,b)** Time course of the effect size averaged across all neurons. (**a**) Morph levels. (**b**) Ambiguity levels. Trials are aligned to face stimulus onset (left grey bar, fixed 1 s duration). Bin size is 500 ms and step size is 100 ms. Neurons from permutation were averaged across 500 runs. Shaded area denotes $\pm$ s.e.m. across neurons. Dashed horizontal lines indicate the 95% confidence interval estimated from the permuted distribution. Asterisk indicates a significant difference between the observed average across neurons versus permuted average across neurons in that bin (permutation $P < 0.05$, Bonferroni-corrected). (**c–l**) Summary of the effect size across all runs. Effect size was computed in a 1.5-s window starting 250 ms after stimulus onset (single fixed window, not a moving window) and was averaged across all neurons for each run. Grey and red vertical lines indicate the chance mean effect size and the observed effect size, respectively. The observed effect size was well above 0, whereas the mean effect size in the permutation test was near 0. (**c**) Regression model for morph levels (permutation $P < 0.001$). (**d**) Regression model for ambiguity levels ($P < 0.001$). (**e–g**) Regression model for decision of emotion (fear or happy) with (**e**) all faces ($P < 0.001$), (**f**) all ambiguous faces (all morphed faces; $P < 0.001$), (**g**) the most ambiguous faces (40–60% morph; $P = 0.006$) and (**h**) the most ambiguous faces with equal number of fear/happy responses for each identity ($P = 0.002$). (**i–l**) Regression model for confidence judgment (**i**) with all faces ($P < 0.001$), (**j**) at the 30% fear/70% happy morph level ($P < 0.001$), (**k**) at the 50% fear/50% happy morph level ($P = 0.026$) and (**l**) at the 30% fear/70% happy morph level ($P = 0.028$).

independent populations (NS; that is, the overlap of 2/234 was no different than expected by chance). There was also a difference in the distribution of the two populations of neurons (see the last section of Results).

We performed additional experiments to show that amygdala neurons not only encoded emotion ambiguity along the fear–happy dimension but also along the anger–disgust dimension. For this control, we recorded in total 57 neurons (mean firing rate $> 0.2$ Hz) from two patients. Using the same selection procedures, we found 9 amygdala neurons (15.8%; binomial $P < 0.0001$) with a significant trial-by-trial correlation with the three levels of ambiguity. This suggests that the amygdala encodes emotion ambiguity in a domain-general manner. Similar to fear–happy morphs, there were also more neurons (7/9) with maximal firing rate for anchor faces than for ambiguous faces (2/9).

Finally, additional behavioural control analysis showed that the ambiguity signal was not primarily driven by valence or intensity dimensions (Supplementary Fig. 1f–q and Supplementary Notes). Adding the mean intensity ratings from the control subjects as covariate in the regression model used to select ambiguity neurons revealed qualitatively similar results (Supplementary Notes). Furthermore, some amygdala neurons responded to the specific identity of faces[36] and our four facial identities did not have exactly the same valence and intensity judgments (Supplementary Fig. 1h,k,n,q). However, a separate control analysis for each facial identity showed that encoding of emotion degree and ambiguity was not driven by differences in facial identity (Supplementary Notes).

Together, this single-neuron analysis reveals two populations of neurons in the human amygdala: one encoding emotion ambiguity, and the other encoding the degree of emotions.

**Regression analysis of emotion degree and ambiguity**. The electrophysiological results presented so far show that there are independent sets of single neurons that track emotion degree and ambiguity. How representative were the subsets of cells described so far of the entire population of amygdala neurons recorded? We next conducted a population regression analysis of all recorded cells, regardless of whether they were selected as emotion tracking or ambiguity coding. Note that this approach is not sensitive to the direction of modulation (that is, which condition had a greater firing rate) because it relies on an effect-size metric.

We first constructed a moving-window regression model (window size 500 ms, step size 100 ms) for every neuron ($n = 234$) and used this model to estimate how much of the trial-by-trial variability in firing rate could be attributed to the factors of time, morph levels and ambiguity levels. We quantified the proportion of variance explained using a metric of effect size, $\omega^2(t)$ (see Methods section). As expected, the population conveyed information about both the emotion degree (Fig. 5a) and emotion ambiguity (Fig. 5b). This was further confirmed by the average effect size within a 1.5-s window starting 250 ms after stimulus onset (Fig. 5c,d). Comparing the observed effect size against the null distribution revealed above-chance performance for both emotion tracking (Fig. 5c, permutation test with 500 runs: $P < 0.001$) and ambiguity coding (Fig. 5d; permutation test with 500 runs: $P < 0.001$). Notably, population regression analysis confirmed that amygdala neurons as a population from both brain sides encoded both emotion degree and ambiguity (but right amygdala had a weaker effect) (Supplementary Fig. 8k–n). Therefore, without selection of significant neurons, amygdala neurons as a population encode both emotion degree and ambiguity.

**Population regression analysis of decision and confidence**. A key goal of our study was to create sufficiently ambiguous stimuli such that, for the identical stimulus (sensory input), decisions vary between trials. This was the case for the most ambiguous morph levels, for which a trial had an approximately 50% chance of being classified as fearful or happy (Fig. 1c). We therefore next examined how the neuronal activity for this subset of stimuli correlated with the decision.

For this, we used the previously constructed regression model but with the decision (fear or happy) as the independent variable rather than stimulus properties as used before. Note that, during unambiguous trials, the stimulus property (ground truth) is identical to the decision, but for ambiguous stimuli, the decision varies independently of the stimulus. As expected, this model explained a significant proportion of variance when considering all trials (Fig. 5e; permutation test with 500 runs: $P < 0.001$). Crucially, however, this model also explained a significant proportion of variance when only considering all faces other than the anchors (all morphed faces; Fig. 5f; permutation test with 500 runs: $P < 0.001$) as well as when only considering the most ambiguous faces (40–60% morph; Fig. 5g; permutation test with 500 runs: $P = 0.006$) and even with a subset of the most ambiguous trials with equal numbers of fear/happy responses for each facial identity (Fig. 5h; permutation test with 500 runs: $P = 0.002$). Subjective confidence judgments can also co-vary independently for identical stimuli and we therefore next investigated whether stimulus-evoked neuronal activity co-varied with confidence. The regression model revealed that neuronal responses significantly co-varied with levels of confidence both for all trials (Fig. 5i; permutation test with 500 runs: $P < 0.001$) and at individual morph levels (Fig. 5j–l). As a control, we used identical numbers of trials for both decisions and found qualitatively the same results for both emotion judgment and

confidence judgment (Supplementary Fig. 8). Together, this shows that the response of human amygdala neurons co-varied with two subjective decision variables, even for identical stimuli: fear/happy and confidence.

**Comparisons between approaches**. Based on mapping of the lesions in the amygdala lesion patients (showing lesioned basolateral nuclei but spared centromedial nuclei) as well as the functional organization of the amygdala, the abnormal emotion judgements, but intact confidence judgements, given by amygdala lesion patients suggested that emotion-tracking neurons might be located in the basolateral nucleus while ambiguity-coding neurons might be located in the centromedial nucleus. We therefore next mapped the single-unit recording sites from our neurosurgical patients onto these amygdaloid nuclei (see Methods). We found that basolateral nuclei (BL and La) not only contained emotion-tracking neurons as expected, but also ambiguity-coding neurons (Supplementary Table 1). Differential anatomical distribution of neuronal response types thus does not explain the lesion results. However, notably, our fMRI data suggested that the activation by emotion degree (Fig. 2a) was centred primarily in the basolateral nucleus (only 1 voxel in the central nucleus; using the atlas of ref. 37), consistent with the altered emotion judgement in lesion patients and the distribution of emotion-tracking neurons (both of which also involved primarily the BLA). Also, activation by emotion ambiguity (Fig. 2c) also appeared in the basolateral nucleus (all voxels), consistent with the distribution of ambiguity-coding neurons (also see Fig. 2g for peak voxel activity). These commonalities should, however, be considered cautiously, given the limited spatial resolution to differentiate individual amygdala nuclei across all our measures, especially fMRI.

With the fMRI study, we were able to detect effects of emotion ambiguity and effects of fear degree in the right and left amygdala, respectively (Fig. 2a–e). Here we found consistent results also with our single-neuron recordings: consistently, the distribution of emotion-tracking neurons showed a similar difference in laterality. A significantly higher proportion of emotion-tracking neurons was in the left versus right amygdala (left: 16.6%, right: 6.9%; $\chi^2$-test: $P = 0.018$; Supplementary Table 1), and interestingly, 11 out of the 12 neurons showing increasing firing rate with the degree of happiness were in the left amygdala, consistent with the BOLD signal in the left amygdala that increased as a function of happy degree. Although different sides of the amygdala might focus on encoding different aspects of the emotion, it is worth noting that both sides of the amygdala encoded both aspects to some extent (Fig. 2f), even though the BOLD signal did not reach statistical significance in all cases. This is consistent with the observation of both types of neurons bilaterally and that ambiguity-coding neurons were observed in approximately equal proportions (left: 11.8%, right: 13.8%; NS; note the average BOLD response encoding ambiguity in the left amygdala in Fig. 2f). Furthermore, population regression analysis showed that amygdala neurons as a population encoded both emotion degree and ambiguity without selection of significant neurons, consistent with the parametric modulation of fMRI BOLD responses (Fig. 2).

Above, we have analysed two types of amygdala neurons: emotion-tracking and ambiguity-coding neurons. However, the majority of neurons were not be classified as either (73.1%). Finally, we analysed the overall mean firing rate of all recorded neurons ($n = 234$) to investigate the overall activity of amygdala neurons in response to faces and how this response compared to the BOLD signal in our fMRI study, a directional approach (unlike population regression analysis) that can show which

condition has a greater normalized firing rate. This revealed that, after stimulus onset, the overall mean firing rate increased most in response to the anchor faces but least to the most ambiguous faces (one-way repeated ANOVA on the mean firing rate: $F(2,466) = 25.9$, $P = 2.12 \times 10^{-11}$, $\eta^2 = 0.038$; Supplementary Fig. 9). Similarly, the population of all recorded neurons also differentiated the two levels of ambiguity (paired $t$-test: $t(233) = 4.93$, $P = 1.58 \times 10^{-6}$, $g = 0.21$; Supplementary Fig. 9c,d,g,h), but the difference between anchor versus intermediate ambiguous faces was similar to that between intermediate versus the most ambiguous faces (paired $t$-test: $t(233) = 1.31$, NS, $g = 0.13$). Together, this shows that the average activity was dominated by ambiguity: neural activity was strongest for the least ambiguous faces. Similar BOLD activation profile was observed for the parametrical modulation by emotion ambiguity as we presented earlier (Fig. 2c–e).

## Discussion

We used happy–fear morphed faces to test for neural representations of emotion degree and emotion ambiguity in the human amygdala across three different approaches. Patients with amygdala lesions had an abnormal sensitivity to the degree of fear but showed normal metrics of ambiguity (confidence and RT). By contrast, our fMRI study showed that the BOLD signal within the amygdala decreased with both the degree of fearfulness and the categorical emotion ambiguity, albeit on different sides of the brain. Finally, our electrophysiological study revealed one population of neurons that tracked the degree of fear or happiness in a face while another population of neurons primarily tracked decreasing categorical ambiguity (the uncertainty that a stimulus is either fear or happy). Taken together, these findings argue for the coding of both emotion intensity and categorical ambiguity in the amygdala. Crucially, we found that these effects could only be fully disentangled at the level of single neurons.

We used a unique combination of approaches to address the debate of whether the human amygdala encodes the degree of fear and happiness and/or categorical ambiguity between emotions. Different methods measure different signals and have therefore often pointed to somewhat different conclusions, likely accounting in good part for discrepant conclusions in the literature. Although we used identical stimuli and task, our three different methods still produced somewhat different conclusions. Since our single-unit data clearly shows that the amygdala encodes both emotion degree and ambiguity, more macroscopic methods (fMRI, lesion) can provide evidence for either emotion degree or ambiguity coding— given that both types of neurons are intermingled, there is in principle signal to produce either interpretation.

It is important to note that the relationship between the BOLD-fMRI signal and neuronal population activity in general remains unclear. For example, there is a marked dissociation during perceptual suppression in non-human primates[38], and we found that the neuronal population activity matched the direction of BOLD signals for emotion ambiguity but not degree. An estimation of the exact proportion of each type of emotion-tracking neurons and the coupling between the BOLD and electrophysiological signals was limited by the number of neurons that we could sample; future studies with substantially more recording of neurons will be necessary to answer these questions.

We previously used sparsely sampled 'bubbled faces' to argue that neurons in the human amygdala encode subjective fear/happy judgments rather than objective stimulus properties[35]. However, with those stimuli, different random parts of faces are revealed in every trial. This makes it difficult to determine the relationship between stimulus input and behavioural output, because both variables change trial-by-trial. In the present study, we used morphed stimuli that disambiguate stimulus properties

and subjective behavioural judgments, because in different trials, different decisions are made for the same stimuli. Using this approach, we here conclusively demonstrated that responses of human amygdala neurons reflect the subjective decision in trials where only the decision, but not the sensory input, is variable. Note that, when selecting neurons based on a contrast such as fear versus happy or ambiguity level, the two variables (sensory input and decision) co-vary and such selections are therefore not appropriate to disambiguate these two scenarios. Instead, our approach was to use a model that considers the activity of all recorded neurons, regardless of selection.

Our conclusions rest on comparing two emotions (fearful and happy), and we chose the fear/happy contrast because of the large existing literature on this pair of emotions, which has been used in a series of prior studies of the amygdala[7,9,22,35]. As a control, we also tested another pair of emotions (anger versus disgust) and found similar conclusions. It remains an open question whether our results generalize to other emotions, or indeed might generalize to ambiguity about stimuli other than facial emotions, or about decisions that do not involve emotions.

The existing literature uses the term ambiguity for two entirely different constructs, and it is important to distinguish the two to properly frame our results. The first definition, which we used throughout, refers to the closeness to categorical boundaries (see ref. 39 for a classical example of perceptual ambiguity that uses the same meaning of 'ambiguity' as we did here). The second definition, which we did not refer to here, is related to missing information about stimuli in economic decision-making. In studies of face processing such as ours, the probability of stimuli belonging to one or the other category (that is, fear/happiness, trustworthy/untrustworthy) is known. Indeed, increased amygdala responses to the second type of ambiguity have been found in studies on decision-making that do not involve ambiguous choices between two facial attributes[24,26,40,41]. In contrast to this finding, other studies find increased amygdala responses to certainty in tasks where an ambiguous choice is made between two options for a face[27–31]. Thus fMRI studies on categorical ambiguity are consistent with our present result by showing that the amygdala tracks the categorical certainty and often shows a minimal response when categorical ambiguity is highest. Therefore, our results fit with a subset of studies on the amygdala's role in coding perceptual ambiguity/certainty, specifically those studies that investigate the same construct of 'ambiguity' as ours[27–31].

Notably, emotion ambiguity and certainty are closely related and inversely correlated, and these neurons might encode emotion certainty or confidence in emotion judgment. Here we interpret any change in firing rate, or in BOLD signal, as carrying information, and therefore do not further interpret the sign of that change. Although all our stimuli should be equally attended given the task demands of having to make judgments about them, we acknowledge that task difficulty, attention, mental efforts and RT are of course all intercorrelated to some extent, and future studies will be needed to further distinguish the possible contribution of attentional effects in our study. Furthermore, future studies will be necessary to investigate whether ambiguity-coding neurons are from a circuit separate from the emotion-tracking neurons. Alternatively, ambiguity-coding neurons might pool the activity of emotion-tracking neurons to generate the ambiguity signal, that is, ambiguity-coding neurons effectively code for the absolute degree of emotions. The second hypothesis predicts that ambiguity-coding neurons would respond later in time relative to emotion-tracking neurons, a hypothesis that remains to be tested.

In our three amygdala lesion patients, most of the basolateral complex of the amygdala was lesioned but the centromedial

nucleus was intact, an important difference to the complete amygdala lesion that has been studied in detail in patient S.M. (refs 32,42). The BLA is the primary source of visual input to the amygdala and the centromedial amygdala is a primary source of output to subcortical areas relevant for the expression of innate emotional responses and associated physiological responses[43]. The centromedial amygdala provides most of the amygdala projections to hypothalamic and brainstem nuclei that mediate the behavioural and visceral responses to fear[44–46]. Furthermore, the projection neurons in the central nucleus are mostly inhibitory, and are, in turn, inhibited by inhibitory intercalated cells in the lateral and basal amygdala. Disinhibition through this pathway is thought to lead to the expression of emotional responses[43]. Although direct evidence of the role of amygdala subregions in threat and fear processing comes predominantly from rodent lesion research[47] and optogenetics[48], our present finding of a lowered threshold for reporting fear for morphed faces is consistent with prior human amygdala lesion results: patient S.M., who has complete amygdala damage including both basolateral and centromedial nuclei, showed an increased threshold for reporting fear (the opposite of our finding)[2], while another group of five patients with only BLA damage demonstrated a hyper-vigilance for fearful faces (similar to our finding)[22,23]. A putative mechanism explaining this difference between the two types of amygdala lesion patients comes from optogenetic work in rodents. Specific activation of the terminals that project from the BLA to the central nucleus reduces fear and anxiety in rodents, whereas inhibition of the same projection increases anxiety[49]. It is thus possible that the partial amygdala lesions in our three subjects removed a normal inhibitory brake on fear sensitivity and resulted in exaggerated sensitivity to emotion mediated by the disinhibited central nucleus, just like in the prior studies[22,23].

It is worth noting that the intact judgment of ambiguity in amygdala lesion patients suggests that the amygdala's response to ambiguity is not an essential component for behavioural judgments of emotion ambiguity and that such judgments may sufficiently rely on structures elsewhere in the brain that also represent information about ambiguity. Future studies could further probe this issue by conducting fMRI studies like ours but in patients with lesions to the amygdala.

Decisions about faces are frequently ambiguous, including those about facial emotions, and optimal decision-making thus requires an assessment of ambiguity. We thus expect that an assessment of uncertainty is a crucial variable represented in areas concerned with decision-making about faces. We have further shown that the activity of amygdala neurons correlates with the confidence in emotion judgment. Together, this shows that two closely related variables with meta-information about the decision itself (fear/happy) are represented in the amygdala, one based on objective discriminability of the stimuli and the second based on the subjective judgment of their discriminability: ambiguity and confidence.

Functional neuroimaging and electrophysiological studies of the amygdala frequently report lateralized amygdala activity, indicating a clear hemisphere-specific processing difference between the left and right amygdalae. Such lateralization has been observed in a wide range of tasks, with diverse approaches, and across species, such as category-specific single-neuron responses to animals in the right human amygdala[50], differential electrophysiological and biochemical properties of the amygdala neurons in pain processing in rats[51] and left amygdala BOLD-fMRI activation to fearful eye whites in humans[9]. In the present study, we also found lateralized fMRI activation: the left amygdala by emotion degree and the right amygdala by emotion ambiguity. The left amygdala activation is consistent with the left-lateralized response reported to fearful eyes[9,52], and the right amygdala activation by ambiguity may be related to findings showing that the right amygdala BOLD signal correlates with the overall strength of a face's trustworthiness[27]. A possible caveat to the study of lateralization of single-neuron responses with epilepsy patients is that people with epilepsy have a higher rate of abnormal lateralization of function, with often more bilateral representations than people without epilepsy[53]. It is also worth noting that the laterality of neuronal response matched BOLD-fMRI in some aspects but not the others in the present study. High-resolution fMRI, precise localization of the electrodes and more neurons recorded will be necessary in future studies to further address this question.

In conclusion, our electrophysiological results demonstrate that neurons in the amygdala can signal both the categorical ambiguity and the degree of emotional facial expressions and that they do so in largely non-overlapping neuronal populations. Findings from fMRI would thus be expected also to show coding of these two parameters, depending on the details of task demands and statistical power. Notably, both the fMRI and the single-unit recordings suggested a relatively lateralized representation of emotion degree in the left amygdala (primarily tracking the degree of happiness in the face), a finding that may be related to the lexical demands of classifying our stimuli into discrete emotions. A methodological contribution of our study is that, while all three methods we used support a role for the amygdala in processing facial emotion, the detailed conclusions about such a role may look quite different, depending on the method used. While unrealistic in most cases, we would nonetheless advocate for single-neuron electrophysiological studies as an essential complement to all more macroscopic approaches in order to help constrain interpretations.

## Methods

**Subjects.** All participants provided written informed consent according to protocols approved by the institutional review boards of the Huntington Memorial Hospital, Cedars-Sinai Medical Center, the California Institute of Technology and the South China Normal University.

There were 14 sessions from 9 neurosurgical patients in total (3 patients did two sessions and 1 patient did three sessions. Each session was treated as an independent sample for behavioural analysis. Supplementary Table 1). Nineteen healthy, right-handed volunteers (15 female, mean age and s.d. $20.9 \pm 2.02$ years) participated in the fMRI experiments and an independent sample of 15 undergraduates served as healthy controls.

AP, AM and BG are three patients with selective bilateral amygdala lesions as a result of Urbach–Wiethe disease[54]. AM and BG are monozygotic twins, both with complete destruction of the BLA and minor sparing of anterior amygdaloid and ventral cortical amygdaloid parts at a rostral level, as well as lateral and medial parts of the central amygdaloid nucleus and the amygdalohippocampal area at more caudal levels[55]. The details of the neuropsychological assessments of these patients have been described previously[42,55]. Anatomical scans of the lesions are shown in Supplementary Fig. 2.

**Stimuli and task.** We asked subjects to discriminate between two emotions, fear and happiness, because these emotions are distinguished by particular facial features[56]. We selected faces of four individuals (two females) each posing fear and happiness expressions from the STOIC database[57], which are expressing highly recognizable emotions. Selected faces served as anchors and were unambiguous exemplars of fearful and happy emotions as evaluated with normative rating data provided by the creators. To generate the morphed expression continua for this experiment, we interpolated pixel value and location between fearful exemplar faces and happy exemplar faces using a piece-wise cubic-spline transformation over a Delaunay tessellation of manually selected control points. We created five levels of fear–happy morphs, ranging from 30% fear/70% happy to 70% fear/30% happy in steps of 10% (see Fig. 1b and Supplementary Fig. 1a for all stimuli). Low-level image properties were equalized by the SHINE toolbox[58] (The toolbox features functions for specifying the (rotational average of the) Fourier amplitude spectra, for normalizing and scaling mean luminance and contrast, and for exact histogram specification optimized for perceptual visual quality). In neurosurgical patients C26, C27, H42, H43 and H44 (9 sessions in total), in each block, each level of the morphed faces was presented 16 times (4 repetitions for each identity) and each anchor face was presented 4 times (1 for each identity). In all other neurosurgical patients (5 sessions in total) and all other subjects, each anchor face and each

morphed face was presented the same number of times (12 times, 3 repetitions for each identity). Neurosurgical patients performed 2–5 blocks, amygdala lesion patients all performed 4 blocks, fMRI subjects all performed 2 blocks and behavioural control subjects all performed 3 blocks. All trials were pooled for analysis.

A face was presented for 1 s followed by a question prompt asking subjects to make the best guess of the facial emotion, either by pushing the left button (using left hand) to indicate that the face was fearful or by pushing the right button (using right hand) to indicate that the face was happy. Subjects were instructed to respond as quickly as possible. After emotion judgment and a 500 ms blank screen, subjects were asked to indicate their confidence of the preceeding decision, by pushing the button '1' for 'very sure', '2' for 'sure' or '3' for 'unsure'. For both prompts, subjects were allowed 2s to respond. If this time was exceeded, the trial was aborted and there was a beep to indicate a time out. No feedback was displayed after either question. The intertrial interval was randomized between 1 and 2 s. The order of faces was randomized for each subject. Subjects practiced 5 faces before the experiment to familiarize themselves with the task. At the end of each block, the overall percentage of 'correct answers' was displayed. One neurosurgical patient (C34) did not provide confidence rating due to difficulty in understanding the instruction.

Using an operational definition of ambiguity—the variability in choices (the percentage of choices that are not the same as the dominant choice, for example, the percentage of choosing fear for a 30% fear/70% happy face where happy judgment is the dominant choice), we found that the variability in choices for three levels of ambiguity increased in similar steps in neurosurgical patients (anchor: $3.15 \pm 3.08$ (mean ± s.d.), intermediate: $14.1 \pm 7.97$, high: $32.3 \pm 5.55$; the difference between intermediate and anchor: $10.9 \pm 7.20$, the difference between high and intermediate: $18.2 \pm 10.0$; paired $t$-test on difference: $t(13) = 1.69$, NS). Furthermore, the psychometric curves were symmetric demonstrating that our grouping of ambiguity levels was not biased.

**Analysis of behaviour.** We used a logistic function to obtain the smooth psychometric curves shown in Fig. 1c:

$$P(x) = \frac{P_{\inf}}{1 + e^{-\alpha(x - x_{\text{half}})}}$$

where $P$ is the percentage of trials in which faces were judges as showing fear, $x$ is the morph level, $P_{\inf}$ is the value when $x$ approaches infinity (the curve's maximum value), $x_{\text{half}}$ is the symmetric inflection point (the curve's midpoint) and $\alpha$ is the steepness of the curve. $P_{\inf}$, $x_{\text{half}}$ and $\alpha$ were fitted to the observed data ($P$ and $x$). We derived these parameters for each subject or recording session.

Flatter curves (smaller $\alpha$) suggest that subjects were less sensitive to the change in emotion degree and vice versa for steeper curves (larger $\alpha$).

**Analysis of spikes.** Only units with an average firing rate of at least 0.2 Hz (entire task) were considered. Only single units were considered. Trials were aligned to face onset, and the baseline firing rate was calculated in a 1 s interval of blank screen right before face onset. Average firing rates (peristimulus time histogram (PSTH)) were computed by counting spikes across all trials in consecutive 250 ms bins. Comparisons between morph levels were made using a one-way ANOVA at $P < 0.05$ and Bonferroni-corrected for multiple comparisons in the group PSTH. PSTHs with different bin sizes were analysed and qualitatively the same results were derived.

**Model comparison.** We compared the linear regression model to one logistic model (sigmoidal) and one step-function model using the Akaike Information Criterion (AIC), which measures the relative quality of statistical models for a given set of data[59]. The AIC is founded on information theory and it offers a relative estimate of the information loss when a given model is used to represent the process that generates the data. In doing so, it deals with the trade-off between the goodness of fit of the model and the complexity of the model. Note that the AIC only estimates the quality of each model relative to the other models in comparison, providing a means for model selection, rather than the absolute quality of the model in a sense of testing a null hypothesis.

For each model, we have (see page 63 and 66 of ref. 59):

$$\text{AIC} = n \cdot \ln \frac{\text{RSS}}{n} + 2k + \frac{2k(k+1)}{n - k - 1}$$

where $n$ is the sample size (the number of observations), $k$ is the number of parameters of the model and RSS is the residual sum of squares between the observed data and the fitted data. Note that we here corrected the relatively small sample size ($n/k < 40$).

Using the same logistic function as to fit the behaviour (see above), we have

$$\Delta\text{AIC} = n \cdot \ln \frac{\text{RSS}_{\text{Logistic}}}{\text{RSS}_{\text{Linear}}} + 2(k_{\text{Logistic}} - k_{\text{Linear}})$$
$$+ \left( \frac{2k_{\text{Logistic}}(k_{\text{Logistic}} + 1)}{n - k_{\text{Logistic}} - 1} - \frac{2k_{\text{Linear}}(k_{\text{Linear}} + 1)}{n - k_{\text{Linear}} - 1} \right)$$

where $k_{\text{Logistic}} = 3$, $k_{\text{Linear}} = 2$ and $n = 7$ (7 morph levels).

A positive $\Delta$AIC indicates that the linear model has less information loss compared to the logistic model, suggesting that the linear fit is more appropriate.

We also fitted a step function of $f(x) = a$ when $x \geq c$, and $f(x) = b$ when $x < c$. We fitted the parameters using multidimensional unconstrained nonlinear minimization (Nelder–Mead) method to minimize the least squares. We used the same method to compute the $\Delta$AIC. Note that the step function model also has three parameters.

**Single-neuron ROC analysis.** Neuronal ROCs were constructed based on the spike counts in a time window 250–1,750 ms after stimulus onset (the same time window as all neuron selections). We varied the detection threshold between the minimal and maximal spike count observed, linearly spaced in 20 steps. The AUC of the ROC was calculated by integrating the area under the ROC curve (trapezoid rule). The AUC value is an unbiased estimate for the sensitivity of an ideal observer that counts spikes and makes a binary decision based on whether the number of spikes is above or below a threshold. We defined the category with higher overall firing rate as 'true positive' and the category with lower overall firing rate as 'false positive'. Therefore, the AUC value was always $> 0.5$ by definition.

**Regression analysis.** We used the regression model $S(t) = \alpha_0(t) + \alpha_1(t) \cdot L$ to estimate whether the firing rate $S$ was significantly related to one of the following factors ($L$): morph level (1–7), ambiguity level (1–3), emotion judgment (0/1) and confidence judgment (1–3). Separate models were fit for each factor. The model was fit to the total spike count in a 500 ms window that was moved in steps of 100 ms for moving-window analysis and in a 1.5-s window starting 250 ms after stimulus onset for fixed-window analysis. We estimated the significance of each factor using $\omega^2$ as described previously[60], which is less biased than percentage variance explained[61]. Here, $\omega_i^2 = \frac{\text{SS}_i - df_i \cdot \text{MSE}}{\text{SS}_{\text{tot}} + \text{MSE}}$, where $\text{SS}_i$ is the sum of squares of factor $i$, $\text{SS}_{\text{tot}}$ is the total sum of squares of the model and MSE is the mean square error of the model. Effect sizes were calculated using the effect size toolbox[62]. We averaged $\omega^2(t)$ across all neurons. The null distribution was estimated by randomly scrambling the labels and fitting the same model. This was repeated 500 times to estimate the statistical significance.

**Electrophysiology.** We recorded bilaterally from implanted depth electrodes in the amygdala from patients with pharmacologically intractable epilepsy. Target locations in the amygdala were verified using postimplantation structural MRIs. At each site, we recorded from eight 40 μm microwires inserted into a clinical electrode as described previously[13]. Efforts were always made to avoid passing the electrode through a sulcus and its attendant sulcal blood vessels, and thus the location varied but was always well within the body of the targeted area. Microwires projected medially out at the end of the depth electrode and examination of the microwires after removal suggests a spread of about 20–30 degrees. Bipolar wide-band recordings (0.1–9 kHz), using one of the eight microwires as reference, were sampled at 32 kHz and stored continuously for off-line analysis with a Neuralynx system (Digital Cheetah; Neuralynx, Inc.). The raw signal was filtered with a zero-phase lag 300-3 kHz bandpass filter and spikes were sorted using a semiautomatic template matching algorithm[63]. Units were carefully isolated and spike sorting quality were assessed quantitatively (Supplementary Fig. 5).

**Electrode mapping.** Preoperative and postoperative images were aligned through an initial automated linear coregistration (using the FLIRT module of FSL[64]), followed by manually guided nonlinear thin-plate-spline warping[65]. Control points for the nonlinear warping were selected according to local anatomical features that corresponded unambiguously between preoperative and postoperative images, which included features bounding the structures of interest as closely as imaging artifacts allowed. Amygdaloid nuclei were projected into the subject's volume through nonlinear warping of structures derived from a stereotactic atlas of the human brain[66]. Atlas-derived structures were projected by first aligning the outer boundary of the atlas-derived amygdala with an amygdala boundary surface obtained through automated subcortical segmentation (FSL FIRST), with manual editing of the latter to improve accuracy, when necessary. These respective surfaces then provided control points for a nonlinear warping from the atlas space to the subject's preoperative image.

**Lesion mapping.** Lesion extents were identified and labeled using ITK-SNAP (version 3.2, University of Pennsylvania)[67]. Calcified areas of the lesions appeared hypointense in the T1-weighted structural images and lesion boundaries were in most cases well delineated. Internal signal heterogeneity was observed in all patients' lesions and was most pronounced in AP. In AM and BG, parts of the lesion margin were contiguous with cerebral spinal fluid spaces, which is also hypointense in T1-weighted images. In these areas, the margin was inferred by extrapolation of local tissue boundaries.

**fMRI imaging acquisition.** MRI scanning was conducted at the South China Normal University on a 3-Tesla Tim Trio Magnetic Resonance Imaging scanner (Siemens, Germany) using a standard 12-channel head-coil system. Whole-brain

data were acquired with echo planar T2*-weighted imaging (EPI), sensitive to BOLD signal contrast (31 oblique axial slices, 3 mm thickness; TR = 2,000 ms; TE = 30 ms; flip angle = 90°; FOV = 224 mm; voxel size: $3 \times 3 \times 3$ mm$^3$). T1-weighted structural images were acquired at a resolution of $1 \times 1 \times 1$ mm$^3$.

**fMRI face localizer task.** We used a standard face localizer paradigm[68] to localize areas activated by faces. The face localizer task consisted of four blocks of faces interleaved with five blocks of a sensorimotor control task in a fixed order (CFCACFCAC; C = control block, F = task block with fear expressions, A = task block with anger expressions). In each face-processing block, six images from the STOIC database[57], three of each gender and target emotion (angry or fearful), were shown. Subjects viewed a trio of faces (expressing either anger or fear) for a fixed duration of 5 s and selected one of the two faces at the bottom that had the same facial expression as the target face at the top by button press (right or left, within 5 s). Six different sets of geometric forms (circles and vertical and horizontal ellipses) were used in the sensorimotor control block. Subjects viewed a trio of simple geometric shapes for a fixed duration of 5 s and selected one of two the shapes at the bottom that had the same geometric shape as the target shape at the top by button press (right or left, within 5 s). Each block consisted of six trials (30 s) and the entire duration of the localizer task was 270 s.

The face localizer task revealed reliable differences in BOLD signal between faces and geometric shapes in bilateral amygdala (peak: MNI coordinate: $x = -24$, $y = -3$, $z = -18$, $Z = 4.27$, 43 voxels; $x = 27$, $y = -3$, $z = -21$, $Z = 5.03$, 60 voxels, FWE $P < 0.05$, SVC based on an anatomical amygdala ROI; Supplementary Fig. 4a), consistent with other findings[10]. We defined the functional ROI in the amygdala as all voxels in the amygdala identified by the localizer task (face > objects; 43 voxels for left amygdala and 60 voxels for right amygdala). Other cortical regions known to be involved in face processing, such as the fusiform face area, visual cortex, inferior frontal gyrus, superior temporal sulcus/gyrus, dorsal medial prefrontal cortex and dorsal lateral prefrontal cortex, were also more activated for faces compared to objects, and objects minus faces activated the ventral anterior cingulate cortex and posterior cingulate cortex (Supplementary Fig. 4a), consistent with the face processing network shown in previous studies[10].

**fMRI face morph task.** We used the identical task as used for all other subject groups (Fig. 1a), except that we omitted the confidence rating part and increased the duration of intertrial interval from 1–2 s to 2–8 s (jittered randomly with a uniform distribution). Subjects reported faces as fearful or happy by pressing a button on the response box with either their left (fearful) or right (happy) index fingers.

Using the above functional ROI, we again found robust activation of the bilateral amygdala when comparing face versus baseline (Supplementary Fig. 4o; peak: $x = -21$, $y = -6$, $z = -18$, $Z = 4.89$, 22 voxels, and $x = 21$, $y = -6$, $z = -18$, $Z = 3.21$, 5 voxels, FWE $P < 0.05$, SVC). No significant activation was found for the reversed contrast (FWE $P > 0.05$).

**fMRI imaging analysis.** Neuroimaging data were preprocessed and analysed using SPM8 (www.fil.ion.ucl.ac.uk/spm/). The first four volumes were discarded to allow the MR signal to reach steady-state equilibrium. The EPI images were sinc interpolated in time for correction of slice-timing differences and realigned to the first scan by rigid-body transformations to correct for head movements. Utilizing linear and nonlinear transformations and smoothing with a Gaussian kernel of full-width-half maximum 6 mm, EPI and structural images were coregistered to the T1 MNI 152 template (Montreal Neurological Institute, International Consortium for Brain Mapping). Global changes were removed by high-pass temporal filtering with a cutoff of 128 s to remove low-frequency drifts in signal.

In the localizer task, we used a block design and modelled BOLD responses using a general linear model (GLM), with the two regressors for face and object conditions modelled as boxcar functions convolved with a 2-gamma hemodynamic response function.

In the face morph task, we used an event-related design. In the GLM design matrix, for every subject we estimated a GLM with autoregressive order 1 [AR(1)] and the following regressors (R): R1 at face presentation; R2 at face presentation modulated by fear levels: 100%, 70%, 60%, 50%, 40%, 30%, 0%; R3 at face presentation modulated by ambiguity levels: anchor, 30%/70% morph, 40–60% morph; and R4 at fixation presentation. Because both the RT and ambiguity levels were correlated with confidence, we also repeated the analysis by adding the $z$-normalized RT (for each subject) as one additional modulator and orthogonalized it to earlier modulators using the default SPM orthogonalization function. We derived similar results when adding RT as a modulator (Supplementary Fig. 4i–n). To compute the time course of ambiguity coding, we built a second GLM with AR(1) and the following regressors: R1 at face presentation with anchor; R2 at face presentation with 30%/70% morph; and R3 at face presentation with 40–60% morph. We found similar results for a model with four ambiguity levels (anchor, 30%/70% morph, 40%/60% morph, and 50% morph; Supplementary Fig. 4f–h).

For all GLM analyses, six head-motion regressors based on SPM's realignment estimation routine were added to the model (aligned to the first slice of each scan). Multiple linear regression was then run to generate parameter estimates for each

regressor for every voxel. The contrast (difference in beta values) images of the first-level analysis were entered into one-sample $t$-tests for the second-level group analysis conducted with a random-effects statistical model[69]. For the localizer task, we used SVC defined by a priori ROIs of the structural amygdala[70]. For the face morph task, we used a functional ROI defined by the parts of the bilateral amygdala identified in the localizer task. Similar results were found when using the structural ROIs of the bilateral amygdala. Activations in other areas were reported if they survived $P < 0.001$ uncorrected, cluster size $k > 10$.

**Data availability.** The data that support the findings of this study are available from the corresponding authors upon reasonable request. The data are not publicly available due to privacy policies relating to clinical recordings.

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

## Acknowledgements

We thank all patients for their participation and Farshad Moradi for providing the morphing algorithm. This research was supported by the Autism Science Foundation (to S.W.), the Simons Foundation (to R.A.), the Cedars-Sinai Department of Neurosurgery (to U.R.), an NSF CAREER award (1554105 to U.R.) and the NIMH Conte Center (P50MH094258 to R.A.). The funders had no role in study design, data collection and analysis, decision to publish or preparation of the manuscript.

## Author contributions

S.W., R.Y., R.A. and U.R. designed experiments; S.W., R.A. and U.R. wrote the paper. S.W., R.Y., J.M.T., S.Z., S.S., I.B.R, J.M.C., A.N.M. and U.R. performed research. S.W., R.Y., J.M.T., S.Z., C.K., Y.H. and U.R. analysed data. R.H. contributed two patients with amygdala lesions. All authors discussed the results and contributed toward the manuscript.

## Additional information

**Competing interests:** The authors declare no competing financial interests.

