## [Peer Review File · Nature Communications]

Reviewers' Comments:

Reviewer #1 (Remarks to the Author):

The authors have effectively combined 3 studies: one with fMRI, one with intracranial (single neuron depth) recordings and one with patients with amygdala lesions to address the question of whether the amygdala responds to the strength of a given emotion (fear versus happiness) or the ambiguity of the expression (highest at middle morphs in a fear to happiness continuum). The use of the same stimuli across these three datasets allows for the powerful comparison of findings across methodologies.

The methods and statistics are generally solid and the paper is well written. There are a number of issues highlighted below that I think warrant addressing but overall this represents a considerable endeavour and a valuable addition to the literature.

Summary of key results.

The fMRI data indicate a left amygdala response that increased with level of happiness (faces varying from 100% fear to 100% happy through intermediate morphs) and a right amygdala response to low ambiguity. Patients with amygdala lesions showed an increase in fear sensitivity - i.e. less % fear in the morphed expression was required for participants to reach the point of equivalence (50% of faces categorized as fear, 50% categorized as happy). The intracranial data, meanwhile, suggested that independent populations of neurons responded to (i) increasing levels of fearfulness (ii) increasing levels of happiness (or decreasing levels of fearfulness) and (iii) (low) levels of ambiguity (these showed a preference for 100% happy or fearful faces).

Originality

While prior studies have used either a neuropsychological approach, fMRI, or intracranial recordings to examine the role of the amygdala in processing different emotional expressions, to my knowledge this is the first to combine all three approaches, while using the same task to facilitate comparability of findings across approaches, and which has also addressed the question of whether different amygdala neurons encode whether the expression is closer to fear or happiness versus high or low on ambiguity without having a preference between continuum ends.

Issues.

1) As the authors recognize (top of page 15), the use of just two emotions to create a single

continuum of expressions (morphed from 100% fear through intermediate steps to 100% happiness) limits the potential generalizability of the current findings. While worth recognizing as a limitation, this is offset by the value of comparability of the datasets across methods and by the need to balance power against stimulus set size when conducting work with patient groups who tire easily, hence limiting the potential length of data collection.

2) I have a number of concerns regarding the 'ambiguity' analyses that I would be keen to see addressed. The findings from these analyses indicate greater amygdala responsivity to the 100% fear or happiness 'anchor' faces than to more ambiguous intermediate face morphs. There is nicely convergent evidence for this from both the fMRI and intracranial data. (i) However, it is potentially confusing to frame this as having shown 'representation of ambiguity' in the amygdala as readers are likely to assume this means an elevated amygdala response to ambiguous stimuli. (ii) Further, a number of previous studies have similarly reported that responses in the amygdala as well as other face-sensitive region increase for stimuli towards the end of face continua and are lower for mid-continuum faces (see Said et al., *Neuropsychologia* 2011 for a discussion and references to this literature). Here, an alternate theoretical proposal put forward to explain these findings is that activity in regions sensitive to faces, such as the amygdala and FFA increase as faces move away from the centre of 'face space'. Said et al. (2011) address the extent to which this is equally true for an expression continuum and for a control non-social continuum - the findings are not clear cut but worth taking a look at. This alternate theoretical interpretation could apply to the results observed here and warrants discussion. (iii) Also relevant here, is a critique by Aguirre and colleagues who further argue that findings of a stronger response to continuum extremes might simply reflect a methodological confound with adaptation effects being unaccounted for and these being likely to be more pronounced for stimuli in the middle than at the extremes of continua (due to greater average 'closeness' in physical similarity to the stimulus presented previously or afterwards in a randomized series), Kahn & Aguirre, *Neuroimage*, 2011. The extent to which this critique might be pertinent to the analyses in the current study should also be addressed. (iv) The authors state that 'in each block each level of the morphed faces was presented 16 or 12 times (4 or 3 repetitions for each intensity) and each anchor face was presented 4 or 12 times (1 or 3 repetitions for each intensity)'. Why were the anchor faces presented less often? This alone might lead to a greater amygdala response to anchor faces (greater novelty, reduced habituation across presentations). Are there blocks where everything is presented 12 times (this is a little unclear)? If so, it would be good to know whether the reported findings hold for these blocks alone. (v) Finally, if it is indeed the low ambiguity of the anchor faces that is critical, then an issue that also requires discussion is how this fits with other accounts suggesting that the amygdala is sensitive to high levels of ambiguity (e.g. Whalen TICS 2007). Indeed the work on ambiguity reviewed in the Introduction points to the amygdala response varying positively with level of ambiguity - i.e. in the opposite direction to the findings reported but this is not made clear in the discussion.

3) Considering each of the experiments (fMRI, intracranial, lesion) separately, I found the intracranial findings especially a pleasure to read with the analyses employed being more sophisticated and the interpretation of the data more nuanced (though I say this as a fMRI researcher so I may not be fully suited to critique all aspects of the methodology). Here, the modelling of the effect of decision (fear/happy) for 50/50 trials is a nice touch with these findings converging well with previous work (e.g. Pessoa et al., *Cerebral Cortex*, 2006). My only query here is that, from the Table in the Supplements, the majority of low ambiguity responsive neurons and neurons that positively track level of face fearfulness appear to come from a single subject (C26). Might it be possible for the authors do a confirmatory analysis to demonstrate the results hold when sampling equal amounts of data from each patient (or omitting this one patient)? If there are too few emotion or ambiguity responsive neurons in the remaining patients to allow this, this should be addressed as a potential limitation with regards to the generalizability of the findings.

4) With regards to the lesion study, how do the authors reconcile the finding that amygdala lesion patients were 'more likely to judge faces as fearful' (pg 5) with prior reports that patients with amygdala damage show reduced ability to recognize fearful expressions (Adolphs et al., *Nature*, 1994)? Given that both the fMRI findings (increased amygdala response as percentage of happiness goes up and percentage of fearfulness goes down) and the lesion study findings (increased sensitivity to fear in patients with amygdala damage) are the opposite to prior findings it makes me worry a little that there might be something specific to the stimuli used here. It is reassuring on this issue that the intracranial study using the same stimuli found that nearly twice the number of neurons positively tracked fearfulness versus happiness but it still needs addressing. (Ideally I would have been keen to also see findings for fearful and happy expressions each morphed with neutral expressions but I appreciate the difficulty of conducting a multi-method study such as this and that such a request at this stage would not be practical).

5) On a related note, there is generally a bit of a disconnect between the Introduction and Discussion. In the Introduction, the authors review literature indicating that many fMRI studies have reported greater amygdala responses to fearful than happy faces, other evidence for an amygdala response to (high) ambiguity, and findings that amygdala lesions are linked to impoverished recognition of fear. In contrast in the discussion the authors argue that their finding that patients with amygdala lesions show increased sensitivity to fear is consistent with prior findings, no longer referencing the well-known studies highlighted in the introduction that link amygdala lesions to impaired recognition of fear. It is fine to bring up at this point that the literature is not as consistent as people might at first think but this complete U turn comes a bit out of the blue. It is important to interpret the current data in the light of a balanced presentation of the existing findings in the field. If the authors are making the case, as it seems, that the

precise location of the amygdala lesions explains why the current findings differ from those studies reviewed in the Introduction that reported reduced fear recognition in patients with amygdala damage then it is important that they make this explicit and that as part of this they contrast the location of the lesions in the current patients with those in the studies showing deficits in fear recognition. It should also be clarified that this was a posthoc analysis contrary to the initial hypothesis (as the Introduction suggests a fear deficit was expected).

Similar comments apply for other findings. A role of the amygdala in processing 'emotional intensity' was predicted, and it seems from the Introduction that the prediction was that amygdala activity would primarily increase with expression fearfulness, but that there might also be a smaller population of neurons more responsive to expression happiness (as indeed suggested by the intracranial data). If this is the case, it needs to be more clearly stated in the early part of the discussion that the fMRI results did not support this prediction showing instead an increase in amygdala responsivity with level of happiness in the expression morph. Stating that the amygdala responded to emotional intensity but not detailing the directionality of the result, and whether or not it fitted with predictions, is too vague. Further, the statement in the abstract that 'our results indicate that the human amygdala processes ... the degree of fear shown in facial expressions' is arguably misleading if by that the authors include the fMRI finding of increased amygdala response to faces as levels of happiness go up and fearfulness goes down.

Small points

- 1) For the measure of 'the slope of the psychometric curve (emotion sensitivity index, ESI)' why did the authors not derive this directly from the logistic function?
- 2) The statement in the abstract that 'our results indicate that the human amygdala processes both the degree of fear shown in facial expressions and their ambiguity and this effect can *only* be distinguished at the level of single neurons' is too strong. To support this *only* statement, the authors would at least need to conduct more sophisticated fMRI analyses (e.g. using representational similarity (MVPA) or adaptation techniques with reduced or no spatial smoothing of the data) and show that it is still impossible to distinguish responses to morph expression versus responses to morph ambiguity.
- 3) The language based explanation of the lateralization of the amygdala response seems to come a bit out of nowhere and is also oddly placed just following an argument that similar findings are also observed in animals. If the authors want to leave it in they need to add a sentence or two on why linguistic processing would lead to lateralized processing (given the wide audience this journal attracts).
- 4) 'We thus here show, for the first time, a reward-and reward-value independent representation of ambiguity in the amygdala.' This might also be a bit strong, Paul Whalen's group have produced some nice findings on ambiguity that do not involve reward-relatedness.

Reviewer #2 (Remarks to the Author):

The authors investigated whether the amygdala is functionally related to the processing of fearful faces, to ambiguity or both. To address this question, the authors used the same behavioral task, a facial expression categorization task of 4 parametrically morphed faces (happy to fear), in three groups of observers: amygdala lesion patients (behavioral), healthy adults (fMRI), and single-neuron recordings in neurosurgical patients. The results from the lesion and healthy subjects indicate that the amygdala is involved in both fear (lesion and healthy observers) and ambiguity (healthy observers) decoding. The novelty of this study relies on the single neuron recordings, that show in fact that those responses might rely on the presence of two distinct populations within this region: one coding for fear and the other coding for ambiguity. A result that could not be observed with behavioral and fMRI measures alone.

In general, I found this paper very difficult to follow. There are numerous analyses and supplementary experiments, and some of them might not be necessary (for instance, the eye movement study carried out on an additional group of Chinese observers). I would suggest the authors to refocus on their question of interest and include the measures/ experiments that are really necessary to address the question at hand. As it stands, I would personally remove for instance the results based on the lesion patients (see below), the eye movement results (see also below), and keep the most appropriate comparison between the fMRI data in healthy controls with the single neuron recordings. Having said that, I also found a series of methodological and theoretical concerns that seriously dampen my enthusiasm for this part of the work:

- The same group has previously shown that attention towards the eye region is critical to effectively decode fear and that the amygdala plays a critical role in this process (i.e., Adolphs et al., Nature, 2005). I was therefore very puzzled to find that in the task used here fear detection can be achieved by only looking at a suboptimal facial feature: the mouth. In fact, fear categorization could be effectively achieved by only paying attention to how wide the mouth is open (see fig. 1 and suppl. fig. 1) most likely because the mouth is only open for this expression and not the happy expression in the stimuli used. Furthermore, the authors only used 4 face exemplars. These caveats in my view seriously undermine the conclusions made by the authors.

- 'Ambiguity' here is still related to the detection of fear, as the morph goes between happy and fear. Therefore, there is always a bit of fear in the ambiguous stimuli. I would have been more convinced by the claims made by the authors if they had used ambiguous stimuli not related to fear, let's say morphed faces between happy and sad, or another facial expression. My point is that if the authors would like to convincingly demonstrate that the amygdala is processing ambiguity **per se** (as in the cited papers on decision making), this effect should be assessed and

observed with facial expressions other than fear.

- This control of an expression other than fear could also be important because the face localizer used here is rather unconventional in terms of stimuli (i.e., faces expressing fear and anger), task, design and control group chosen. By doing so, the authors selected neural populations sensitive to those expressions and not the entire range of expressions. Moreover, it appears as though this localizer does not locate the commonly described face processing network (see also Mende-Siedlecki et al 2013). It would also be interesting to report how the FFA respond to the experimental manipulation made by the authors.

- The variations in stimuli, which did not include all morph levels, together with the grouping by ambiguity lead to anchor / unambiguous faces occurring *comparatively* less frequently; therefore, the differences observed that are attributed to ambiguity might actually reflect a novelty signal for rare events (for a similar problem with unbalanced design see also Rotshtein et al., 2005; Egner et al., 2010; Summerfield & Koechlin, 2008).

- The stimuli which are averaged across are not in fact similar, this evident from subjects' behavioral judgments in sFig 1d-g (also the y axis, as well as in the text, should refer to levels of facial expression *intensity* judged, NOT 'arousal')

- Eye movements: why were Caucasian faces used as stimuli, when Asian subjects were tested? Or more simply, why testing Asian participants? We know that there are differences between Westerners and Easterners in the way observers decode facial expressions (Jack. et al. 2009). Also: the patients show no differences in eye gaze patterns, while eye avoidance has been reported previously - how do the authors reconcile this? Moreover, note the substantial inter-subject variation in terms of center fixations in patients (see SFig6).

- I think that the 3 amygdala lesion patients should be excluded; the main findings concern the distinction between responses recorded from single cell versus population level. If they remain included however, the most important aspect needs to be concerned: the sampling error that leads to the erroneous impression that these subjects are different compared to controls (see SFig4). In fact one can detect controls that show psychometric functions resembling that of the patients; this is not evident given the choice of the statistical analyses chosen here. Instead, approaches comparing patients' behavior on a single subject level to a group of appropriate, i.e. age- and gender-matched, controls should be selected to ascertain whether manipulations lead to differences that are comparable in controls and patients (see e.g. <http://homepages.abdn.ac.uk/j.crawford/pages/dept/SingleCaseMethodology.htm>).

- More emphasis should be put on recent studies assessing the role of the amygdala under different conditions (e.g. Freeman et al 2014; Ramon et al 2015).

-The authors fitted behavioral data with a psychometric function (i.e. a logistic model), and computed the emotion judgment index (EJI) and emotion sensitivity index (ESI) using the fitted model. However, both EJI (50% fear judgment) and ESI (steepness of the curve) could be inferred directly from the model parameter and xhalf, which also return confidence interval for each individual subjects. I suggest the author to replace EJI and ESI with the model estimation and xhalf. Moreover, they should use a hierarchical logistic regression (logistic mixed-model) to

fit the psychometric function for all subjects with the group as fixed effect and each subject/session as random effect. Doing so will better take into account repeat measurements (sessions), subject-level variance, and more appropriate group level statistics (instead of paired group t-test). It will also return a better estimation of the group level parameters and confidence intervals.

- The model selection part on page 8 nicely demonstrates the linear relationship between fear intensity and the spike rate in the fear selective amygdala neurons. However, the rationale for this analysis is not clearly motivated by a specific hypothesis, and also not applied in the analysis of the ambiguous selective neurons. It would be better to move this session to the supplementary result.

- The authors should also report whether there is any lateralization effect observed in the pooled results of the single neuron recordings, as this point was previously addressed in their fMRI data. It would be interesting to report whether or not there was a difference here too.

Minor:

- The authors should also update supplementary figure 2 to include 95%CI for the individual plot.

- Page 5 first paragraph "ESI significantly steeper (Fig. 1c,e;" here should be Fig. 1d,e

- Figure 2f and the first paragraph on page 6 should be removed, as the interaction is not significant.

- Page 8 first paragraph: "There were two subtypes of responses: 21/33 neurons increased their firing rate as a function of fear intensity (Fig. 3a and Supplementary Fig. 10a,c), whereas 12/33 decreased their firing rate as a function of fear intensity (Fig. 3b, Supplementary Fig. 9a and Supplementary Fig. 10a,c)." The figure being referred to is incorrect. It should be "(Fig. 3a, Supplementary Fig. 9a and Supplementary Fig. 10a,c), ... (Fig. 3b and Supplementary Fig. 10b,d)"

- In suppl. Fig. 1f and g: what are SF3 and SM3. I guess it is 2, but I wanted to be sure.

- In supplementary figure 12 legend "Population analysis with balanced trials in the regression model. Figure legend is the same as Fig. 7. a,b," However there is no Fig 7.

Reviewer #3 (Remarks to the Author):

The present study usefully addresses the question of emotion intensity processing and ambiguity processing in a single paradigm to assess amygdala function via lesion, intracranial recording and fMRI data in human subjects. The use of these three subject types makes this a very powerful study. Below I offer the authors comments that might clarify the relationship between the recordings and the fMRI BOLD signal changes.

1. The direction of the single cell neuronal data matched the direction of the BOLD signal changes in the ambiguity condition (i.e., both BOLD signal and neuronal activity were higher (in most neurons measured) to anchors vs ambiguity), but these directions did not match up in the emotion intensity condition (i.e., while BOLD signal was increased to happy anchors compared to fear anchors, the neuronal data was opposite with most neurons measured showing greater activity to the fearful anchors). I do not see this as problematic, as it is yet unclear what the relationship between neuronal activity and BOLD signal should be. This is an opportunity for the authors to discuss this. They might have a look at David Leopold's work in humans and non-human primates showing that in repetition suppression, which produces initial BOLD signal increases and subsequent BOLD signal decreases in humans, neuronal recordings in cortex show spike increases at both time periods (see also Kim et al 2010 in SCAN for opposite BOLD responses observed in two separate, but similar face processing tasks).

2. Related, the laterality data are also relevant to this point perhaps since the emotion condition neuronal data matched the laterality of the BOLD signal changes better than the ambiguity condition which was bilateral for neuronal data and right sided for ambiguity data.

3. There is a presentation of population recordings that ends on page 11, stating that population recordings differentiated between emotion and ambiguity, but the direction of these effects is not presented. Then on page 13 population recordings are revisited. Is this the same analysis or two different analyses. My overall question is did the direction of the population recordings match the direction of the single cell data? Or is it possible they matched the BOLD signal better than the single cell data? The presentation on page 13 suggests the populations matched the BOLD signal better, which, if this is the case, is very relevant I think to my question 1 above.

Minor Comment:

1. A more accurate reference for the notion that amygdala responses are related to the source ambiguity of fearful faces would be either Whalen et al 2001 in Emotion (if it is brain data you seek) or Whalen, 1998 in Current Directions for the original formulation of the notion.

Reviewer #4 (Remarks to the Author):

The study aims at demonstrating differential representation in the human amygdala for ambiguity and emotion. The authors use two main approaches, imaging in humans and single-cell recordings in patients, and examined behavior in healthy controls, neurological patients, and amygdala-lesion patients. The physiological approaches and the populations complement each other nicely and enable examination of micro and macro properties of the representation in the human amygdala. This is rarely done, and is definitely a strong aspect of this study.

Unfortunately, the behavioral paradigm does not provide a reliable dissociation between the two main factors that are the target of this study: emotion and ambiguity. This is a major confound that casts doubt on the interpretation and therefore dampens my enthusiasm. In addition, the analyses are not convincing and not rigorous enough, and fall short of providing strong conclusions. As a result, the conclusions are not fully supported. The authors have valuable data in their hands, but much more work is required.

Concerns

- Ambiguity was defined operationally, yet a clear definition of ambiguity is missing. Therefore, the rationale of the study and the task design is problematic, as can be seen by the operational examination of ambiguity. Ambiguous faces were created using morphs, which guarantees indeed that the stimulus is ambiguous, but there is no validation of ambiguity in the perceptual emotion judgment.

- Moreover and importantly, it is not clear that emotion and ambiguity are indeed dissociated in this design. For example, the possibility of morphed faces being perceived as neutral was not examined (and confidence rating does not rule out this possibility). Another possibility is that ambiguity is not homogenous or linear along the emotional axis (e.g. it is different next to fear vs next to happy etc.). This can be tested behaviorally in principle, but they do not do so, and it would be much better if the design took care of it to begin with. These are major confounds for all presented results.

- There is no dissociation between the face perception and the decision. While the results reflect both processes, they are attributed and interpreted to the perception. Similarly, there was no clear dissociation between the decision and the confidence judgment.

In this context, fMRI results were mostly significant 6 seconds after stimulus onset, which was after a decision was made and after confidence ranking. Therefore, these results may be correlated with confidence levels. Similarly, single neurons PSTH should be separated for 250-1000ms and 1000-1750ms to separate perception and decision.

It should be examined if results are more/less robust for decision time locked or confidence rating time locked analysis.

- In the manuscript, participants pushed a right button, using right hand, for FEAR and a left one, using left hand, for HAPPY. Laterality of responses to fear/happy faces was not counter balanced, which can confound the results regarding amygdala response laterality in the fMRI experiment.

- The authors suggest that patients with lesions in the lateral amygdala were not impaired in ambiguity judgment. Nonetheless, results that relate ambiguity to neural activity are mainly focused on the lateral amygdala, both in single neurons and fMRI. If these neurons were encoding ambiguity, one should expect a difference between lesioned and healthy patients in ambiguity judgment.

- Both in fMRI and single neurons results, the authors interpret the increased response to both

anchors (with decreased to no response to morphs) as ambiguity coding. Essentially, this is a response to emotional intensity or confidence, and does not imply ambiguity coding. In line with this, it was suggested that subpopulations of neurons in the amygdala encode both reward and punishment error (reference # 50 - this reviewer is not an author on this study).

Specifically, single neurons ambiguity coding was reported for a large fraction of units with increased firing rates in response to both valences and only three neurons increased the response to the morphed faces, i.e. respond to actual ambiguity. In addition, in figure 4h the upper unit out of the three does not seem to be positively correlated with ambiguity. In addition, there was no example of a true two steps gradual response to ambiguity that was also gradual in both happy and fear faces.

Minor concerns

- In the manuscript, there is a comment that the representation of ambiguity is the same in all amygdala subnuclei, but the authors only show 4 neurons in the CE, of them only one responded to fear. Thus, the above cannot be stated.
- Figure 1C. Please add a psychometric curve for each of the four different faces sets.
- Three levels of confidence allow very little variability in confidence levels.
- Was there a difference in behavior as a function of the block number? Namely, was there learning during the experiment that resulted in higher consistency in the last block?
- All lesion conclusions are based on three, possibly over trained, lesioned patients. Statistical analysis, for example t-test, on such small group is problematic.
- Please avoid reporting non-significant statistical test as a result. For instance, page 5, $p=0.12$ for lesioned patient.
- General statements of accepting the null hypothesis since they were not rejected are incorrect and should be avoided. For instance, page 10, first paragraph, chi square test.
- Figure 2. Please show both results for both hemispheres, to show the dissociation between responses in the right and left amygdala.
- Page 6, bottom: the interpretation of figure 2e is incorrect, should be referring to anchor vs. intermediate and anchor vs. most ambiguous.
- Page 8, first paragraph: please correct supplementary figure 10 referencing.
- Page 8, second paragraph: errors of R squared are not interpreted and appear to be larger than the detected effect.
- Figure 3: PSTH results were examined for 250ms bins. Please check results across different bin sizes. In addition, error bars should be added to the PSTH to demonstrate that effects do not result from outliers.
- Please add analysis for decision coding in single neurons (and not the entire population).

Original (black), our reply (blue), copy of text added to manuscript (blue underline)

We thank all four reviewers for their constructive and expert comments. We note that there was disagreement amongst the reviewers on two issues, which we briefly comment on here. First and foremost, several reviewers (#1, #3) found the unique collection of our three different types of dependent measures valuable (single-units, fMRI, lesion behavior), while others (#2) did not because the different measures do not all yield the same conclusions. We agree with both these positive and negative views, but in the end have decided to retain all three types of measures for two reasons: first, their combination is unique and publishing them together permits comparisons that would not be possible, or would be very cumbersome if they end up scattered in separate publications; second, publishing them together is a more accurate and honest representation of our study. Deleting a dataset from a measure that does not fit so well with the others would produce an artificial consensus that, we feel, is not in the best interests of the field.

The second issue that several reviewers commented on is the limited range of stimuli we used in our study—we only tested morphs between two emotions, fear and happy. As Reviewer 1 correctly notes, the reason for this was purely practical: in order to conduct this study, and in particular collect the data from the neurosurgical patients as well as the three patients with amygdala lesions, we simply had to make a decision to focus on morphs between two emotions. Future studies would be required to probe generalizability to the many other possible emotions, to stimuli other than faces, and so forth. We also point out that, in response to these concerns, we did indeed collect additional data on another emotion pair (anger-disgust). For this, we decided to focus on the measure that was generally agreed to be of the most decisive nature: the single-unit recordings (the amygdala patients were not available, and fMRI would have required a full new group study to achieve the requisite statistical power). As we note in response to Reviewer 1 below, these new data fully support our previous conclusions, and while of course not comprehensive, do offer an argument for the generalizability of our findings.

Reply to comments from Reviewer 1

The authors have effectively combined 3 studies: one with fMRI, one with intracranial (single neuron depth) recordings and one with patients with amygdala lesions to address the question of whether the amygdala responds to the strength of a given emotion (fear versus happiness) or the ambiguity of the expression (highest at middle morphs in a fear to happiness continuum). The use of the same stimuli across these three datasets allows for the powerful comparison of findings across methodologies.

The methods and statistics are generally solid and the paper is well written. There are a number of issues highlighted below that I think warrant addressing but overall this represents a considerable endeavour and a valuable addition to the literature.

Summary of key results.

The fMRI data indicate a left amygdala response that increased with level of happiness (faces varying from 100% fear to 100% happy through intermediate morphs) and a right amygdala response to low ambiguity. Patients with amygdala lesions showed an increase in fear sensitivity - i.e. less % fear in the morphed expression was required for participants to reach the point of equivalence (50% of faces categorized as fear; 50% categorized as happy). The intracranial data, meanwhile, suggested that independent populations of neurons responded to (i) increasing levels of fearfulness (ii) increasing levels of happiness (or decreasing levels of fearfulness) and (iii) (low) levels of ambiguity (these showed a preference for 100% happy or fearful faces).

Originality

While prior studies have used either a neuropsychological approach, fMRI, or intracranial recordings to examine the role of the amygdala in processing different emotional expressions, to my knowledge this is the first to combine all three approaches, while using the same task to facilitate comparability of findings across approaches, and which has also addressed the question of whether different amygdala neurons encode whether the expression is closer to fear or happiness versus high or low on ambiguity without having a preference between continuum ends.

We thank the reviewer for noting the value of combining the three different methods, and the positive comments on the methods, statistics, and value to the literature.

Issues. 1) As the authors recognize (top of page 15), the use of just two emotions to create a single continuum of expressions (morphed from 100% fear through intermediate steps to 100% happiness) limits the potential generalizability of the current findings. While worth recognizing as a limitation, this is offset by the value of comparability of the datasets across methods and by the need to balance power against stimulus set size when conducting work with patient groups who tire easily, hence limiting the potential length of data collection.

We thank the reviewer for correctly noting the justifications for our use of a limited range of emotions; indeed our ambition to collect data across all three types of dependent measures necessitated a simple and brief protocol. We made two changes to clarify the generalizability of our findings. First, we added new single-unit data and analyses to show that our results also hold for a second emotional dimensions: the anger-disgust dimension. We have added the following text on p.10:

“We performed additional experiments to show that amygdala neurons not only encoded emotion ambiguity along the fear-happy dimension, but also along the anger-disgust dimension. For this control, we recorded in total 57 neurons (mean firing rate>0.2Hz) from two patients. Using the same selection procedures, we found 9

amygdala neurons (15.8%; binomial $P < 0.0001$) with a significant trial-by-trial correlation with the three levels of ambiguity. This suggests that the amygdala encodes emotional ambiguity in a domain-general fashion. Similar to fear-happy morphs, there were also more neurons (7/9) with maximal firing rate for anchor faces than for ambiguous faces (2/9).”

Second, we clarified our conclusions and title to limit claims to “emotion ambiguity” rather than ambiguity in general. The revised title is “Parametric encoding of emotion ambiguity and intensity in the human amygdala”.

2) I have a number of concerns regarding the 'ambiguity' analyses that I would be keen to see addressed. The findings from these analyses indicate greater amygdala responsivity to the 100% fear or happiness 'anchor' faces than to more ambiguous intermediate face morphs. There is nicely convergent evidence for this from both the fMRI and intracranial data. (i) However, it is potentially confusing to frame this as having shown 'representation of ambiguity' in the amygdala as readers are likely to assume this means an elevated amygdala response to ambiguous stimuli.

We have clarified this issue in the text by referring to ‘parametric encoding’ of ambiguity instead of representation. We agree that ‘representation’ might raise the mistaken interpretation of a proportional relationship instead of the inverse-proportion relationship we found. We have explicitly pointed out this fact in Discussion.

(ii) Further, a number of previous studies have similarly reported that responses in the amygdala as well as other face-sensitive region increase for stimuli towards the end of face continua and are lower for mid-continuum faces (see Said et al., Neuropsychologia 2011 for a discussion and references to this literature). Here, an alternate theoretical proposal put forward to explain these findings is that activity in regions sensitive to faces, such as the amygdala and FFA increase as faces move away from the centre of 'face space'. Said et al. (2011) address the extent to which this is equally true for an expression continuum and for a control non-social continuum - the findings are not clear cut but worth taking a look at. This alternate theoretical interpretation could apply to the results observed here and warrants discussion.

We have added discussion (p.17) of this potential interpretation of the findings:

“The most ambiguous faces are in the middle of a face continuum whereas anchor faces are at the extremes of the continuum. Thus, another possible explanation of the amygdala responses to emotion ambiguity that we observed could be that the amygdala encodes the absolute distance from the average face, consistent with a previous finding that the amygdala has a stronger response to the extremes of the dimensions than to faces near the average face (Said et al., 2010)”

(iii) Also relevant here, is a critique by Aguirre and colleagues who further argue that findings of a stronger response to continuum extremes might simply reflect a methodological confound with adaptation effects being unaccounted for and these being likely to be more pronounced for stimuli in the middle than at the extremes of continua (due to greater average 'closeness' in physical similarity to the stimulus presented previously or afterwards in a randomized series), Kahn & Aguirre, Neuroimage, 2011. The extent to which this critique might be pertinent to the analyses in the current study should also be addressed.

We thank the reviewer for pointing this out. We now acknowledge this possibility. However, a new analysis that we have performed argues that it is unlikely that our results are due to novelty or ambiguity (please refer to our response to Reviewer 2’s 4th major question, below). We have included the following discussion in the revised manuscript (on p.17):

“Furthermore, adaptation studies seek a graded recovery from neural adaptation with ever greater dissimilarity between pairs of stimuli (Kahn and Aquirre, 2012). In the present study, anchor faces are more distinctive compared to ambiguous faces and are thus less subject to adaptation from perceptual neighbors. Therefore, more pronounced adaptation for stimuli in the middle than at the extremes of the face continuum could in principle be one mechanism that explains the amygdala’s response to ambiguity we found in our study. However, we used a sufficient number of distinct stimuli, and their order was completely randomized, making face adaption implausible in our protocol.”

(iv) The authors state that 'in each block each level of the morphed faces was presented 16 or 12 times (4 or 3 repetitions for each intensity) and each anchor face was presented 4 or 12 times (1 or 3 repetitions for each intensity)'. Why were the anchor faces presented less often? This alone might lead to a greater amygdala response to anchor faces (greater novelty, reduced habituation across presentations). Are there blocks where everything is presented 12 times (this is a little unclear)? If so, it would be good to know whether the reported findings hold for these blocks alone.

Yes, there are blocks where each stimulus is presented an equal number of times. Based on new separate analysis of these blocks, we can exclude the possibility that the reported findings are driven by non-equal number of presentation of stimuli. We clarified the methods (on p.27):

“In neurosurgical patients C26, C27, H42, H43, and H44 (9 sessions in total), in each block, each level of the morphed faces was presented 16 times (4 repetitions for each identity) and each anchor face was presented 4 times (1 for each identity). In all other neurosurgical patients (5 sessions in total) and all other subjects, each anchor face and each morphed face was presented the same number of times (12 times, 3 repetitions for each identity).”

Importantly, separate analysis of the data from the patients who had an equal number of repetitions for each stimulus level revealed results very similar to those reported from the subjects with unequal number of trials. A significant population of amygdala neurons encoding emotion ambiguity (14 ambiguity-coding neurons among 102 neurons, 13.7%, binomial $P < 10^{-3}$; 11 neurons had the maximal firing rate for anchors and 3 neurons had the maximal firing rate for the most ambiguous faces; see Reviewer Figure 1 below), and this percentage was not significantly different from the entire population including unbalanced trials (χ^2 -test: $P=0.99$). This shows that our results were not primarily driven by different levels of adaptation due to unequal numbers of repetitions.

Reviewer Figure 1. Similar ambiguity coding in patients who had an equal number of repetitions for each stimulus level. **a-b**, Average normalized firing rate of ambiguity-coding neurons that increased ($n=11$) and decreased ($n=3$) their firing rate for the least ambiguous faces, respectively. **c-d**, Mean normalized firing rate at each morph level (**c**) and ambiguity level (**d**) for 14 units that increased their spike rate for less ambiguous faces. **e-f**, Mean normalized firing rate at each morph level (**e**) and ambiguity level (**f**) for 3 units that increased their spike rate for more ambiguous faces. Normalized firing rate for each unit (left) and mean \pm SEM across units (right) are shown at each level. Asterisks indicate significant difference between conditions using paired two-tailed t-test. *: $P < 0.05$.

We have included the following results in Discussion (on p.17):

“To further exclude the possibility that different numbers of repetitions for each stimulus level might have resulted in a greater response to anchor faces because the anchor faces were shown less often (i.e., leading to reduced habituation across presentations and thus weaker adaptation (Kahn and Aquirre, 2012)), we separately analyzed the recordings performed in sessions where all stimuli (anchors and morphs) were shown exactly the same number of times (see **Methods**). We found very similar proportions of amygdala neurons encoding emotion ambiguity in this group of patients (14 ambiguity-coding neurons among 102 neurons, 13.7%, binomial $P < 10^{-3}$; 11 neurons had the maximal firing rate for anchors and 3 neurons had the maximal firing rate for the most ambiguous faces). This percentage of ambiguity-coding neurons was not significantly different from the

entire population (χ^2 -test: $P=0.99$). This shows that our results did not arise simply from different levels of adaptation due to unequal numbers of repetitions.”

(v) Finally, if it is indeed the low ambiguity of the anchor faces that is critical, then an issue that also requires discussion is how this fits with other accounts suggesting that the amygdala is sensitive to high levels of ambiguity (e.g. Whalen TICS 2007). Indeed the work on ambiguity reviewed in the Introduction points to the amygdala response varying positively with level of ambiguity - i.e. in the opposite direction to the findings reported but this is not made clear in the discussion.

We have clarified this issue in the Discussion (on p.16):

“Some studies have shown that the amygdala responds maximally to ambiguous stimuli or ambiguous choices (Hsu et al., 2005, Herry et al., 2007). In contrast, we found that both the neuronal population response and BOLD signal were maximal for the least ambiguous faces. This is consistent with other studies showing suppression of BOLD-fMRI activity for faces with the most ambiguous trustworthiness (Todorov et al., 2008, Freeman et al., 2014). Further single-neuron studies are needed to determine whether there are neurons firing maximally for the most ambiguous stimuli in tasks as (Hsu et al., 2005, Herry et al., 2007), or whether these results are attributable to the complex relationship between single-neuron activity and the BOLD-fMRI signal. Although we consider it unlikely, it is conceivable that the amygdala has separate populations of neurons, some of which increase and some which decrease their firing rate to ambiguity (e.g., projection and interneurons), and that different studies sampled different sets of these neurons. Here we interpret any change in firing rate, or in BOLD signal, as carrying information, and therefore do not further interpret the sign of that change.”

We have included the following also in the Introduction (on p.3):

“However, other studies have shown suppression of BOLD activity for faces with the most ambiguous trustworthiness (Todorov et al., 2008, Freeman et al., 2014).”

3) Considering each of the experiments (fMRI, intracranial, lesion) separately, I found the intracranial findings especially a pleasure to read with the analyses employed being more sophisticated and the interpretation of the data more nuanced (though I say this as a fMRI researcher so I may not be fully suited to critique all aspects of the methodology). Here, the modelling of the effect of decision (fear/happy) for 50/50 trials is a nice touch with these findings converging well with previous work (e.g. Pessoa et al., Cerebral Cortex, 2006). My only query here is that, from the Table in the Supplements, the majority of low ambiguity responsive neurons and neurons that positively track level of face fearfulness appear to come from a single subject (C26). Might it be possible for the authors do a confirmatory analysis to demonstrate the results hold when sampling equal amounts of data from each patient (or omitting this one patient)? If there are too few emotion or ambiguity responsive neurons in the remaining patients to allow this, this should be addressed as a potential limitation with regards to the generalizability of the findings.

We conducted a control analysis to confirm that our results remain valid after omitting Patient C26. 190 neurons remained after excluding Patient C26. Of these, 22 were emotion-tracking neurons (11.6%, binomial $P=8.80 \times 10^{-5}$; 10 neurons increased firing rate as a function of fear intensity and 12 decreased firing rate as a function of fear intensity; see Reviewer Figure 2 below) and 17 ambiguity coding neurons (8.95%, binomial $P=0.0073$; 14 neurons had the maximal firing rate for anchors and 3 neurons had the maximal firing rate for the most ambiguous faces; see figure below). Thus, results were similar when excluding Patient C26. This confirms that our conclusions were not driven by neurons from this patient alone.

Reviewer Figure 2. Similar emotion and ambiguity when omitting Patient C26. **a**, Histogram of regression R^2 . Neurons that had a higher firing rate for fearful faces are shown on the right of 0 whereas neurons that had a higher firing rate for happy faces are shown on the left. Fear-tracking neurons are in red, happy-tracking neurons are in blue, whereas non-emotion-tracking neurons are in gray. **b**, Slope of linear regression fit. Gray: non-emotion-tracking neurons. Red: fear-tracking neurons. Blue: happy-tracking neurons. **c-d**, Average normalized firing rate of ambiguity-coding neurons that increased ($n=14$) and decreased ($n=3$) their firing rate for the least ambiguous faces, respectively. **e-f**, Mean normalized firing rate at each morph level (**e**) and ambiguity level (**f**) for 14 units that increased their spike rate for less ambiguous faces. **g-h**, Mean normalized firing rate at each morph level (**g**) and ambiguity level (**h**) for 3 units that increased their spike rate for more

ambiguous faces. Normalized firing rate for each unit (left) and mean±SEM across units (right) are shown at each level. Asterisks indicate significant difference between conditions using paired two-tailed t-test. *: P<0.05 and **: P<0.01.

4) *With regards to the lesion study, how do the authors reconcile the finding that amygdala lesion patients were 'more likely to judge faces as fearful' (pg 5) with prior reports that patients with amygdala damage show reduced ability to recognize fearful expressions (Adolphs et al., Nature, 1994)? Given that both the fMRI findings (increased amygdala response as percentage of happiness goes up and percentage of fearfulness goes down) and the lesion study findings (increased sensitivity to fear in patients with amygdala damage) are the opposite to prior findings it makes me worry a little that there might be something specific to the stimuli used here. It is reassuring on this issue that the intracranial study using the same stimuli found that nearly twice the number of neurons positively tracked fearfulness versus happiness but it still needs addressing. (Ideally I would have been keen to also see findings for fearful and happy expressions each morphed with neutral expressions but I appreciate the difficulty of conducting a multi-method study such as this and that such a request at this stage would not be practical).*

Please note that the group of lesion patients tested in this paper did not include patient SM, who is the patient tested in (Adolphs et al., Nature, 1994). The principal difference between our present study and (Adolphs et al., Nature, 1994) is the extent of lesions: our patients have most of the basolateral complex (lateral, basal and accessory basal nuclei) lesioned. In contrast, the central, medial and cortical nuclei of the amygdala were intact. Patient SM in (Adolphs et al., Nature, 1994) has the entire amygdala lesioned, including both basolateral and centromedial amygdala. Due to the different roles of the basolateral and centromedial amygdala, our results were as expected (see below). Notably, our results are consistent with (Terburg et al., Transl Psychiatry 2012), whose patients have similar lesions compared to ours. We now explain the lesion results as follows (on p.17):

“In our three amygdala lesion patients, most of the basolateral complex of the amygdala was lesioned but the centromedial nucleus was intact. The basolateral amygdala is the primary source of visual input to the amygdala and the centromedial amygdala is a primary source of output to subcortical areas relevant for the expression of innate emotional responses and associated physiological responses (LeDoux, 2007). The centromedial amygdala provides most of the amygdala projections to hypothalamic and brainstem nuclei that mediate the behavioral and visceral responses to fear (Aggleton et al., 1980, Davis, 1992, Fudge and Tucker, 2009). Furthermore, the projection neurons in the central nucleus are mostly inhibitory, and are, in turn, inhibited by inhibitory intercalated cells in the lateral and basal amygdala. Disinhibition through this pathway is thought to lead to the expression of emotional responses (LeDoux, 2007). Although direct evidence into the role of amygdala sub-regions in threat and fear processing comes predominantly from rodent lesion research (Phelps and LeDoux, 2005) and optogenetics (Haubensak et al., 2010), our findings are consistent with prior human amygdala lesion results: one patient with complete amygdala damage including both basolateral and centromedial nuclei showed decreased fear judgments (Adolphs et al., 1994), while another group of five patients with only basolateral amygdala damage demonstrated a hyper-vigilance for fearful faces (Terburg et al., 2012, van Honk et al., 2016). These findings are consistent with our present finding of a lowered threshold for reporting fear for morphed faces. Our findings are also consistent with the hypothesis that the basolateral amygdala inhibits fear and anxiety (Tye et al., 2011).”

More fear-tracking neurons (21) than happy-tracking neurons (12) may arise by chance due to our limited sampling (χ^2 -test: P=0.10). But our single-unit data clearly shows that the amygdala encodes both fear and happy emotional expressions. Given that both types of neurons appear to be intermingled in the amygdala, we

could in principle observe a macroscopic signal for either fear-tracking or happy-tracking by fMRI. We have included a discussion of this issue in Question 5 below.

5) *On a related note, there is generally a bit of a disconnect between the Introduction and Discussion. In the Introduction, the authors review literature indicating that many fMRI studies have reported greater amygdala responses to fearful than happy faces, other evidence for an amygdala response to (high) ambiguity, and findings that amygdala lesions are linked to impoverished recognition of fear. In contrast in the discussion the authors argue that their finding that patients with amygdala lesions show increased sensitivity to fear is consistent with prior findings, no longer referencing the well-known studies highlighted in the introduction that link amygdala lesions to impaired recognition of fear. It is fine to bring up at this point that the literature is not as consistent as people might at first think but this complete U turn comes a bit out of the blue. It is important to interpret the current data in the light of a balanced presentation of the existing findings in the field. If the authors are making the case, as it seems, that the precise location of the amygdala lesions explains why the current findings differ from those studies reviewed in the Introduction that reported reduced fear recognition in patients with amygdala damage then it is important that they make this explicit and that as part of this they contrast the location of the lesions in the current patients with those in the studies showing deficits in fear recognition. It should also be clarified that this was a posthoc analysis contrary to the initial hypothesis (as the Introduction suggests a fear deficit was expected).*

Similar comments apply for other findings. A role of the amygdala in processing 'emotional intensity' was predicted, and it seems from the Introduction that the prediction was that amygdala activity would primarily increase with expression fearfulness, but that there might also be a smaller population of neurons more responsive to expression happiness (as indeed suggested by the intracranial data). If this is the case, it needs to be more clearly stated in the early part of the discussion that the fMRI results did not support this prediction showing instead an increase in amygdala responsivity with level of happiness in the expression morph. Stating that the amygdala responded to emotional intensity but not detailing the directionality of the result, and whether or not it fitted with predictions, is too vague. Further, the statement in the abstract that 'our results indicate that the human amygdala processes ... the degree of fear shown in facial expressions' is arguably misleading if by that the authors include the fMRI finding of increased amygdala response to faces as levels of happiness go up and fearfulness goes down.

We have strengthened our discussion. Specifically, we have included the discussion of amygdala lesions as in Question 4, and we have included the following discussion (on p.16):

“Although most studies find activation within the amygdala that is highest for fearful faces (Morris et al., 1996, Phillips et al., 1998, Whalen et al., 2004), there are also studies showing that the amygdala responds to neutral or happy faces (Mende-Siedlecki et al., 2013) as well as to some extent all facial expressions (Fitzgerald et al., 2006). Such general coding of facial expressions is also evident at the single-neuron level (Fried et al., 1997, Viskontas et al., 2009, Rutishauser et al., 2011, Quian Quiroga et al., 2014). Notably, even using the same faces, amygdala BOLD response increases for fearful faces vs. happy faces when using a face mask whereas it decreases for fearful faces vs. happy faces when using a pattern mask (Kim et al., 2010). Therefore, the sign of the amygdala’s BOLD response to facial emotions may largely depend on the task. In the present study, we found both fear-tracking and happy-tracking neurons. There were overall more fear-tracking neurons than happy-tracking neurons (21 vs. 12; although not significant, χ^2 -test: P=0.10), whereas we found a greater BOLD signal for happy faces in the left amygdala. However, these findings are actually not discrepant, since 11 out of 12 neurons showing increasing firing rate with happy intensity were in the left amygdala.”

We also noted the discrepancy between methods as a potential limitation (on p.14):

“We used a unique combination of approaches to address the debate of whether the human amygdala encodes emotion intensity or ambiguity. Different methods measure different signals, and have therefore often pointed to somewhat different conclusions, likely accounting in good part for discrepant conclusions in the literature. Although we used identical stimuli and task, our three different methods still produced somewhat different conclusions. Since our single-unit data clearly shows that the amygdala encodes both emotion intensity and ambiguity, more macroscopic methods (fMRI, lesion) can provide evidence for either emotion intensity or ambiguity coding—given that both types of neurons are intermingled, there is in principle signal to produce either interpretation.”

We have changed the statement in the abstract to be:

“Together, our results indicate that the human amygdala processes both the intensity of emotion in facial expressions as well as the ambiguity of the emotion shown, and that these two aspects of amygdala processing can be most clearly distinguished at the level of single neurons.”

Small points

1) For the measure of 'the slope of the psychometric curve (emotion sensitivity index, ESI)' why did the authors not derive this directly from the logistic function?

We thank the reviewer for pointing this out. We have changed to use the parameters from the logistic functions. We have further used a logistic mixed model to deal with repeated measures (see Reviewer 2's third last major question). Qualitatively the same results were derived.

*2) The statement in the abstract that 'our results indicate that the human amygdala processes both the degree of fear shown in facial expressions and their ambiguity and this effect can *only* be distinguished at the level of single neurons' is too strong. To support this *only* statement, the authors would at least need to conduct more sophisticated fMRI analyses (e.g. using representational similarity (MVPA) or adaptation techniques with reduced or no spatial smoothing of the data) and show that it is still impossible to distinguish responses to morph expression versus responses to morph ambiguity.*

We agree with the reviewer, and have changed our statement to **can be most clearly distinguished at the level of single neurons**.

3) The language based explanation of the lateralization of the amygdala response seems to come a bit out of nowhere and is also is oddly placed just following an argument that similar findings are also observed in animals. If the authors want to leave it in they need to add a sentence or two on why linguistic processing would lead to lateralized processing (given the wide audience this journal attracts).

We have removed the argument regarding language.

4) 'We thus here show, for the first time, a reward-and reward-value independent representation of ambiguity in the amygdala.' This might also be a bit strong, Paul Whalen's group have produced some nice findings on ambiguity that do not involve reward-relatedness.

We removed **for the first time**.

Reply to comments from Reviewer 2

The authors investigated whether the amygdala is functionally related to the processing of fearful faces, to ambiguity or both. To address this question, the authors used the same behavioral task, a facial expression categorization task of 4 parametrically morphed faces (happy to fear), in three groups of observers: amygdala lesion patients (behavioral), healthy adults (fMRI), and single-neuron recordings in neurosurgical patients. The results from the lesion and healthy subjects indicate that the amygdala is involved in both fear (lesion and healthy observers) and ambiguity (healthy observers) decoding. The novelty of this study relies on the single neuron recordings, that show in fact that those responses might rely on the presence of two distinct populations within this region: one coding for fear and the other coding for ambiguity. A result that could not be observed with behavioral and fMRI measures alone.

In general, I found this paper very difficult to follow. There are numerous analyses and supplementary experiments, and some of them might not be necessary (for instance, the eye movement study carried out on an additional group of Chinese observers). I would suggest the authors to refocus on their question of interest and include the measures/ experiments that are really necessary to address the question at hand. As it stands, I would personally remove for instance the results based on the lesion patients (see below), the eye movement results (see also below), and keep the most appropriate comparison between the fMRI data in healthy controls with the single neuron recordings.

We thank the reviewer for the constructive comments. As suggested, we have removed the eye tracking work as well as 5 supplementary figures, which greatly streamlines the paper. We have chosen to retain the basic behavioral results from the lesion and control groups to contrast with the fMRI and electrophysiology groups, since we feel that the inclusion of the lesion data offers a unique array of different approaches to the same overall question (and since we in fact planned the study this way, so that removing a subset of the data would bias our findings).

Having said that, I also found a series of methodological and theoretical concerns that seriously dampen my enthusiasm for this part of the work:

We have conducted additional experiments and analysis, which together conclusively address the points raised (see below). Together, we believe that we are able to address all methodological and theoretical concerns raised.

- The same group has previously shown that attention towards the eye region is critical to effectively decode fear and that the amygdala plays a critical role in this process (i.e., Adolphs et al., Nature, 2005). I was therefore very puzzled to find that in the task used here fear detection can be achieved by only looking at a suboptimal facial feature: the mouth. In fact, fear categorization could be effectively achieved by only paying attention to how wide the mouth is open (see fig. 1 and suppl. fig. 1) most likely because the mouth is only open for this expression expression and not the happy expression in the stimuli used. Furthermore, the authors only used 4 face exemplars. These caveats in my view seriously undermine the conclusions made by the authors.

We apologize for any confusion in the results we presented. In the present study, we did not show evidence that fear detection can be achieved by only looking at the mouth (as stated by the reviewer). Indeed, a previous study by our group found the opposite: the eyes, but not the mouth, predominantly drive neuronal responses that differentiate fear from happy faces (Wang et al., PNAS, 2014). Thus, we certainly share the same expectation as the reviewer and our data do not argue against this. To investigate how specific features of faces like the eyes or mouth would be used in the present study, we would have had to use a different approach, as we did in our prior studies, like Wang et al., PNAS 2014; that study used “bubbles” to construct sparsely revealed faces, but this is

not the approach we used, or the question we asked, in the present study. We removed the eye tracking results (as suggested), since indeed they are uninteresting and merely confirm that all participants looked at the face stimuli. It is correct that we used 4 face exemplars, which was necessitated in order to have sufficient statistical power for our analyses; examining effects of face identity was not within the scope of our study. Again, to reiterate: nowhere in the paper do we show or claim that fear categorization could be achieved by only paying attention to how wide the mouth is open. Nor was it an aim of this study to investigate this. We apologize if this was unclear in the prior version.

*- 'Ambiguity' here is still related to the detection of fear, as the morph goes between happy and fear. Therefore, there is always a bit of fear in the ambiguous stimuli. I would have been more convinced by the claims made by the authors if they had used ambiguous stimuli not related to fear, let's say morphed faces between happy and sad, or another facial expression. My point is that if the authors would like to convincingly demonstrate that the amygdala is processing ambiguity *per se* (as in the cited papers on decision making), this effect should be assessed and observed with facial expressions other than fear.*

We have clarified that we are only making claims about “emotional ambiguity”, and not in ambiguity in general as referred to in decision making. We changed the title accordingly to “Parametric encoding of emotion ambiguity and intensity in the human amygdala”. We have also updated Fig. 1b to make the distinction between emotion intensity and emotion ambiguity clearer.

Also, we have conducted additional experiments and analysis using ambiguous faces along a different axis of emotions (anger-disgust). This dimension does not have a fear component, but we nevertheless found very similar ambiguity coding in the amygdala. Please refer to our response to Reviewer 1’s Question 1.

Finally, we show (for some of our results) that there is a parametric relationship between the degree of ambiguity and the magnitude of the response (e.g., BOLD-fMRI and spikes of single units encoding ambiguity). This finding does separate the intensity of fear from the degree of ambiguity, since these two regressors are not the same. Although the reviewer is correct that even ambiguous faces show some degree of fear, that is not a confound for our conclusions, provided that ambiguity and intensity of fear can vary independently of one another (which they do).

- This control of an expression other than fear could also be important because the face localizer used here is rather unconventional in terms of stimuli (i.e., faces expressing fear and anger), task, design and control group chosen. By doing so, the authors selected neural populations sensitive to those expressions and not the entire range of expressions. Moreover, it appears as though this localizer does not locate the commonly described face processing network (see also Mende-Siedlecki et al 2013). It would also be interesting to report how the FFA respond to the experimental manipulation made by the authors.

We believe this is a misunderstanding and have clarified the text accordingly. Indeed, the localizer we used is well-established (Hariri et al., Science, 2002) and reliably activates the amygdala as well as the commonly described face processing network consisting of regions including FFA, dlPFC and STS (Mende-Siedlecki et al 2013). We have clarified this in the methods (see below). Furthermore, also note that our results do not depend on the localizer because they remain qualitatively the same when using an anatomical ROI of the entire amygdala (**Supplementary Fig. 4d,e**) (on p.33):

“Other cortical regions known to be involved in face processing, such as the fusiform face area (FFA), visual cortex, inferior frontal gyrus (IFG), superior temporal sulcus/gyrus (STS/STG), dorsal medial prefrontal cortex (DMPFC), dorsal lateral prefrontal cortex (DLPFC) were also more activated for faces compared to objects, and

objects minus faces activated the ventral anterior cingulate cortex (ACC) and posterior cingulate cortex (PCC) (Supplementary Fig. 4a), consistent with the face processing network shown in previous studies (Mende-Siedlecki et al., 2013)."

As suggested, we performed a new analysis to determine how the FFA responds to emotion intensity and ambiguity, and included the following results in the Supplementary Results (on p.11):

"Lastly, the fusiform face area (FFA) also tracked the emotion intensity and ambiguity (Supplementary Results)."

"We first identified a functional ROI within the FFA sensitive to faces using the face localizer task (left FFA: peak: x=-36, y=-54, z=-18, 305 voxels; right FFA: peak: x=48, y=-48, z=-21, 342 voxels). Within the FFA ROIs, we found a significant increase of activity in the left FFA with increasing level of fear (peak: x=-24, y=-63, z=-15, Z=3.76, 11 voxels, FWE P<0.05, SVC) whereas we found a significant increase of activity in the right FFA with increasing level of happiness (peak: x=21, y=-60, z=-12, Z=3.73, 11 voxels, FWE P<0.05, SVC). We also found a significant increase of activity in the right FFA with decreasing level of ambiguity (peak: x=30, y=-51, z=-12, Z=3.63, 6 voxels, FWE P<0.05, SVC), however, we did not find any significant increase of FFA activity with increasing level of ambiguity."

*- The variations in stimuli, which did not include all morph levels, together with the grouping by ambiguity lead to anchor / unambiguous faces occurring *comparatively* less frequently; therefore, the differences observed that are attributed to ambiguity might actually reflect a novelty signal for rare events (for a similar problem with unbalanced design see also Rotshtein et al., 2005; Egner et al., 2010; Summerfield & Koechlin, 2008).*

The reviewer raises an important question here. We have now addressed this confound by separately analyzing a group of patients where all stimuli were shown the same number of times. Please refer to our response to Reviewer 1's Question 2-iv, above, where the same issue was raised. Our new analyses confirm that a novelty signal due to the different numbers of stimuli shown cannot explain our results.

*- The stimuli which are averaged across are not in fact similar, this evident from subjects' behavioral judgments in sFig 1d-g (also the y axis, as well as in the text, should refer to levels of facial expression *intensity* judged, NOT 'arousal')*

We performed additional control analysis to confirm that our results cannot be explained by differences attributable to different facial identities. This work shows that while some neurons encode facial identities (as expected from prior studies), this group of neurons is separate from that encoding ambiguity. We included the following new results in the main manuscript (on p.10):

"Furthermore, some amygdala neurons respond to the specific identity of faces (Mormann et al., 2015) and our four facial identities did not have exactly the same valence and intensity judgments (Supplementary Fig. 1d-g). However, a separate control analysis for each facial identity showed that encoding of emotion intensity and ambiguity was not driven by differences in facial identity (Supplementary Results)."

And we now present these new analyses in detail in the Supplementary Results (on p.14) as follows:

"Emotion intensity and ambiguity coding by each facial identity

Subsets of amygdala neurons have been reported in prior studies to be sensitive to facial identities (Mormann et al., 2015), an effect which might interact with emotion and ambiguity coding if present also in our data. To explore this possibility, we selected for neurons selective for facial identity (n=24 neurons, 10.3%, binomial

$P=3.25 \times 10^{-4}$). However, none of these neurons were emotion-tracking neurons and only 8/24 neurons were ambiguity-coding neurons. This selection used all trials; when only using anchor trials, we found that there were 10 neurons (4.27%, binomial $P=0.63$) selective for facial identity and similarly, none of these neurons were emotion-tracking neurons and only 2 neurons were ambiguity-coding neurons. Second, we separately analyzed each facial identity. With Face Model F1 only (**Supplementary Fig. 1a**), we found 22 emotion-tracking neurons (9.40%, binomial $P=0.0017$; 16 neurons increased firing rate as a function of fear intensity and 6 neurons decreased firing rate as a function of fear intensity) and 15 ambiguity coding neurons (6.41%, binomial $P=0.13$; 13 neurons had the maximal firing rate for anchors and 2 neurons had the maximal firing rate for the most ambiguous faces). With Face Model F2 only, we found 21 emotion-tracking neurons (8.97%, binomial $P=0.0036$; 11 neurons increased firing rate as a function of fear intensity and 10 neurons decreased firing rate as a function of fear intensity) and 24 ambiguity coding neurons (10.3%, binomial $P=3.25 \times 10^{-4}$; 20 neurons had the maximal firing rate for anchors and 4 neurons had the maximal firing rate for the most ambiguous faces). With Face Model M1 only, we found 31 emotion-tracking neurons (13.3%, binomial $P=2.90 \times 10^{-7}$; 16 neurons increased firing rate as a function of fear intensity and 15 neurons decreased firing rate as a function of fear intensity) and 20 ambiguity coding neurons (8.55%, binomial $P=0.0074$; 15 neurons had the maximal firing rate for anchors and 5 neurons had the maximal firing rate for the most ambiguous faces). With Face Model M2 only, we found 23 emotion-tracking neurons (9.83%, binomial $P=7.57 \times 10^{-4}$; 9 neurons increased firing rate as a function of fear intensity and 14 neurons decreased firing rate as a function of fear intensity) and 13 ambiguity coding neurons (5.56%, binomial $P=0.28$; 10 neurons had the maximal firing rate for anchors and 3 neurons had the maximal firing rate for the most ambiguous faces). Our results thus show that although different facial identities had different strengths for selections, emotion-tracking and ambiguity-coding neurons could be selected independently for each identity, and thus encoding of emotion and ambiguity was not driven by facial identity. Note that fewer neurons were selected due to fewer trials and thus reduced statistical power.”

We have also changed the figure label correspondingly.

- *Eye movements: why were Caucasian faces used as stimuli, when Asian subjects were tested? Or more simply, why testing Asian participants? We know that there are differences between Westerners and Easterners in the way observers decode facial expressions (Jack. et al. 2009). Also: the patients show no differences in eye gaze patterns, while eye avoidance has been reported previously - how do the authors reconcile this? Moreover, note the substantial inter-subject variation in terms of center fixations in patients (see SFig6).*

We agree with the reviewer that the eye tracking results are unclear, and we have removed the eye movement results as suggested by the reviewer before.

It is worth noting that the face stimuli used in this study were Caucasian faces (**Supplementary Fig. 1**) and our subjects were from a variety of ethnicity. However, there was considerable consistency between neurosurgical patients and controls in judging faces, showing that consistent judgment on this face dataset was independent of ethnicity or cultural difference. As can be seen from our own data, all subjects could judge the anchor faces with clear emotions very well, and we could fit a psychometric curve for each individual subject, meaning that all subjects could well track the graded change of facial emotions. On the other hand, amygdala lesion patients AM and BG are Caucasian and AP lives in the United States. Thus, the altered facial emotion judgment compared to neurosurgical patients could not be attributed to racial or cultural difference.

It is true that some patients with complete amygdala lesions (different from the ones tested here) do not fixate onto the eyes in faces, when shown under quite different conditions than in the present study. However, there is to our knowledge no active avoidance of the eyes. There are several reasons why one would not expect to see any differences in eye fixations in our task in any case. First, our face stimuli were small so subjects did not

need to move eyes to judge emotions. Second, faces were presented briefly and preceded by a central fixation cross, with the consequence that most fixations are in the center of the faces. The aim of this protocol was to minimize any eye movement difference between the amygdala patients and controls, as indeed our eye tracking data verify: we would not expect any fixation differences for the reasons above; and we did not observe any.

- I think that the 3 amygdala lesion patients should be excluded; the main findings concern the distinction between responses recorded from single cell versus population level. If they remain included however, the most important aspect needs to be concerned: the sampling error that leads to the erroneous impression that these subjects are different compared to controls (see SFig4). In fact one can detect controls that show psychometric functions resembling that of the patients; this is not evident given the choice of the statistical analyses chosen here. Instead, approaches comparing patients' behavior on a single subject level to a group of appropriate, i.e. age- and gender-matched, controls should be selected to ascertain whether manipulations lead to differences that are comparable in controls and patients (see e.g. <http://homepages.abdn.ac.uk/j.crawford/pages/dept/SingleCaseMethodology.htm>).

We respectfully wish to include the three amygdala lesion patients. The main findings concern the valuable data from all three different dependent measures (lesions, single-unit recordings, fMRI), and we would feel that deleting any one of these datasets would produce a biased picture of the actual full data we collected.

We appreciate the reviewer's concern about the best statistical test to do here, and have replicated our original findings using a multiple case-study approach, in which each patient was compared to the control group using a permutation test. Please refer to our response in the question below.

- More emphasis should be put on recent studies assessing the role of the amygdala under different conditions (e.g. Freeman et al 2014; Ramon et al 2015).

We have included more discussions about these two recent studies (on p.15):

“In particular, using faces with varying levels of trustworthiness, it has been shown that regions in the amygdala track both how untrustworthy a face appeared (i.e., negative-linear responses) and the overall strength of a face's trustworthiness signal (i.e., nonlinear responses), despite faces not being subjectively perceived (Freeman et al., 2014). Using a gradient of unfamiliar or personally familiar individuals with slowly increasing visual information, ventral occipitotemporal face-preferential regions increase with visual information regardless of face familiarity, whereas the amygdala responds abruptly when sufficient information for familiar face recognition is accumulated (Ramon et al., 2015).”

-The authors fitted behavioral data with a psychometric function (i.e. a logistic model), and computed the emotion judgment index (EJI) and emotion sensitivity index (ESI) using the fitted model. However, both EJI (50% fear judgment) and ESI (steepness of the curve) could be inferred directly from the model parameter and x_{half} , which also return confidence interval for each individual subjects. I suggest the author to replace EJI and ESI with the model estimation and x_{half} . Moreover, they should use a hierarchical logistic regression (logistic mixed-model) to fit the psychometric function for all subjects with the group as fixed effect and each subject/session as random effect. Doing so will better take into account repeat measurements (sessions), subject-level variance, and more appropriate group level statistics (instead of paired group t-test). It will also return a better estimation of the group level parameters and confidence intervals.

We thank the reviewer for this suggestion. As a result, we have changed our approach and adopted the logistic function as suggested. Our EJI is essentially the same as x_{half} (inflection point), and ESI is very similar to α (the

steepness of the curve). We have updated all statistics to use x_{half} and α , and all our conclusions are the same (on p.4):

“We quantified each psychometric curve using two metrics derived from the logistic function: i) x_{half} —the midpoint of the curve (in units of %fearful) at which subjects were equally likely to judge a face as fearful or happy, and ii) α —the steepness of the psychometric curve. Based on these two metrics, the behavior of the neurosurgical patients was indistinguishable from the control subjects (Fig. 1d,e; x_{half} : unpaired two-tailed t-test: $t(27)=1.10$, $P=0.28$; α : $t(27)=1.98$, $P=0.058$, permutation $P=0.30$). In contrast, the amygdala lesion patients ($x_{half}=44.2\pm 1.88\%$) were more likely to judge faces as fearful, with x_{half} significantly lower than neurosurgical patients (Fig. 1d; $x_{half}=53.2\pm 4.97\%$; $t(15)=3.00$, $P=0.0089$, effect size in Hedges’ g : $g=1.81$, permutation $P<0.001$) and controls ($x_{half}=51.1\pm 5.16\%$; $t(16)=2.23$, $P=0.040$, $g=1.34$, permutation $P=0.058$) and α significantly steeper (Fig. 1e; $t(15)=3.85$, $P=0.0016$, $g=2.33$, permutation $P=0.002$). Our results were further confirmed using a logistic mixed model (Supplementary Results).”

We have further used a logistic mixed model to deal with repeated measures. We have included the following results (on Supplementary p.10):

“Logistic mixed model

We used a logistic mixed model to fit behavioral judgments (fear or happy choice) for all subjects with subject group and fear level as the fixed effects and each subject as the random effect. Statistical significance of the model was computed by likelihood ratio tests of the full model with the fixed effect of subject group or fear level against a null model without the fixed effect of subject group or fear level. We found that in the full model, fear level could predict behavioral judgments with a significant regression coefficient (9.84 ± 0.20 (mean \pm SEM), 95% CI: [9.44 10.2], $t(333)=49.0$, $P<0.001$), and the full model with the fixed effect of fear level significantly outperformed the null model ($\chi^2(4)=6100.1$, $P<0.001$), suggesting that fear levels could well predict behavioral judgments. Furthermore, in the full model, subject group could not predict behavioral judgments with a significant regression coefficient (0.075 ± 0.064 , 95% CI: [-0.051 0.20], $t(333)=1.18$, $P=0.24$), and the full model with the fixed effect of subject group did not significantly outperform the null model ($\chi^2(4)=1.36$, $P=0.24$), suggesting that behavioral judgments were similar among subject groups.”

- The model selection part on page 8 nicely demonstrates the linear relationship between fear intensity and the spike rate in the fear selective amygdala neurons. However, the rationale for this analysis is not clearly motivated by a specific hypothesis, and also not applied in the analysis of the ambiguous selective neurons. It would be better to move this session to the supplementary result.

We have moved this part to the Supplementary Results and summarized results in the main text (on p.8):

“The linear model we used revealed that a substantial proportion of variance was explained by a continuous response (see Fig. 3c for R^2), with an average absolute slope of 0.80 ± 0.75 Hz / 100%fear (Fig. 3d; 0.71 ± 0.72 for positive slope and -0.95 ± 0.80 for negative slope). We further compared our linear model to more complex models and demonstrated that a linear relationship fitted our data better and our data were not best described by a step-like threshold model (Supplementary Results).

Overall, these findings argue that human amygdala neurons parametrically encode gradual changes of facial emotions, a significantly more fine-grained representation relative to the binary discrimination between fearful and happy facial expressions we and others have previously reported in studies that did not explicitly test for a more continuous representation (Morris et al., 1996, Wang et al., 2014).”

The selection of ambiguity-coding neurons did not involve a model, so we did not further compare against other models. We have included the following results in Supplementary Results (on p.12) and clarified the motivation for model selection/comparison:

“Model comparisons

Did emotion-tracking neurons differentiate continuously between levels of fear/happy in a face, or might they respond only once a certain threshold level of “fear” was present in a face? To confirm that our linear model was a better fit of our data and that our data could be better described by a linear relationship rather than a step-like thresholded model, we assessed how good this linear model was compared to two more complex models using the Akaike Information Criterion (AIC; see **Methods**): a logistic function and a step function. The better a model explains the data (with penalty on complexity), the smaller its AIC value. Compared to the logistic function model, 32/33 emotion-selective neurons had smaller AIC values for the linear model ($\Delta\text{AIC}=13.2\pm 11.4$ (mean \pm SD); t-test against 0: $P=1.69\times 10^{-7}$). Similarly, compared to a step function model, 30/33 emotion-selective neurons had smaller AIC values for the linear model ($\Delta\text{AIC}=5.47\pm 4.29$; $P=2.51\times 10^{-8}$). As a control, we also performed the same comparison for a group of neurons selected using a “step function” model, comparing fearful vs. happy trials with anchor faces only. While many fewer neurons were significant (13/234), we again found that most were fit best by the linear model, even though they were selected using a binary contrast: Compared to the logistic function model, linear fitting was better for all neurons (13/13, $\Delta\text{AIC}=13.6\pm 8.36$; $P=7.41\times 10^{-6}$), and compared to the step function model, linear fitting was better for 11/13 neurons ($\Delta\text{AIC}=3.56\pm 2.74$; $P=5.34\times 10^{-4}$).”

- *The authors should also report whether there is any lateralization effect observed in the pooled results of the single neuron recordings, as this point was previously addressed in their fMRI data. It would be interesting to report whether or not there was a difference here too.*

Unfortunately, it is very difficult to make laterality comparisons in single-neuron recordings and we have thus refrained from doing so for the reasons outlined in the manuscript. The principal reason is that patients with epilepsy have higher rates of abnormal lateralization of function, with more frequent bilateral representations compared to controls. Furthermore, it is difficult to compare left and right in single unit recordings because the recording sites are never in the exact same places. So any kind of uncontrolled sampling bias can produce apparently lateralized effects (this is not a problem for fMRI, which takes the amygdala ROI from an independent localizer to begin with, but there is no equivalent localizer for single-neuron recordings). We thus refrain from drawing conclusions related to laterality based on the single-neuron data.

Furthermore, population regression analysis confirmed that amygdala neurons as a population from both sides encoded both emotion intensity and ambiguity without selection of significant neurons (right amygdala has a weaker strength though), consistent with the parametric modulation of fMRI BOLD responses (see Reviewer Figure 3 below).

Reviewer Figure 3. Population regression analysis for left (a,b) and right (c,d) amygdala. Summary of the effect size across all runs. Left amygdala neurons as a population encoded both emotion intensity (a, $P < 0.001$) and ambiguity (b, $P < 0.001$). Right amygdala neurons as a population also encoded both emotion intensity (c, $P = 0.064$) and ambiguity (d, $P < 0.001$). Effect size was computed in a 1.5-second window starting 250 ms after stimulus onset (single fixed window, not a moving window) and was averaged across all neurons for each run. Gray and red vertical lines indicate the chance mean effect size and the observed effect size, respectively. The observed effect size was well above 0 whereas the mean effect size in the permutation test was near 0.

We have included these results and figure panels as follows (on p.11):

“Notably, population regression analysis confirmed that amygdala neurons as a population from both sides encoded both emotion intensity and ambiguity (right amygdala has a weaker strength though) (Supplementary Fig. 8k-n).”

Lastly, we would like to note that the relationship between the BOLD-fMRI signal and neuronal population activity in the amygdala remains largely unknown. Thus, discrepancies between BOLD-fMRI and single-neuron findings with respect to lateralization remain to be explored in future work. We now raise this point in the discussion. Please also see our response to Reviewer 3’s Question 1 and 2.

Minor:

- The authors should also update supplementary figure 2 to include 95%CI for the individual plot.

We have removed this figure to make the paper more concise.

- Page 5 first paragraph "ESI significantly steeper (Fig. 1c,e;" here should be Fig. 1d,e

We have corrected it.

- Figure 2f and the first paragraph on page 6 should be removed, as the interaction is not significant.

We thank the reviewer for pointing this out. We would like to keep this result because (1) we have related results in the last section of results when we compared between approaches, and (2) Reviewer 4 specifically asked for this result. But we explicitly state that it did not achieve statistical significance.

- Page 8 first paragraph: "There were two subtypes of responses: 21/33 neurons increased their firing rate as a function of fear intensity (Fig. 3a and Supplementary Fig. 10a,c), whereas 12/33 decreased their firing rate as a function of fear intensity (Fig. 3b, Supplementary Fig. 9a and Supplementary Fig. 10a,c)." The figure being referred to is incorrect. It should be "(Fig. 3a, Supplementary Fig. 9a and Supplementary Fig. 10a,c), ... (Fig. 3b and Supplementary Fig. 10b,d)"

We thank the reviewer for pointing it out and we have corrected it.

- In suppl. Fig. 1f and g: what are SF3 and SM3. I guess it is 2, but I wanted to be sure.

These are the identity names from the original STOIC face database. We have clarified the labels.

- In supplementary figure 12 legend "Population analysis with balanced trials in the regression model. Figure legend is the same as Fig. 7. a,b," However there is no Fig 7.

We thank the reviewer for pointing it out and we have corrected it.

Reply to comments from Reviewer 3

The present study usefully addresses the question of emotion intensity processing and ambiguity processing in a single paradigm to assess amygdala function via lesion, intracranial recording and fMRI data in human subjects. The use of these three subject types makes this a very powerful study. Below I offer the authors comments that might clarify the relationship between the recordings and the fMRI BOLD signal changes.

1. The direction of the single cell neuronal data matched the direction of the BOLD signal changes in the ambiguity condition (i.e., both BOLD signal and neuronal activity were higher (in most neurons measured) to anchors vs ambiguity), but these directions did not match up in the emotion intensity condition (i.e., while BOLD signal was increased to happy anchors compared to fear anchors, the neuronal data was opposite with most neurons measured showing greater activity to the fearful anchors). I do not see this as problematic, as it is yet unclear what the relationship between neuronal activity and BOLD signal should be. This is an opportunity for the authors to discuss this. They might have a look at David Leopold's work in humans and non-human primates showing that in repetition suppression, which produces initial BOLD signal increases and subsequent BOLD signal decreases in humans, neuronal recordings in cortex show spike increases at both time periods (see also Kim et al 2010 in SCAN for opposite BOLD responses observed in two separate, but similar face processing tasks).

We share the sentiment regarding the discrepancy between the BOLD signal and single-neuron recordings, which not necessarily measure the same neural signal. We have strengthened this point by adding the following to Discussion (on p.16):

“Although most studies find activation within the amygdala that is highest for fearful faces (Morris et al., 1996, Phillips et al., 1998, Whalen et al., 2004), there are also studies showing that the amygdala responds to neutral or happy faces (Mende-Siedlecki et al., 2013) as well as to some extent all facial expressions (Fitzgerald et al., 2006). Such general coding of facial expressions is also evident at the single-neuron level (Fried et al., 1997, Viskontas et al., 2009, Rutishauser et al., 2011, Quian Quiroga et al., 2014). Notably, even using the same faces, amygdala BOLD response increases for fearful faces vs. happy faces when using a face mask whereas it decreases for fearful faces vs. happy faces when using a pattern mask (Kim et al., 2010). Therefore, the sign of the amygdala’s BOLD response to facial emotions may largely depend on the task. In the present study, we found both fear-tracking and happy-tracking neurons. There were overall more fear-tracking neurons than happy-tracking neurons (21 vs. 12; although not significant, χ^2 -test: P=0.10), whereas we found a greater BOLD signal for happy faces in the left amygdala. However, these findings are actually not discrepant, since 11 out of 12 neurons showing increasing firing rate with happy intensity were in the left amygdala.”

And on p.14:

“It is important to note that the relationship between the BOLD-fMRI signal and neuronal population activity in general remains unclear. For example, there is a marked dissociation during perceptual suppression in non-human primates (Maier et al., 2008), and we found that the neuronal population activity matched the direction of BOLD signals for emotion ambiguity but not intensity. An estimation of the exact proportion of each type of emotion-tracking neurons and the coupling between the BOLD and electrophysiological signals was limited by the number of neurons that we could sample; future studies with substantially more recording of neurons will be necessary to answer these questions.”

2. Related, the laterality data are also relevant to this point perhaps since the emotion condition neuronal data matched the laterality of the BOLD signal changes better than the ambiguity condition which was bilateral for neuronal data and right sided for ambiguity data.

We agree with the reviewer and we have further analyzed neuronal population response for laterality. Please refer to our response to Reviewer 2's last major question. We would like to note that it is very difficult to make laterality comparisons in single-neuron recordings. We have also included the following discussion (on p.20):

“It is also worth noting that the laterality of neuronal response matched BOLD-fMRI in some aspects but not the others in the present study. High-resolution fMRI, precise localization of the electrodes, and more neurons recorded will be necessary in future studies to further address this question.”

3. There is a presentation of population recordings that ends on page 11, stating that population recordings differentiated between emotion and ambiguity, but the direction of these effects is not presented. Then on page 13 population recordings are revisited. Is this the same analysis or two different analyses. My overall question is did the direction of the population recordings match the direction of the single cell data? Or is it possible they matched the BOLD signal better than the single cell data? The presentation on page 13 suggests the populations matched the BOLD signal better, which, if this is the case, is very relevant I think to my question 1 above.

We conducted two separate population analyses with different emphasis. The first approach is the population regression analysis, where we fitted a regression model to estimate whether the firing rate was significantly related to one of the following factors: morph level, ambiguity level, emotion judgment, or confidence judgment. Our models estimated the percentage variance explained by each factor. Thus, this approach is not sensitive to the direction of modulation (which condition has a greater firing rate). Using this approach, we could examine whether the neuronal population could differentiate each factor (morph level, ambiguity level, emotion judgment, or confidence judgment), regardless of direction of modulation.

The second approach is the simple linear average of normalized firing rate. This approach is directional and can show which condition (e.g., ambiguity level) has a greater normalized firing rate. Notably, the direction of this population average matched the BOLD signal for ambiguity (see Supplementary Fig. 9). As the reviewer mentioned, the exact relationship between the BOLD signal and neuronal population activity is unknown. Thus, we applied both approaches, which might reveal different information.

We have clarified these two approaches (on p.10):

“We next conducted a population regression analysis of all recorded cells, regardless of whether they were selected as emotion tracking or ambiguity coding. Note that this approach is not sensitive to the direction of modulation (which condition has a greater firing rate).”

And (on p.13):

“We lastly analyzed the overall mean firing rate of all recorded neurons (n=234) to investigate the overall activity of amygdala neurons in response to faces and how this response compared to the BOLD signal in our fMRI study, a directional approach (unlike population regression analysis) that can show which condition has a greater normalized firing rate.”

Minor Comment:

1. A more accurate reference for the notion that amygdala responses are related to the source ambiguity of fearful faces would be either Whalen et al 2001 in Emotion (if it is brain data you seek) or Whalen, 1998 in Current Directions for the original formulation of the notion.

We thank the reviewer for pointing this out and we have corrected the reference.

Reply to comments from Reviewer 4

The study aims at demonstrating differential representation in the human amygdala for ambiguity and emotion. The authors use two main approaches, imaging in humans and single-cell recordings in patients, and examined behavior in healthy controls, neurological patients, and amygdala-lesion patients. The physiological approaches and the populations complement each other nicely and enable examination of micro and macro properties of the representation in the human amygdala. This is rarely done, and is definitely a strong aspect of this study.

Unfortunately, the behavioral paradigm does not provide a reliable dissociation between the two main factors that are the target of this study: emotion and ambiguity. This is a major confound that cast doubt on the interpretation and therefore dampens my enthusiasm. In addition, the analyses are not convincing and not rigorous enough, and fall short of providing strong conclusions. As a result, the conclusions are not fully supported. The authors have valuable data in their hands, but much more work is required.

Below, we reply in detail to each concern. Together, we believe we were able to address all concerns raised. In particular, we would like to emphasize that our behavioral paradigm is well suited to make our principal claim, which is that emotion intensity and ambiguity can be disassociated at the neuronal level (not behavioral). Several other concerns are related to claims that we are not making, and we have clarified this accordingly.

Concerns

1. Ambiguity was defined operationally, yet a clear definition of ambiguity is missing. Therefore, the rationale of the study and the task design is problematic, as can be seen by the operational examination of ambiguity. Ambiguous faces were created using morphs, which guarantees indeed that the stimulus is ambiguous, but there is no validation of ambiguity in the perceptual emotion judgment.

First, we would like to clarify that our conclusions are specifically about emotion ambiguity, and not ambiguity in general. To clarify this, we modified the title to ‘Parametric encoding of emotion ambiguity and intensity in the human amygdala’.

Second, we added a specific definition of emotional ambiguity (and not ambiguity in general). Ambiguity here is the consistency in judging a given morphed face as happy or fearful. If this judgment is highly variable, ambiguity is high. Our behavioral data validates this definition of ambiguity: as can be seen in the psychometric curves, the anchor faces were judged consistently towards one particular emotion, while the most ambiguous faces were judged variably (50% chance towards one emotion and 50% chance towards the other). We explicitly define emotion ambiguity in the revised manuscript (on p.5):

“We define emotion ambiguity as the variability or consistency in judging a given morphed face towards a particular emotion. The more variable is the judgment, the more ambiguous is the face (Fig. 1b,c).”

Emotion ambiguity was thus designed into the experiment by construction (morphing the faces stimuli), and verified perceptually by the behavioral data.

Third, we have conducted additional experiments and analysis using ambiguous faces along the anger-disgust dimension. These faces do not have a fear component. Nevertheless, we found similar ambiguity coding in the amygdala. This validates that these neurons code emotion ambiguity. Please refer to our response to Reviewer 1’s Question 1 for details of this new analysis.

2. Moreover and importantly, it is not clear that emotion and ambiguity are indeed dissociated in this design. For example, the possibility of morphed faces being perceived as neutral was not examined (and confidence rating does not rule out this possibility). Another possibility is that ambiguity is not homogenous or linear along the emotional axis (e.g. it is different next to fear vs next to happy etc.). This can be tested behaviorally in principle, but they do not do so, and it would be much better if the design took care of it to begin with. These are major confounds for all presented results.

We thank the reviewer for raising these questions, which we have now clarified. Our conclusions do not require that emotion intensity and emotion ambiguity are completely orthogonal, only that they are not perfectly correlated. Indeed, emotion intensity and emotion ambiguity co-vary for each stimulus, but the relationship of these two variables to the morph levels was very different. Whereas intensity was maximal for anchors (fear or happy), ambiguity was maximal for intermediate morphs. Our principal result is that there is a group of neurons which co-vary systematically with ambiguity such that they fire the same to unambiguous stimuli of different emotions. Thus, this group of neurons does not differentiate emotions. This clearly shows that, at the level of single neurons, that ambiguity and emotion is disassociated. This single-neuron result is the claim of the paper, and this claim does not depend on the stimuli co-varying independently along these variables.

As already noted, we did define emotion ambiguity very clearly, based on behavioral data. The behavioral data confirms that emotion ambiguity peaks for 50/50 morphs and is lowest for the anchors, just as one would expect.

The reviewer also wondered about the possibility that ambiguity is not homogenous or linear along the emotional axis. To address this question, we have included the following quantifications in Methods (on p.27):

“Using an operational definition of ambiguity—the variability in choices (the percentage of choices that are not the same as the dominant choice, e.g., the percentage of choosing fear for a 30% fear / 70% happy face where happy judgment is the dominant choice), we found that the variability in choices for three levels of ambiguity increased in similar steps in neurosurgical patients (anchor: 3.15 ± 3.08 (mean \pm SD), intermediate: 14.1 ± 7.97 , most ambiguous: 32.3 ± 5.55 ; the difference between intermediate and anchor: 10.9 ± 7.20 , the difference between most ambiguous and intermediate: 18.2 ± 10.0 ; paired t-test on difference: $t(13)=1.69$, n.s.). Furthermore, the psychometric curves were symmetric about our morphs levels, arguing that our grouping of ambiguity levels was not biased towards a particular extreme of emotions.”

3. There is no dissociation between the face perception and the decision. While the results reflect both processes, they are attributed and interpreted to the perception. Similarly, there was no clear dissociation between the decision and the confidence judgment. In this context, fMRI results were mostly significant 6 seconds after stimulus onset, which was after a decision was made and after confidence ranking. Therefore, these results may be correlated with confidence levels.

Similarly, single neurons PSTH should be separated for 250-1000ms and 1000-1750ms to separate perception and decision. It should be examined if results are more/less robust for decision time locked or confidence rating time locked analysis.

The reviewer is correct that, to some extent, and in particular for the fMRI data, we cannot dissociate perception from decision processes. First, please note that we do not in fact claim that we disassociate perception and decision making processes (in fact, the word perception does not appear in our manuscript). In general, it is not possible to map these psychological concepts to discrete processes in the brain, which at any point of time reflect these and many other processes. All that we claim is that at the level of single-neurons and the BOLD

signal, there is a representation, by separate groups of neurons, of emotional intensity as well as emotional ambiguity for faces. However, we do not make any claims regarding whether these processes relate to perception, decision making, or both (or indeed any other particular psychological process that could utilize the information in these representations). Using a population analysis, we further show that the neuronal population response predicts the choice that subjects make. Based on this, the third claim is that the population response doesn't simply represent the input, but we similarly refrain from making any inference about perceptual processes (which do not need to faithfully represent sensory input). We have clarified that we do not make claims about perception.

Second, we do not claim to disassociate confidence and the decision itself and it is in fact increasingly becoming clear that these are not separable (i.e., see our recent work on this, Rutishauser et al., Nat Neurosci, 2015; also see Kiani and Shadlen, Science, 2009).

Third, BOLD-fMRI typically has a long latency of response. In our fMRI design, we omitted confidence rating and increased the duration of inter-trial-interval (ITI) from 1-2 seconds to 2-8 seconds (jittered randomly with a uniform distribution), for the reason mentioned by the reviewer (see Methods). Furthermore, in our single-neuron task design, responses were only made after face offset, and there was a blank screen of 500 ms before confidence rating. Therefore, in both fMRI and single-neuron designs, perception is not correlated with confidence levels.

Fourth, to directly address the additional analysis suggested by the reviewer, we further aligned trials to button presses. We have included the following results in Supplementary Results (on p.15):

“To provide an initial analysis that might partly separate perceptual and decision processes, we also aligned trials to button presses and quantified the response of each neuron based on the number of spikes in preparation to button press (1s window, starting 1000 ms before button press). We first found that 16 neurons (6.84%; binomial $P=0.080$) differentiated fear and happy decisions (11 increased and 5 decreased with fear response; unpaired t-test). We could still select 26 emotion-tracking neurons (11.1%, binomial $P=5.29\times 10^{-5}$; 14 neurons increased firing rate with fear levels and 12 neurons decreased firing rate with fear levels) and 31 ambiguity-coding neurons (13.3%, binomial $P=2.90\times 10^{-7}$; all neurons decreased firing rate with ambiguity levels), suggesting that the representations of emotion intensity and emotion ambiguity were still a robust finding, even using this later time window for analysis.”

We further carried out another new analysis, as suggested by the reviewer, that aligned responses to confidence ratings. We have included the following results in Supplementary Results (on p.16):

“In another analysis of a different temporal window, we aligned response to confidence ratings and quantified the response of each neuron based on the number of spikes in response to confidence rating (1s window, starting at confidence rating onset). Since we omitted confidence rating for Patient C34, all neurons from this patient were excluded. In the remaining 205 neurons, we found 31 emotion-tracking neurons (15.1%, binomial $P=1.25\times 10^{-8}$; 15 neurons increased firing rate with fear levels and 16 neurons decreased firing rate with fear levels), 12 ambiguity-coding neurons (5.85%, binomial $P=0.23$; 9 neurons increased firing rate with ambiguity levels and 3 neurons decreased firing rate with ambiguity levels), as well as 16 neurons (7.80%; binomial $P=0.029$) that differentiated fear vs. happy decisions (9 increased and 7 decreased with fear response). Since Patient C26 only rated ‘Very Sure’ as confidence across all trials, we further excluded all neurons from this patient (two sessions), resulting in a total 161 neurons remaining for the analysis of confidence. We found that 28 neurons (17.4%, binomial $P=2.13\times 10^{-9}$) could differentiate levels of confidence (one-way ANOVA) and 24 neurons (14.9%, binomial $P=5.11\times 10^{-7}$) significantly correlated with levels of confidence (trial-by-trial

correlation). However, only 2 neurons selected by ANOVA and 1 neuron selected by correlation were emotion-tracking neurons, only 3 neurons selected by ANOVA and 3 neurons selected by correlation were ambiguity-coding neurons, and only 1 neuron selected by ANOVA and 2 neurons selected by correlation also encoded decisions. These results suggest that emotion-tracking and ambiguity-coding representations are no longer evident at this later time window.”

Lastly, as suggested by the reviewer, to explore whether the neural signal is present throughout the time interval that we used to select these neurons, we separately used two intervals (250-1000ms and 1000-1750ms after stimulus onset) to select emotion-tracking and ambiguity-coding neurons. We have found similar results: Using an early interval of 250 ms to 1000 ms after stimulus onset, we still found 28 emotion-tracking neurons (12.0%, binomial $P=7.36\times 10^{-6}$; 16 neurons increased firing rate with fear levels and 12 neurons decreased firing rate with fear levels) and 28 ambiguity-coding neurons (12.0%, binomial $P=7.36\times 10^{-6}$; 4 neurons increased firing rate with ambiguity levels and 24 neurons decreased firing rate with ambiguity levels). Using a late interval of 1000 ms to 1750 ms after stimulus onset, we also found 21 emotion-tracking neurons (8.97%, binomial $P=0.0036$; 15 neurons increased firing rate with fear levels and 6 neurons decreased firing rate with fear levels) and 29 ambiguity-coding neurons (12.4%, binomial $P=2.60\times 10^{-6}$; 5 neurons increased firing rate with ambiguity levels and 24 neurons decreased firing rate with ambiguity levels). It is also worth noting that for each interval, the composition of emotion-tracking neurons (χ^2 -test; early: $P=0.60$; late: $P=0.55$) and ambiguity-coding neurons (χ^2 -test; early: $P=0.55$; late: $P=0.36$) did not differ from those selected using the entire time interval (250 ms to 1750 ms after stimulus onset). Therefore, our results have confirmed that the neural signal is present throughout the time interval that we used to select these neurons.

4. In the manuscript, participants pushed a right button, using right hand, for FEAR and a left one, using left hand, for HAPPY. Laterality of responses to fear/happy faces was not counter balanced, which can confound the results regarding amygdala response laterality in the fMRI experiment.

We did not counterbalance button press in the fMRI experiment because we wanted to keep it as similar as possible to the single-neuron recording. The observed BOLD-fMRI was not likely due to button press, because (1) when we regressed out RT, we still observed very similar results (Supplementary Fig. 4), and (2) the pattern of activation, especially the ambiguity coding in the right amygdala which had balanced button press for each level, could not be simply a result from button press (note that although not significant, both sides of the amygdala had signals for both emotion intensity and ambiguity). Had we examined regions of the brain involved in motor representations (motor cortex, SMA, basal ganglia, cerebellum), there would be reason for concern here, but we would have no reason to expect any effect of which hand was used on amygdala responses.

We further analyzed the relationship between RT and neuronal firing rate, and found that the amygdala's response to emotion ambiguity and intensity was barely related to the output button responses. Therefore, the BOLD-fMRI was also not likely associated with button responses.

5. The authors suggest that patients with lesions in the lateral amygdala were not impaired in ambiguity judgment. Nonetheless, results that relate ambiguity to neural activity are mainly focused on the lateral amygdala, both in single neurons and fMRI. If these neurons were encoding ambiguity, one should expect a difference between lesioned and healthy patients in ambiguity judgment.

We thank the reviewer for pointing this out. There is a network of brain areas that encodes ambiguity. Indeed, in a separate manuscript, we found a strong ambiguity signal in a distributed network of brain areas including both the anterior and posterior cingulate cortices, and through fMRI functional connectivity analysis (DCM), we

found that the ACC functionally modulates the amygdala's response to ambiguity. It is of course logically quite possible, and often observed, that regions of the brain that represent certain information are not causally necessary to produce behavior based on that information—likely because the information is also coded redundantly elsewhere in the brain, as we believe is the case for ambiguity, as noted above. We have consequently added the text below to the Discussion (on p.18):

“It is worth noting that the intact judgment of ambiguity in amygdala lesion patients suggests that the amygdala's response to ambiguity is not an essential component for behavioral judgments of emotion ambiguity, and that such judgments may sufficiently rely on structures elsewhere in the brain that also represent information about ambiguity. Future studies could further probe this issue by conducting fMRI studies like ours, but in patients with lesions to the amygdala.”

6. Both in fMRI and single neurons results, the authors interpret the increased response to both anchors (with decreased to no response to morphs) as ambiguity coding. Essentially, this is a response to emotional intensity or confidence, and does not imply ambiguity coding. In line with this, it was suggested that subpopulations of neurons in the amygdala encode both reward and punishment error (reference # 50 - this reviewer is not an author on this study).

Specifically, single neurons ambiguity coding was reported for a large fraction of units with increased firing rates in response to both valences and only three neurons increased the response to the morphed faces, i.e. respond to actual ambiguity. In addition, in figure 4h the upper unit out of the three does not seem to be positively correlated with ambiguity. In addition, there was no example of a true two steps gradual response to ambiguity that was also gradual in both happy and fear faces.

We respectfully disagree and have clarified the manuscript to make this clear. It is important to keep in mind that intensity and ambiguity are of course defined relative to a particular emotion. First, we measured emotion intensity and valence (**Supplementary Fig. 1b-g** and **Supplementary Results**) and found that they monotonically increased and decreased with fear levels, respectively, and did not have the same response profile as ambiguity response. Therefore, the ambiguity signal could not be explained by valence or intensity dimensions. We have further updated Fig. 1b to clarify the relationship between emotion intensity and emotion ambiguity.

Second, ambiguity and confidence are clearly related as we discussed (on p.18):

“Ambiguity and confidence

Decisions about faces are frequently ambiguous, including those about facial emotions, and optimal decision making thus requires an assessment of ambiguity. We thus expect that an assessment of uncertainty is a crucial variable represented in areas concerned with decision-making about faces. We have further shown that the activity of amygdala neurons correlates with the confidence in emotion judgment. Together, this shows that two closely related variables with meta-information about the decision itself (fear/happy) are represented in the amygdala, one based on objective discriminability of the stimuli, and the second based on the subjective judgment of their discriminability: ambiguity and confidence. This is interesting, because a judgment of confidence is thought to be a direct consequence of an assessment of uncertainty (Kepecs and Mainen, 2012). The mechanisms by which such confidence judgments are made, however, remain poorly understood. It has been suggested that confidence judgments rely on a modified “race to threshold” approach, which relies on integrating the evidence for and against a hypothesis separately. The difference between the two quantities of integrated evidence is proportional to the subjective confidence in the decision (Kepecs and Mainen, 2012). In a

recognition memory task, we have recently shown that the activity of a specific subset of human amygdala neurons is compatible with this model: the stronger the integrated difference between a familiarity and a novelty signal carried by individual neurons, the larger the subjective confidence (Rutishauser et al., 2015b). Together, this raises the important question of whether the amygdala provides a general ambiguity signal that provides the underlying information necessary to judge the confidence in decisions about internal states in general. This important hypothesis remains to be investigated by comparing the activity of the same neurons in several different decision-making tasks.”

Third, the ambiguity signal identified in our study is not likely associated with reward (on p.19):

“Ambiguity vs. reward-related signals

Many neurons in the macaque amygdala change their firing rate in response to rewards or stimuli predicting the later delivery of rewards or punishments. For example, amygdala neurons differentiate between cues that predict delivery of positively-or negatively valued rewards (Paton et al., 2006) and they respond to reinforcements that are unexpected (Belova et al., 2007). Reward-related coding of uncertainty has also been identified in macaque midbrain dopamine (Fiorillo et al., 2003) and septal neurons (Monosov and Hikosaka, 2013), which signal after cues that predict unreliable rather than reliable rewards (Tobler et al., 2005). In these studies, individual visual stimuli are associated with the probability of obtaining reward after extensive conditioning. In contrast, in our task no trial-by-trial feedback or reward is delivered, no training on the ambiguity associated with each stimulus is provided and the neurons we identified have a stimulus-evoked response unrelated to reward delivery or expectation. We thus here show a reward- and reward-value independent representation of ambiguity in the amygdala. Human single-neuron recordings are uniquely suited to test this hypothesis, because no training before the recording is required and patients are able to perform the task without rewards and punishments to incentivize correct performance.”

As the reviewer pointed out, we indeed found that the majority of ambiguity-coding neurons have a greater response to anchors than morphed faces. We have discussed this finding in the revised manuscript (please refer to our response to Reviewer 1’s Question (2-v)). Since we used trial-by-trial correlation rather than the average to select ambiguity-coding neurons, the average response at the intermediate ambiguity level could be greater than that at the most ambiguous level (the level average but not the trial-by-trial correlation was possibly driven by trial outliers). Furthermore, although ambiguity-coding neurons were selected by trial-by-trial correlation, some neurons also (partially) showed two steps between levels (unpaired two-tailed t-test), for example the neuron in Fig. 4b (anchor: 9.11 ± 5.87 Hz (mean \pm SD), intermediate: 5.67 ± 3.42 Hz, $t(65)=2.74$, $P=0.0080$; most ambiguous: 4.66 ± 3.14 Hz, $t(134)=1.78$, $P=0.077$) and the neuron in Supplementary Fig. 6e (anchor: 7.11 ± 3.16 Hz, intermediate: 4.73 ± 3.11 Hz, $t(65)=2.40$, $P=0.019$; most ambiguous: 3.65 ± 3.44 Hz, $t(134)=1.86$, $P=0.066$). We have further included these statistics in the figures and figure legends. But note that our data have suggested a sharper transition from anchor face to ambiguity—the difference between anchor vs. intermediate ambiguous faces (mean normalized firing rate: 0.53 ± 0.44 ; mean \pm SD) was greater than that between intermediate vs. most ambiguous faces (0.23 ± 0.24 ; paired t-test: $t(28)=2.96$, $P=0.0062$, $g=0.86$; **Fig. 4c,f**). We found two neurons that were both emotion-tracking and ambiguity-coding.

Minor concerns

- In the manuscript, there is a comment that the representation of ambiguity is the same in all amygdala subnuclei, but the authors only show 4 neurons in the CE, of them only one responded to fear. Thus, the above cannot be stated.

We removed this statement.

- Figure 1C. Please add a psychometric curve for each of the four different faces sets.

We added psychometric curves for each face set to Supplementary Fig. 1 as requested.

Figure legend. Group average of psychometric curves for each facial identity (a-d, each is one identity). The psychometric curves show the proportion of trials judged as fearful as a function of morph levels (ranging from 0% fearful (100% happy; on the left) to 100% fearful (0% happy; on the right)). Shaded area denotes \pm SEM across subjects/sessions.

- Three levels of confidence allow very little variability in confidence levels.

Three levels of confidence are widely used in many experiments, and have been found to be sufficient for many critical comparisons. Given limited number of trials, it is not feasible to use more levels of confidence. For example, (Rutishauser et al., Nat Neurosci, 2015) and (Kirwan et al., J Neurosci 2008) used 3 levels of confidence, (Kanai et al., Consciousness and Cognition, 2010) and (Persaud et al., Nat Neurosci 2007) (high vs. low in post-decision wagering) used 2 levels of confidence, and in a similar idea of confidence rating, (Kiani and Shadlen, Science, 2009) used a sure and safe choice (a form of post-decision wagering) to probe high vs. low confidence in monkeys (see (Kepecs and Mainen, Phil. Trans. R. Soc. B, 2012) for a comprehensive review). Our own behavioral results have also shown that three confidence levels can well capture the difference in ambiguity levels. (Fig. 1 and Supplementary Fig. 3).

- Was there a difference in behavior as a function of the block number? Namely, was there learning during the experiment that resulted in higher consistency in the last block?

We compared behavior between the first half and last half of the trials. For lesion patients, neurosurgical patients, and controls, their psychometric curves look very similar, indicating no learning across trials (see figure below). x_{half} (midpoint point of the psychometric curve) did not differ significantly between the first and second half of trials for lesion patients (first: $42.0 \pm 3.37\%$ (mean \pm SD), second: $46.6 \pm 1.48\%$; paired t-test; $t(2)=2.02$, $P=0.18$), neurosurgical patients (first: $53.8 \pm 7.52\%$, second: $53.7 \pm 4.26\%$; $t(13)=0.068$, $P=0.95$), and controls (first: $50.6 \pm 5.50\%$, second: $51.8 \pm 6.60\%$; $t(14)=0.72$, $P=0.48$).

We computed within-subject consistency using the logistic model parameter α (the steepness of the curve)—the greater the α , the steeper the transition from fear choices to happy choices, and thus the more consistent the behavior. We found that α did not differ significantly between the first and second half of trials for lesion patients (first: 0.17 ± 0.057 (mean \pm SD), second: 0.16 ± 0.040 ; $t(2)=0.26$, $P=0.82$), neurosurgical patients (first: 0.098 ± 0.034 , second: 0.12 ± 0.093 ; $t(13)=0.76$, $P=0.46$), and controls (first: 0.13 ± 0.071 , second: 0.13 ± 0.052 ; $t(14)=0.25$, $P=0.81$), again indicating no learning across trials.

We further computed the between-subject consistency as the mean variance across subjects (averaged across all morph levels). As can be seen from the psychometric curves, lesion patients (first: 64.2, second: 26.2) and neurosurgical patients (first: 138, second: 90.3) had lower between-subject consistency (higher variance) in the first half of trials. However, controls (first: 117, second: 150) had higher between-subject consistency in the first half of trials.

- All lesion conclusions are based on three, possibly over trained, lesioned patients. Statistical analysis, for example t -test, on such small group is problematic.

We agree with the reviewer that such lesion patients, especially rare patients, are often tested in various tasks and can be over trained, a caveat that is present in many lesion studies (as well as monkey studies). Nevertheless, due to rareness of these subjects this is unavoidable. We have further replicated our original findings using a multiple case-study approach, in which each patient was compared to the control group using a permutation test. Please refer to our response to Reviewer 2's third last major question.

Furthermore, our results were consistent with another study (Terburg et al., Translational Psychiatry 2012) showing that a different group of 5 amygdala lesion patients with the same lesions as ours have increased fear judgment of emotions in a similar task. This independent replication makes it less likely that our results were specific to our group of 3 subjects.

- Please avoid reporting non-significant statistical test as a result. For instance, page 5, $p=0.12$ for lesioned patient.

We have now ensured that all results with $P>0.05$ are reported explicitly as not statistically significant in the main text. We continue to report the tests we carried out and the numerical results, of course, since doing otherwise would provide biased reporting of our study.

- General statements of accepting the null hypothesis since they were not rejected are incorrect and should be avoided. For instance, page 10, first paragraph, chi square test.

We thank the reviewer for pointing this out and we have corrected it.

- Figure 2. Please show both results for both hemispheres, to show the dissociation between responses in the right and left amygdala.

Fig. 2f shows this.

- Page 6, bottom: the interpretation of figure 2e is incorrect, should be referring to anchor vs. intermediate and anchor vs. most ambiguous.

We thank the reviewer for pointing this out, but we did mean to compare anchor vs. intermediate and intermediate vs. most ambiguous, because we would like to know whether different levels of ambiguity were encoded with similar strength.

- Page 8, first paragraph: please correct supplementary figure 10 referencing.

We thank the reviewer for pointing this out and we have corrected it.

- Page 8, second paragraph: errors of R squared are not interpreted and appear to be larger than the detected effect.

We thank the reviewer for pointing this out and we have added "mean \pm SD".

- Figure 3: PSTH results were examined for 250ms bins. Please check results across different bin sizes. In addition, error bars should be added to the PSTH to demonstrate that effects do not result from outliers.

We have tried different bins for both single neuron examples and group PSTH. Please find the plots below. We have added the following text to Methods (on p.28):

"PSTHs with different bin sizes were analyzed and qualitatively the same results were derived."

Single neuron examples:

Group PSTH:

Figure legend. PSTH with different bin sizes. PSTH is color coded according to morph or ambiguity levels. The left gray bar shows face stimulus onset (fixed 1s duration). Shaded area and error bars denote \pm SEM across trials. Asterisk indicates a significant difference between the conditions in that bin ($P < 0.05$, one-way ANOVA, Bonferroni-corrected).

We have included error bars for PSTH:

Single neuron examples shown in Fig. 3 (the same convention for figure legend):

Group PSTH shown in Supplementary Fig. 7 (the same convention for figure legend):

- Please add analysis for decision coding in single neurons (and not the entire population).

We have included the results of this analysis in Supplementary Results (on p.15):

“Decision coding

First, we found that 20 neurons (8.55%; binomial $P=0.0074$) differentiated fear and happy decisions using all trials (13 increased and 7 decreased with fear response; unpaired t-test), but only 3 of these neurons were also emotion-tracking neurons and the overlap between these neurons and emotion-tracking neurons was not significant (χ^2 -test: $P=0.91$), suggesting that it was an independent population of neurons compared with emotion-tracking neurons. Breaking down for morph levels, there were 11 neurons (4.70%) showing higher mean firing for fear judgment at all levels whereas there were 9 neurons (3.85%) showing higher mean firing for happy judgment at all levels. At the most ambiguous morph level (50% fear / 50% happy), 15 neurons (6.41%; binomial $P=0.13$) could differentiate fear vs. happy response, showing that given identical stimulus, neurons could encode subjective judgment, consistent with our previous finding (Wang et al., 2014). Similarly, at the most ambiguous 3 morph levels (40% fear / 60% happy to 60% fear / 40% happy), 15 neurons could differentiate fear vs. happy response. Interestingly, 28 neurons (12.0%, binomial $P=7.36 \times 10^{-6}$) had a higher firing rate for fear response given fearful faces while a higher firing rate for happy response given happy faces, whereas 5 neurons (2.14%) showed an opposite trend. Such interaction between stimulus and response indicates that these neurons track how much decisions match with stimulus.”

Reviewers' Comments:

Reviewer #1 (Remarks to the Author):

I thank the authors for their detailed response to my comments and those of the other reviewers. Many of the issues I raised have been fully addressed. However, I do still have a number of outstanding concerns as detailed below.

Comments on initial general response from authors:

(1) I agree with the authors that the paper is best served by leaving in all three datasets (fMRI, lesion, single unit recording) and with their sentiment that 'Deleting a dataset from a measure that does not fit so well with the others would produce an artificial consensus that, we feel, is not in the best interests of the field. '

(2) The inclusion of the anger/disgust morph single unit recording data is valuable and goes a long way to addressing my concerns that the main findings might be specific to the happy/fear morph stimuli used. In particular, happy – fear expression blends are rarely found in real-world settings, so it is valuable to establish that effects of ambiguity are similar for a dimension where blends of the anchor expressions are more likely to be encountered in real-world settings.

Comments on authors' response to my first round review

Issue 1. Generalizability of findings given use of a single expression continuum (happy/fear). See (2) above. The authors have fully addressed my concerns here.

Issue 2i. Prior comment 'The findings from these analyses indicate greater amygdala responsivity to the 100% fear or happiness 'anchor' faces than to more ambiguous intermediate face morphs.... It is potentially confusing to frame this as having shown 'representation of ambiguity' in the amygdala.'

Authors' response: We have clarified this issue in the text by referring to 'parametric encoding' of ambiguity instead of representation.

New comments: I still think the phrasing is not ideal and may cause confusion. For example in the abstract the authors state that 'At the single-neuron level, we studied single neurons and found two populations of neurons, one whose response correlated with the gradual change of emotion intensity and a second whose response signaled levels of ambiguity. Together, our results indicate that the human amygdala processes both the intensity of emotion in facial expressions as well as the ambiguity of the emotion shown'.

I think several of the terms used here (and in the main paper) are likely to confuse the reader. The Introduction focuses on dominant prior findings that the amygdala responds positively to (i)

face fearfulness and (ii) face ambiguity. Readers are likely to assume a statement that ‘the amygdala processes the ambiguity of the emotion shown’ replicates prior findings of heightened amygdala activity to ambiguity rather than the reverse. Hence, I think it is important that the authors phrase their key findings in a manner that is completely clear. So for example in the abstract (and elsewhere) they could state that one population of neurons ‘showed activity that decreased as a linear function of ambiguity’.

An equivalent issue pertains to the description of the other population of neurons ‘whose response correlated with the gradual change of emotion intensity’. Here the authors are referring to the finding that some neurons show increased firing as a linear function of expression fearfulness and others increased firing as a linear function of expression happiness (which could also be phrased as decreased firing or suppression of activity as a linear function of expression fearfulness). I would ask the authors not to use the term ‘emotion intensity’ as it isn’t transparent as to whether this equates to expression fearfulness, expression happiness or (as some might have predicted) closeness to dimension ends. Referring instead to ‘expression fearfulness’ or ‘percentage of fear in the expression’ would be much clearer. Similarly for the fMRI data, the authors should specify that activity decreased as a linear function of expression fearfulness and expression ambiguity.

Issue2ii I thank the authors for adding text to address the point raised here.

Issue 2iii. I thank the authors for adding text to address the point raised here. I have just one remaining comment, the authors state that ‘Therefore, more pronounced adaptation for stimuli in the middle than at the extremes of the face continuum could in principle be one mechanism that explains the amygdala’s response to ambiguity we found in our study. However, we used a sufficient number of distinct stimuli, and their order was completely randomized, making face adaption implausible in our protocol.’

I think the last statement is too strong given that Aguirre and colleagues have shown that even when stimulus order is randomized adaptation can occur both for (i) face identity and (ii) face expression. I do however think it would be reasonable to state something along the lines that ‘adaptation to facial emotion is likely to be lower when facial identity is also manipulated on a trial to trial basis as in the current study’

Issue 2iv. I thank the authors for clarifying this – my concerns here are fully addressed.

2v. The issue raised here was a very important one given the focus of the paper:

‘Finally, if it is indeed the low ambiguity of the anchor faces that is critical, then an issue that also requires discussion is how this fits with other accounts suggesting that the amygdala is sensitive to high levels of ambiguity (e.g. Whalen TICS 2007). Indeed the work on ambiguity reviewed in the Introduction points to the amygdala response varying positively with level of ambiguity.’

I think it needs addressing more comprehensively than the authors have done as yet. In the Introduction the authors originally had the following text:

'On the other hand, the amygdala is also crucial for identifying ambiguous stimuli and in modulating vigilance and attention as a function thereof. For instance, the BOLD-fMRI signal in the amygdala is correlated with the level of ambiguity in decision making and focal amygdala damage undermines decision making under ambiguity. In addition, unpredictability of stimuli even without any motivational information prevents rapid habituation of single neuron activity in the basolateral amygdala in mice and causes sustained neural activity in the amygdala in humans.'

This is strong, cross-species evidence for the amygdala showing increased firing with ambiguity (i.e. the opposite to the current finding). The authors have now added a single sentence to provide (fMRI) evidence that, in some cases the amygdala might show activity that decreases with ambiguity-

'However, other studies have shown suppression of BOLD activity for faces with the most ambiguous trustworthiness'

This is really very limited relative to the huge evidence in the opposite direction given above and seems a bit disjointed. Is there nothing else? If not, then the authors really need to acknowledge that the majority of the findings in the literature support a prediction that amygdala activity will increase as a function of ambiguity.

In this context, I also do not find the discussion of the counter-intuitive ambiguity result in the discussion sufficient. Here the authors state that 'Some studies have shown that the amygdala responds maximally to ambiguous stimuli or ambiguous choices. In contrast, we found that both the neuronal population response and BOLD signal were maximal for the least ambiguous faces. This is consistent with other studies showing suppression of BOLD-fMRI activity for faces with the most ambiguous trustworthiness'. This would be okay if there really was an equal balance of evidence in both directions in the prior literature as this seems to imply, but there isn't (or if there is the authors need to include a lot more on findings that amygdala activity increases to low ambiguity in the introduction). This is compounded by the subsequent statement that 'Here we interpret any change in firing rate, or in BOLD signal, as carrying information, and therefore do not further interpret the sign of that change'. I found this rather frustrating / inadequate. While it might be justifiable to say it is hard to interpret the direction of BOLD activity, this is not true for single neuron recordings (which indeed is their advantage) so the authors really need to tackle the issue that their ambiguity finding is in the opposite direction to what many would predict rather than just state they don't think this is important.

Issue 3) The authors have fully satisfied my concerns here.

Issue 4) Prior comment: 'With regards to the lesion study, how do the authors reconcile the finding that amygdala lesion patients were 'more likely to judge faces as fearful' (pg 5) with prior

reports that patients with amygdala damage show reduced ability to recognize fearful expressions (Adolphs et al., Nature, 1994)?'

I thank the authors for clarifying their position on this - that the prior literature would lead to the prediction that patients with basolateral lesions might show hyper-vigilance for fear. Given the importance of this to the authors' predictions I believe that this should be included in the Introduction, where at the moment only lesion data linking amygdala damage to impaired fear recognition is discussed. (Currently when the authors present their lesion results, they introduce the BL specific lesion background for the first time and then refer the reader to the discussion, this isn't optimal).

Minor issues: all fully addressed.

Additional issues

1) request relating to Issue 4.

The authors now clarify that their lesion predictions are based on the damage being limited to the BL nucleus. For their single neuron recording, they also comment on the location of the neurons that track expression fearfulness and ambiguity (BL vs Central Nucleus). Given this, it would also be valuable if the authors could likewise add a small section to their fmri analyses to discuss if the location of the activation observed is more consistent with it being a BL or CN signal. (There are probabilistic atlases of the amygdala which give the likelihood that a given activation is in the BL or CN, or better still the authors could use each individual's anatomical scan to roughly estimate this). It is fine to indicate that fMRI data is not of the resolution to make firm conclusions regarding this but it would still be valuable to have something along these lines (see also work by Whalen who has argued that a given 'z' co-ordinate can be used to crudely distinguish which part of the amygdala one is in, Kim et al., 2003; Somerville et al., 2004.)

2) I agree with reviewer 2 that ideally the ratings of intensity and valence should be acquired from participants of the same demographic background (ethnicity and country of birth /long-term residence). This also raises the broader issue of whether any discrepancies in findings across methodologies may actually reflect different responses to the morphed faces in Western Caucasian versus Eastern Asian participant groups. As reviewer 2 points out there are known differences in expression processing between these groups. Hence, if possible, it would be useful for the authors to acquire an additional set of face ratings from control Western Caucasian participants. If these do not differ from those of the Eastern Asian control group it would both provide reassurance that the control ratings are pertinent to the patient groups (as well as the fMRI participants) and that differences in expression perception between the different participant groups do not account for differences in findings between methodologies.

Reviewer #2 (Remarks to the Author):

I think the authors took great care of my comments and did an excellent job in addressing all of them. I can thus recommend this paper for publication.

Reviewer #3 (Remarks to the Author):

The authors have addressed all my questions.

Reviewer #4 (Remarks to the Author):

this is a resubmission of a revised manuscript.

The authors did considerable work addressing the concerns.

However, there are still fundamental concerns, and in general, I think this study is more suitable for a specialized journal. The paradigm is not sophisticated enough, rich, or well controlled, when compared to other studies (behavioral and fmri) these days. Recording human single-units, even if valuable and rare data, is not enough to compensate for this. In addition, the findings in single-units is not analyzed to the level that one can expect in physiology studies.

I am really sorry to say so, because I can see there was considerable work done here and they have valuable data, but currently the main claim is only indirectly supported.

Major

- Ambiguity is not well defined here and can be attributed to other factors. The main one here is that ambiguity is the inverse of intensity. I understand the authors' claim, but this task does not decorrelate the 2 factors. Here is why: the anchors have the higher intensity, and therefore any correlate that is high/low at the morphs and low/high at the anchors can be interpreted as intensity coding rather than ambiguity coding. Since a big majority of their finding in single neurons suggest that there is higher response to anchors, this makes it even more suspicious.

- If there is gradual coding of emotion intensity within the amygdala, a very reasonable hypothesis based on the wide literature (call it attention if you like), then it will be expressed as inverse coding to what they call ambiguity.

Indeed, "most of which (30/36) had a higher firing rate for the anchor compared to the morphed faces"; again, it seems most neurons by far respond to the emotionally identified face (anchor), which raise the concern that the claim for ambiguity coding is a bit exaggerated.

And later "most (29/32) of which had the maximal firing rate for the anchors (which have low ambiguity). We refer to this group of cells as ambiguity-coding neurons" ...

Yes, I am aware that information and coding can be in inverse correlations as well, and decreasing FR can be as informative as increasing FR. Yet it still casts doubt whether there is real coding for ambiguity or simply to intensity of emotion (independent of positive/negative).

- Single units are very poorly sorted (likely multi-units). I know this is the unfortunate standard with human recordings, but in the current case it is a real confound for interpreting ambiguity-coding (because it can be a summation of 2 or more neurons each with a tuning curve increasing or decreasing in opposing manner as a function of fear).

- The definition of ambiguity as consistency is problematic. It is identical to the mean in this case (the choice is binary/AFC), am I missing something here?

Less major

- In fig.3b, there is no real linear relation, only 0% is higher. Is this really the “best typical example”?

- The exemplar rasters are not convincing in fig.4, and the matching PSTH only very weakly show a trend towards the finding. Again, is this the best example?

- The imaging analysis is not so clear as well. I trust it is correctly done, but I could not understand how they used the localizer to later test for left or right amygdala in looking for which codes what. Also, In fig2b, the result comes from the happy face alone, so It seems; and in fig.2d., there is no real tuning/gradual-coding, just a higher response to no-emotion (ambiguity is the interpretation). overall, as a result, the interaction in fig.2f is a very weak effect (even if significant).

- A main motivation is very argumentative: “and leaves open the possibility that perhaps prior findings in regard to fear are derivative to this function, since fear can signal ambiguity (by indicating a threat without specifying where or what that threat is”.

Fear signals ambiguity? Why not the other way around (ambiguity drives fear, I think this is intuitive, and indeed much of the data can be interpreted this way). Not to mention that in many studies it is very clear to the animal what and where the threat is.

I simply do not see these options as ‘two theories’ of amygdala function.

- They find that behaviorally, amygdala patients have altered threshold for reporting fear. This is another example of a confound, showing that the ambiguity regime can actually covariate with emotional (and there is no independent measure of intensity to rely on).

- In general, they find that anchor faces induce more activity in the amygdala. The evidence for

similar levels of activity across ambiguous faces from both sides (happy and fear), is obviously indirect (statistically).

- “neurons from each individual recording session were considered independent even if they were from the same patient because it was not possible to reliably record the same neurons across sessions”

there is a huge difference between not claiming these are the same neurons (e.g. that these are not chronological recordings of neurons to be compared across days), and treating them as independent neurons to be added to the overall pool. I hope the authors can see this. This can increase the statistical power without taking into account dependency. Please repeat analyses on a subset as well to see if the results are replicated.

- Overall, there are neurons of each possible type, and I am unconvinced that there is real ambiguity and not some homogenous distribution of responses that vary with the stimulus. The claim for proportions that are beyond expected from chance is not well established in my view. Some better controls for estimating the proportions of different populations are required. E.g. reshuffle the stimuli and re-analyze, to see if similar proportions of neurons would correlate with such imaginary property.

Original (black), our reply (blue), copy of text added to manuscript (blue underline)

We thank all four reviewers for their constructive and expert comments. In this revised manuscript, we have made the following major improvements: (1) As requested, we collected new behavioral ratings from Western Caucasian subjects and found that they are similar to those from Asian subjects. (2) As both reviewers noted, our findings showed that the amygdala primarily decreased response to ambiguous faces, and we clarify this now throughout. (3) We replaced “emotion intensity” with the more precise expressions “degree of fear” or “degree of happiness” throughout the manuscript. We have retained “intensity” in the title but clarified that this refers to specific emotions; our title now reads, “The intensity of specific emotions and their categorical ambiguity in facial expressions are parametrically encoded in the human amygdala”. (4) We strengthened the Introduction and Discussion to have a more comprehensive review of literature regarding ambiguity coding. (5) We have clarified the term “ambiguity coding” and conducted several control analyses to confirm that the results correspond to our definition of this term. We strongly believe that these additional changes make the paper suitable for publication in *Nature Communications*.

Reply to comments from Reviewer 1

I thank the authors for their detailed response to my comments and those of the other reviewers. Many of the issues I raised have been fully addressed. However, I do still have a number of outstanding concerns as detailed below.

Comments on initial general response from authors:

(1) I agree with the authors that the paper is best served by leaving in all three datasets (fMRI, lesion, single unit recording) and with their sentiment that 'Deleting a dataset from a measure that does not fit so well with the others would produce an artificial consensus that, we feel, is not in the best interests of the field.'

(2) The inclusion of the anger/disgust morph single unit recording data is valuable and goes a long way to addressing my concerns that the main findings might be specific to the happy/fear morph stimuli used. In particular, happy – fear expression blends are rarely found in real-world settings, so it is valuable to establish that effects of ambiguity are similar for a dimension where blends of the anchor expressions are more likely to be encountered in real-world settings.

Comments on authors' response to my first round review

Issue 1. Generalizability of findings given use of a single expression continuum (happy/fear). See (2) above. The authors have fully addressed my concerns here.

Issue 2i. Prior comment 'The findings from these analyses indicate greater amygdala responsivity to the 100% fear or happiness 'anchor' faces than to more ambiguous intermediate face morphs.... It is potentially confusing to frame this as having shown 'representation of ambiguity' in the amygdala.'

Authors' response: We have clarified this issue in the text by referring to 'parametric encoding' of ambiguity instead of representation.

New comments: I still think the phrasing is not ideal and may cause confusion. For example in the abstract the authors state that 'At the single-neuron level, we studied single neurons and found two populations of neurons, one whose response correlated with the gradual change of emotion intensity and a second whose response signaled levels of ambiguity. Together, our results indicate that the human amygdala processes both the intensity of emotion in facial expressions as well as the ambiguity of the emotion shown'.

I think several of the terms used here (and in the main paper) are likely to confuse the reader. The Introduction focuses on dominant prior findings that the amygdala responds positively to (i) face fearfulness and (ii) face ambiguity. Readers are likely to assume a statement that 'the amygdala processes the ambiguity of the emotion shown' replicates prior findings of heightened amygdala activity to ambiguity rather than the reverse. Hence, I think it is important that the authors phrase their key findings in a manner that is completely clear. So for example in the abstract (and elsewhere) they could state that one population of neurons 'showed activity that decreased as a linear function of ambiguity'.

An equivalent issue pertains to the description of the other population of neurons 'whose response correlated with the gradual change of emotion intensity'. Here the authors are referring to the finding that some neurons show increased firing as a linear function of expression fearfulness and others increased firing as a linear function of expression happiness (which could also be phrased as decreased firing or suppression of activity as a linear function of expression fearfulness). I would ask the authors not to use the term 'emotion intensity' as it isn't transparent as to whether this equates to expression fearfulness, expression happiness or (as some might

have predicted) closeness to dimension ends. Referring instead to ‘expression fearfulness’ or ‘percentage of fear in the expression’ would be much clearer. Similarly for the fMRI data, the authors should specify that activity decreased as a linear function of expression fearfulness and expression ambiguity.

We once again thank the reviewer for the detailed and valuable comments.

We agree with the reviewer and now state explicitly the directionality of coding in the revised manuscript (whether neuronal responses increased or decreased). We also now provide a clearer explanation of the amygdala’s role in coding ambiguity. Please refer to our response to Question 2v below. As the reviewer suggested, we have replaced “emotion intensity” with the more precise expressions “degree of fear” or “degree of happiness” throughout the manuscript. In particular, we have modified the text in the following places:

On p.2:

“At the single-neuron level, we studied 234 single neurons and found two populations of neurons, one whose response correlated with increasing degree of fear, or of happiness, of a face, and a second whose response primarily decreased as a linear function of emotion ambiguity (the uncertainty that a stimulus is categorically either fear or happy, which was the task subjects performed).”

On p.4:

“At the single-neuron level, in contrast, we found clear evidence for a segregation of these two functions: we identified two separate populations of neurons, one whose response correlated with the gradual change of fearfulness or happiness of a face and a second whose response primarily correlated with decreasing level of categorical ambiguity of the emotion.”

On p.15:

“By contrast, our fMRI study showed that the BOLD signal within the amygdala decreased with both the degree of fearfulness and the categorical emotion ambiguity, albeit on different sides of the brain. Finally, our electrophysiological study revealed one population of neurons that tracked the degree of fear or happiness in a face while another population of neurons primarily tracked decreasing categorical ambiguity (the uncertainty that a stimulus is either fear or happy).”

On p.21:

“We thus here show a reward- and reward-value independent decreasing response to ambiguity (increasing response to certainty) in the amygdala.”

Issue2ii I thank the authors for adding text to address the point raised here.

Issue 2iii. I thank the authors for adding text to address the point raised here. I have just one remaining comment, the authors state that ‘Therefore, more pronounced adaptation for stimuli in the middle than at the extremes of the face continuum could in principle be one mechanism that explains the amygdala’s response to ambiguity we found in our study. However, we used a sufficient number of distinct stimuli, and their order was completely randomized, making face adaption implausible in our protocol.’

I think the last statement is too strong given that Aguirre and colleagues have shown that even when stimulus order is randomized adaptation can occur both for (i) face identity and (ii) face expression. I do however think

it is would be reasonable to state something along the lines that ‘adaptation to facial emotion is likely to be lower when facial identity is also manipulated on a trial to trial basis as in the current study’

We thank the reviewer for pointing this out. We have modified our statement to be (on p.18):

“Therefore, more pronounced adaptation for stimuli in the middle than at the extremes of the face continuum could in principle be one mechanism that explains the amygdala’s response to ambiguity we found in our study. However, we used a sufficient number of distinct stimuli, and their order was completely randomized, making face adaptation not likely in our protocol. Moreover, adaptation to facial emotion was likely to be lower when facial identity also changed on a trial-by-trial basis as in the present study.”

It is worth noting that Kahn and Aguirre (NeuroImage, 2012) had a substantially shorter inter-stimulus-interval of only 200ms. Therefore, the adaptation between stimuli is expected to be much stronger (also see Aguirre NeuroImage 2007 for the adaptation/carry-over effects in such rapid event-related design). However, we had a much longer inter-stimulus-interval of 2-8 seconds (jittered randomly with a uniform distribution). And as Kahn and Aguirre also noted, “a sparse fMRI design with long inter-stimulus intervals (e.g., greater than 6 seconds) would both plausibly reduce adaptation effects and reduce the degree of confound.”

Issue 2iv. I thank the authors for clarifying this – my concerns here are fully addressed.

2v. The issue raised here was a very important one given the focus of the paper:

'Finally, if it is indeed the low ambiguity of the anchor faces that is critical, then an issue that also requires discussion is how this fits with other accounts suggesting that the amygdala is sensitive to high levels of ambiguity (e.g. Whalen TICS 2007). Indeed the work on ambiguity reviewed in the Introduction points to the amygdala response varying positively with level of ambiguity.'

I think it needs addressing more comprehensively than the authors have done as yet. In the Introduction the authors originally had the following text:

'On the other hand, the amygdala is also crucial for identifying ambiguous stimuli and in modulating vigilance and attention as a function thereof. For instance, the BOLD-fMRI signal in the amygdala is correlated with the level of ambiguity in decision making and focal amygdala damage undermines decision making under ambiguity. In addition, unpredictability of stimuli even without any motivational information prevents rapid habituation of single neuron activity in the basolateral amygdala in mice and causes sustained neural activity in the amygdala in humans.'

This is strong, cross-species evidence for the amygdala showing increased firing with ambiguity (i.e. the opposite to the current finding). The authors have now added a single sentence to provide (fMRI) evidence that, in some cases the amygdala might show activity that decreases with ambiguity-

'However, other studies have shown suppression of BOLD activity for faces with the most ambiguous trustworthiness'

This is really very limited relative to the huge evidence in the opposite direction given above and seems a bit disjointed. Is there nothing else? If not, then the authors really need to acknowledge that the majority of the findings in the literature support a prediction that amygdala activity will increase as a function of ambiguity.

In this context, I also do not find the discussion of the counter-intuitive ambiguity result in the discussion sufficient. Here the authors state that ‘Some studies have shown that the amygdala responds maximally to

ambiguous stimuli or ambiguous choices. In contrast, we found that both the neuronal population response and BOLD signal were maximal for the least ambiguous faces. This is consistent with other studies showing suppression of BOLD-fMRI activity for faces with the most ambiguous trustworthiness'. This would be okay if there really was an equal balance of evidence in both directions in the prior literature as this seems to imply, but there isn't (or if there is the authors need to include a lot more on findings that amygdala activity increases to low ambiguity in the introduction). This is compounded by the subsequent statement that 'Here we interpret any change in firing rate, or in BOLD signal, as carrying information, and therefore do not further interpret the sign of that change'. I found this rather frustrating / inadequate. While it might be justifiable to say it is hard to interpret the direction of BOLD activity, this is not true for single neuron recordings (which indeed is their advantage) so the authors really need to tackle the issue that their ambiguity finding is in the opposite direction to what many would predict rather than just state they don't think this is important.

We thank the reviewer for pointing this out. We have clarified and strengthened the Introduction as follows (on p.3):

“On the other hand, the amygdala is also crucial for identifying ambiguous stimuli and in modulating vigilance and attention as a function thereof (Whalen, 1998, Adams et al., 2003, Roesch et al., 2010). For instance, the BOLD-fMRI signal in the amygdala is correlated with increasing ambiguity in decision making (Hsu et al., 2005) and focal amygdala damage undermines decision-making under ambiguity (Brand et al., 2007). In addition, unpredictability of stimuli even without any motivational information causes sustained neural activity in the human amygdala (Herry et al., 2007). Although a large neuroimaging literature shows higher amygdala responses to more ambiguous stimuli, it is worth noting that studies using faces along a dimension that admits of categorical ambiguity show higher amygdala responses to less ambiguous (more categorically certain) faces. For instance, it has been consistently shown that the amygdala responds strongest to highly trustworthy and highly untrustworthy faces (Todorov et al., 2008, Said et al., 2009, Freeman et al., 2014), but less to faces of intermediate (ambiguous) trustworthiness, even if the faces are perceived unconsciously (Freeman et al., 2014). Furthermore, for both faces varying along a valence dimension and faces varying along an orthogonal non-valence dimension, the amygdala has a stronger response to the anchor faces (Said et al., 2010). Consistent with this idea, all emotional stimuli are associated with a higher amygdala activity compared to neutral stimuli, even though there are comparable effects for most negative and positive emotions (Costafreda et al., 2008). Together, these more recent findings argue that the amygdala plays a key role in processing categorical ambiguity of dimensions represented in faces (just like in our study), and suggests that the prior studies in ambiguity (Whalen, 1998, Hsu et al., 2005, Herry et al., 2007, Roesch et al., 2010) actually tap a different construct of “ambiguity”. Notably, these latter studies generally focus on the economic usage of that term, which refers to an absence of information about a stimulus above and beyond categorical uncertainty (which is called “risk”). Thus, if the probability distribution of a stimulus belonging to either category A or B can be estimated, the stimulus has risk corresponding to its estimated probability; whereas if the probability distribution itself is unknown, this would be termed “ambiguity” (Camerer and Weber, 1992). By contrast, the studies on face processing we cited (Costafreda et al., 2008, Todorov et al., 2008, Said et al., 2009, Said et al., 2010, Freeman et al., 2014) generally present stimuli for which their probability of belonging to category A or B (fear/happiness, trustworthy/untrustworthy, etc) can be estimated: in those cases, the amygdala appears to track the categorical certainty and often shows a minimum response where categorical ambiguity is highest. While we use the terminology “ambiguity” in our study to refer to the categorical ambiguity (that is, the uncertainty) of a face belonging to one of two categories (see (Sterzer and Kleinschmidt, 2007) for a classical example of perceptual ambiguity that shares the same meaning of “ambiguity” as ours), it is important to note this difference in terminology from some of the other studies, such as those on decision-making.”

We have also rewritten the Discussion (on p.17):

“Increased amygdala responses to ambiguity have been found in studies on decision-making (Whalen, 1998, Hsu et al., 2005, Herry et al., 2007, Roesch et al., 2010) that do not involve ambiguous choices between two face attributes, whereas increased amygdala responses to certainty have been found in studies where an ambiguous choice is made between two options for a face (Costafreda et al., 2008, Todorov et al., 2008, Said et al., 2009, Said et al., 2010, Freeman et al., 2014). Thus, our results are entirely consistent with a subset of studies on the amygdala’s role in coding perceptual ambiguity/certainty, specifically those studies that investigate the same construct of “ambiguity” as ours (see **Introduction**).”

Issue 3) The authors have fully satisfied my concerns here.

Issue 4) Prior comment: 'With regards to the lesion study, how do the authors reconcile the finding that amygdala lesion patients were 'more likely to judge faces as fearful' (pg 5) with prior reports that patients with amygdala damage show reduced ability to recognize fearful expressions (Adolphs et al., Nature, 1994)?'

I thank the authors for clarifying their position on this - that the prior literature would lead to the prediction that patients with basolateral lesions might show hyper-vigilance for fear. Given the importance of this to the authors’ predictions I believe that this should be included in the Introduction, where at the moment only lesion data linking amygdala damage to impaired fear recognition is discussed. (Currently when the authors present their lesion results, they introduce the BL specific lesion background for the first time and then refer the reader to the discussion, this isn’t optimal).

We thank the reviewer for pointing this out. We have explicitly mentioned this important difference in lesion extent in the Introduction, and explained it in more detail in the Discussion.

We have included the following text in the Introduction (on p.3):

“It is worth noting that the above-mentioned lesion studies encompass damage to both basolateral and centromedial nuclei. In our present three amygdala lesion patients, most of the basolateral complex of the amygdala was lesioned but the centromedial nucleus was spared. Given the lesion extent and the known inhibitory projections from the basolateral to the centromedial nuclei as gleaned from animal studies (see **Discussion** for details), we therefore expected our lesion patients to show increased sensitivity to the degree of fear in our stimuli, as has indeed been reported in other patients with such partial amygdala lesions (Terburg et al., 2012, van Honk et al., 2016).”

We updated and included the following text in the Discussion (on p.19):

“In our three amygdala lesion patients, most of the basolateral complex of the amygdala was lesioned but the centromedial nucleus was intact, an important difference to the complete amygdala lesion that has been studied in detail in patient S.M. (Buchanan et al., 2009, Adolphs et al., 2016). The basolateral amygdala is the primary source of visual input to the amygdala and the centromedial amygdala is a primary source of output to subcortical areas relevant for the expression of innate emotional responses and associated physiological responses (LeDoux, 2007). The centromedial amygdala provides most of the amygdala projections to hypothalamic and brainstem nuclei that mediate the behavioral and visceral responses to fear (Aggleton et al., 1980, Davis, 1992, Fudge and Tucker, 2009). Furthermore, the projection neurons in the central nucleus are mostly inhibitory, and are, in turn, inhibited by inhibitory intercalated cells in the lateral and basal amygdala. Disinhibition through this pathway is thought to lead to the expression of emotional responses (LeDoux, 2007). Although direct evidence into the role of amygdala subregions in threat and fear processing comes

predominantly from rodent lesion research (Phelps and LeDoux, 2005) and optogenetics (Haubensak et al., 2010), our present finding of a lowered threshold for reporting fear for morphed faces is consistent with prior human amygdala lesion results: patient S.M., who has complete amygdala damage including both basolateral and centromedial nuclei, showed an increased threshold for reporting fear (the opposite of our finding) (Adolphs et al., 1994), while another group of five patients with only basolateral amygdala damage demonstrated a hyper-vigilance for fearful faces (similar to our finding) (Terburg et al., 2012, van Honk et al., 2016). A putative mechanism explaining this difference between the two types of amygdala lesion patients comes from optogenetic work in rodents. Specific activation of the terminals that project from the basolateral amygdala to the central nucleus reduces fear and anxiety in rodents, whereas inhibition of the same projection increases anxiety (Tye et al., 2011). It is thus possible that the partial amygdala lesions in our three subjects removed a normal inhibitory brake on fear sensitivity and resulted in exaggerated sensitivity to emotion mediated by the disinhibited central nucleus, just like in the prior studies (Terburg et al., 2012, van Honk et al., 2016).”

Minor issues: all fully addressed.

Additional issues

1) request relating to Issue 4.

The authors now clarify that their lesion predictions are based on the damage being limited to the BL nucleus. For their single neuron recording, they also comment on the location of the neurons that track expression fearfulness and ambiguity (BL vs Central Nucleus). Given this, it would also be valuable if the authors could likewise add a small section to their fmri analyses to discuss if the location of the activation observed is more consistent with it being a BL or CN signal. (There are probabilistic atlases of the amygdala which give the likelihood that a given activation is in the BL or CN, or better still the authors could use each individual’s anatomical scan to roughly estimate this). It is fine to indicate that fMRI data is not of the resolution to make firm conclusions regarding this but it would still be valuable to have something along these lines (see also work by Whalen who has argued that a given ‘z’ co-ordinate can be used to crudely distinguish which part of the amygdala one is in, Kim et al., 2003; Somerville et al., 2004.)

We thank the reviewer for pointing this out. We have included the following results (on p.13):

“However, notably, our fMRI data suggested that the activation by emotion degree (Fig. 2a) was centered primarily in the basolateral nucleus (only 1 voxel in the central nucleus; using the atlas of (Tyszka and Pauli, 2016)), consistent with the altered emotion judgment in lesion patients and the distribution of emotion-tracking neurons (both of which also involved primarily the BLA), and activation by emotion ambiguity (Fig. 2c) also appeared to be in the basolateral nucleus (all voxels), consistent with the distribution of ambiguity-coding neurons (also see Fig. 2g for peak voxel activity). These commonalities should, however, be considered preliminary since they are fundamentally limited by the spatial resolution that we have to resolve individual amygdala nuclei across all our measures, especially the fMRI measure.”

We further included a plot (Fig. 2g) to show this result:

“g. Peak voxel activity within the basolateral nuclei (BLA) and central nuclei (CeA) for each aspect of the emotion coding.”

2) I agree with reviewer 2 that ideally the ratings of intensity and valence should be acquired from participants of the same demographic background (ethnicity and country of birth /long-term residence). This also raises the broader issue of whether any discrepancies in findings across methodologies may actually reflect different responses to the morphed faces in Western Caucasian versus Eastern Asian participant groups. As reviewer 2 points out there are known differences in expression processing between these groups. Hence, if possible, it would be useful for the authors to acquire an additional set of face ratings from control Western Caucasian participants. If these do not differ from those of the Eastern Asian control group it would both provide reassurance that the control ratings are pertinent to the patient groups (as well as the fMRI participants) and that differences in expression perception between the different participant groups do not account for differences in findings between methodologies.

We thank the reviewer for pointing this out. We collected new data from 13 Western Caucasians. This now shows that the patterns of ratings are similar to those of the Eastern Asians. We have updated the following results (on Supplemental p.13):

“Could the decreased neural responses to ambiguity be explained by systematic variability in valence and intensity? To test this, we acquired valence and intensity ratings on our stimuli from 23 additional subjects (Supplementary Fig. 1f-q). Ten subjects were Eastern Asians (Supplementary Fig. 1f-k) and 13 were Western Caucasians (Supplementary Fig. 1l-q). As expected, there was a relationship between decreasing valence and increasing intensity as a function of the fearfulness of the morphed faces (Supplementary Fig. 1f,i,l,o). This is consistent with the general evaluation of happy and fear in a two-dimensional structure of affect². In addition, this control experiment also demonstrates that the subtle and gradual changes of facial emotions could be resolved by subjects, a result we also found for both genders (Supplementary Fig. 1g,j,m,p) and face identities (Supplementary Fig. 1h,k,n,q). Crucially, the valence/intensity ratings did not exhibit the U-shaped pattern that we found for ambiguity-coding signals. In particular, note that stimuli with the smallest and largest values on both valence and intensity were equally ambiguous. We therefore conclude that the ambiguity signal was not

driven by valence or intensity dimensions. Notably, Asians (Supplementary Fig. 1f-k) and Caucasians (Supplementary Fig. 1l-q) demonstrated similar ratings.”

And Supplementary Fig. 1:

Supplementary Figure 1

“f,i, Valence ratings across morph levels. i,o, Intensity ratings across morph levels. We asked an independent 10 Asian subjects from South China Normal University (similar to the fMRI subjects) and an independent 13 Caucasian subjects from the United States and Germany (similar to the neurosurgical subjects and amygdala patients) to rate the faces. This group of subjects was independent from any of the subject pools that contributed other data to the paper and generated only ratings of the stimuli as shown in this figure. Each face was rated 5 times on a 1 to 10 scale. We asked ‘how pleasant is this emotion that the face shows’ for valence, with 1 for very unpleasant and 10 for very pleasant. We asked ‘how intense is this emotion that the face shows’ for intensity, with 1 for very mild/calm and 10 for very intense/excited. Valence ratings decreased with morph levels whereas intensity ratings increased with morph levels. Error bars denote \pm SEM across subjects. g,j,m,p, Valence (g,m) and intensity (j,p) ratings shown separately by subject gender. h,k,n,q, Valence (h,n) and intensity (k,q) ratings shown separately for each identity shown in a. Shaded area denotes \pm SEM across subjects.”

It is also worth noting that neurosurgical (Caucasian) and control subjects (Asian) had remarkably similar emotion judgments (Fig. 1c), further arguing against any cultural differences on our task.

Reply to comments from Reviewer 2

I think the authors took great care of my comments and did an excellent job in addressing all of them. I can thus recommend this paper for publication.

We once again thank the reviewer for the encouraging and constructive comments.

Reply to comments from Reviewer 3

The authors have addressed all my questions.

We once again thank the reviewer for the encouraging and constructive comments.

Reply to comments from Reviewer 4

this is a resubmission of a revised manuscript.

The authors did considerable work addressing the concerns.

However, there are still fundamental concerns, and in general, I think this study is more suitable for a specialized journal. The paradigm is not sophisticated enough, rich, or well controlled, when compared to other studies (behavioral and fmri) these days. Recording human single-units, even if valuable and rare data, is not enough to compensate for this. In addition, the findings in single-units is not analyzed to the level that one can expect in physiology studies.

I am really sorry to say so, because I can see there was considerable work done here and they have valuable data, but currently the main claim is only indirectly supported.

We thank the reviewer for the constructive comments. We believe our manuscript has a broad readership because it makes four key novel contributions. First, it tests two key hypothesized functions of the amygdala (fear and ambiguity) using a combination of fMRI, single-neuronal recordings and behavioral analysis of patients with focal bilateral amygdala lesions. Together, this addresses the important open question of what the amygdala encodes. Second, our results link and reconcile the findings from a large number of studies performed with a single modality alone. Third, our approach will be of interest for its uniquely multimodal mapping with lesion, electrophysiology, and functional neuroimaging. Lastly, our results provide valuable data that informs our understanding of how amygdala dysfunction might lead to mood and anxiety disorders.

Due to the limitations in recording from neurosurgical patients, our task had to be relatively simple. However, our task was well suited to answer our research questions and make our principal claims. We have answered all questions from this reviewer below.

Major

- Ambiguity is not well defined here and can be attributed to other factors. The main one here is that ambiguity is the inverse of intensity. I understand the authors' claim, but this task does not decorrelate the 2 factors. Here is why: the anchors have the higher intensity, and therefore any correlate that is high/low at the morphs and low/high at the anchors can be interpreted as intensity coding rather than ambiguity coding. Since a big majority of their finding in single neurons suggest that there is higher response to anchors, this makes it even more suspicious.

As we noted above, for clarity, we have changed to use “the degree of fear or happiness” rather than “emotion intensity”. We apologize if the reviewer was confounded by “emotion intensity” and our “valence/intensity ratings”.

Emotion intensity in our previous version of manuscript means the intensity (degree) of fear and intensity (degree) of happiness. As Fig. 1b clearly shows, emotion intensity is completely a different factor as emotion ambiguity. Ambiguity is the inverse of category certainty, but not of the intensity of a specific emotion. Ambiguity is low (certainty is high) for both anchors, whereas intensity is high for one anchor but low for the other anchor. To further clarify, we have specified the direction of emotion coding (i.e., increase/decrease with fear/happiness) and spelled out which degree (fear degree or happy degree) we mean in the revised manuscript. Please refer to our response to Reviewer 1's first question. Again, we refer the reviewer to Figure 1 where we explicitly plot ambiguity, and the degree of each of the two emotions. We found neurons whose responses

tracked each of these, and they are clearly not the inverse of one another. Ambiguity shows an inverted U-shape as a function of the morphs, whereas each of the degrees of emotion is linear.

Notably, our explicit valence/intensity ratings are also not correlated with ambiguity. Neuronally, we have shown that neurons encoding emotion degree and ambiguity are distinct classes and they have different response profiles (Fig. 4i-k). Thus, these could not be neuronal responses to constructs that are simply the inverse of one another.

- If there is gradual coding of emotion intensity within the amygdala, a very reasonable hypothesis based on the wide literature (call it attention if you like), then it will be expressed as inverse coding to what they call ambiguity.

Again, please see our response above. We demonstrate in our paper that these neuronal populations are distinct, and show in Figure 1 that the two psychological constructs are not simply the inverse of one another.

Indeed, “most of which (30/36) had a higher firing rate for the anchor compared to the morphed faces”; again, it seems most neurons by far respond to the emotionally identified face (anchor), which raise the concern that the claim for ambiguity coding is a bit exaggerated.

And later “most (29/32) of which had the maximal firing rate for the anchors (which have low ambiguity). We refer to this group of cells as ambiguity-coding neurons” ...

Yes, I am aware that information and coding can be in inverse correlations as well, and decreasing FR can be as informative as increasing FR. Yet it still casts doubt whether there is real coding for ambiguity or simply to intensity of emotion (independent of positive/negative).

We thank the reviewer for pointing this out. We have been careful now to note that responses to ambiguity were generally decreasing, as also noted by Reviewer 1. As the reviewer correctly pointed out, ambiguity and certainty are two sides of the same coin, and both increasing FR and decreasing FR can be informative. In the revised manuscript, we have explicitly mentioned what these ambiguity-coding neurons may encode (on p.18):

“Notably, emotion ambiguity and certainty are closely related and inversely correlated, and these neurons might encode emotion certainty, or confidence in emotion judgment. Here we interpret any change in firing rate, or in BOLD signal, as carrying information, and therefore do not further interpret the sign of that change.”

And we acknowledge the role of attention (on p.18):

“Although all our stimuli should be equally attended given the task demands of having to make judgments about them, we acknowledge that task difficulty, attention, mental efforts, and RT are of course all intercorrelated to some extent, and future studies will be needed to further distinguish the possible contribution of attentional effects in our study.”

- Single units are very poorly sorted (likely multi-units). I know this is the unfortunate standard with human recordings, but in the current case it is a real confound for interpreting ambiguity-coding (because it can be a summation of 2 or more neurons each with a tuning curve increasing or decreasing in opposing manner as a function of fear).

We respectfully disagree with this comment, which appears to us completely ungrounded. We explicitly showed the waveforms of each individual example neuron, and we had a spike sorting quality figure (Supplementary Fig. 5) showing many quality controls. The various metrics shown in Supplementary Fig. 5 show, using well

accepted spike sorting quality metrics, that our recordings are properly sorted. In particular, the quality of our data is well above the standard as shown in Harris Nat Neurosci 2016. For example, even compared to animal research (see Figure 2 of this paper), the isolation distance of our neurons is comparable and probably larger (note our log10 scale in Supplementary Fig. 5g).

With regards to “standard with human recordings”, we respectfully point out that this work is performed with neurosurgical human subjects. These represent the only ability to gain access to this highly valuable data from humans. While one certainly wishes to perform recordings of the quality achievable in animals, including tetrode recordings, this at present is simply not possible due to ethical constraints.

To further address this claim, we added a new analysis that showed that there was no correlation between isolation distance and response strength, as quantified by ω^2 . We found this for both aspects that we analyzed: isolation distance was not correlated with ω^2 for either emotion degree coding ($r=-0.094$, $P=0.21$) or emotion ambiguity coding ($r=-0.047$, $P=0.53$). We further included the following two panels in Supplementary Figure 5. Together, this shows that spike sorting quality (x axis) did not have an influence on the strength of our effect (y axis).

Figure Legend. **h**, Absence of correlation between isolation distance and response strength (as quantified by ω^2) for emotion degree coding ($r=-0.094$, $P=0.21$). **i**, Absence of correlation between isolation distance and response strength (as quantified by ω^2) for emotion ambiguity coding ($r=-0.047$, $P=0.53$). Each circle represents a neuron (all neurons are shown, regardless whether selected or not). The black line represents the linear fit.

- The definition of ambiguity as consistency is problematic. It is identical to the mean in this case (the choice is binary/AFC), am I missing something here?

We apologize if our prior explanation was unclear. Ambiguity is not reflected in consistency of choices, but in the opposite, their variability. If a face is maximally ambiguous as to whether it is fear or happy, then the responses to that stimulus will categorize it as “fear” 50% of the time and as “happy” the other 50% of the time.

Less major

- In fig.3b, there is no real linear relation, only 0% is higher. Is this really the “best typical example”?

This example clearly shows a decreasing linear relationship, which has been reflected by the selection ($r=-0.80$, $P=0.032$).

- The exemplar rasters are not convincing in fig.4, and the matching PSTH only very weakly show a trend towards the finding. Again, is this the best example?

The rasters shown in Fig. 4 clearly had denser spikes around 1000ms for anchor (yellow at bottom) than intermediate (green in the middle) than the most ambiguous (black at top). Fig. S6 shows more examples.

- The imaging analysis is not so clear as well. I trust it is correctly done, but I could not understand how they used the localizer to later test for left or right amygdala in looking for which codes what. Also, In fig2b, the result comes from the happy face alone, so It seems; and in fig.2d., there is no real tuning/gradual-coding, just a higher response to no-emotion (ambiguity is the interpretation). overall, as a result, the interaction in fig.2f is a very weak effect (even if significant).

The purpose of the localizer task was to find those amygdala regions that are activated by faces (regions that responded differentially to faces compared to geometric shapes), a standard approach in fMRI studies. Instead of using the entire anatomical amygdala, the ROIs defined by the localizer task would provide an a priori region (chosen on independent criteria) that shows sensitivity to faces in the first place. We then further analyzed how these face-responsive regions responded to our task aspects (i.e., emotion degree and emotion ambiguity). Each side of the amygdala was analyzed independently. Both sides of the amygdala showed activation in the localizer task (Supplementary Fig. 4a). We did not use the localizer task to test for functional laterality, but we averaged the BOLD signal from all voxels identified by the localizer task within each side of the amygdala to compare the coding of emotion degree and ambiguity (Fig. 2f). In Fig. 2b, the correlation was from all morph levels but not a particular face/morph level. BOLD signals are very noisy; but our GLM did reveal the coding of ambiguity.

- A main motivation is very argumentative: “and leaves open the possibility that perhaps prior findings in regard to fear are derivative to this function, since fear can signal ambiguity (by indicating a threat without specifying where or what that threat is”.

Fear signals ambiguity? Why not the other way around (ambiguity drives fear; I think this is intuitive, and indeed much of the data can be interpreted this way). Not to mention that in many studies it is very clear to the animal what and where the threat is.

I simply do not see these options as ‘two theories’ of amygdala function.

We thank the reviewer for pointing this out. Indeed, it should be “ambiguity can signal fear”. We apologize for our confusing prose in the original version.

We now provide a clearer explanation of the terminology “ambiguity” used in our present study (compared to those studies on decision-making). Please refer to our response to Reviewer 1’s Question 2v.

- They find that behaviorally, amygdala patients have altered threshold for reporting fear. This is another example of a confound, showing that the ambiguity regime can actually covariate with emotional (and there is no independent measure of intensity to rely on).

The change in threshold for reporting fear does not affect the analysis of ambiguity—even if patients were more likely to report fear, they still had the least variance/highest consistency when judging anchors but the highest

variance/least consistency when judging the most ambiguous faces. In other words, the systematic response to ambiguity (both choices and reaction times) was not altered by change in threshold for reporting fear.

Again, we would like to reiterate that our conclusions do not require that emotion degree and emotion ambiguity are completely orthogonal, only that they are not perfectly correlated. Indeed, emotion degree and emotion ambiguity co-vary for each stimulus, but the relationship of these two variables to the morph levels was very different (see Fig. 1b).

- In general, they find that anchor faces induce more activity in the amygdala. The evidence for similar levels of activity across ambiguous faces from both sides (happy and fear), is obviously indirect (statistically).

We grouped morph levels into ambiguity levels to increase the statistical power. There was a difference in normalized firing rate between two intermediate morph levels ($P=0.0041$), but there was no difference between 40% fear/60% happy and 60% fear/40% happy ($P=0.17$). We were unable to understand the reviewer's second sentence.

- "neurons from each individual recording session were considered independent even if they were from the same patient because it was not possible to reliably record the same neurons across sessions"

there is a huge difference between not claiming these are the same neurons (e.g. that these are not chronological recordings of neurons to be compared across days), and treating them as independent neurons to be added to the overall pool. I hope the authors can see this. This can increase the statistical power without taking into account dependency. Please repeat analyses on a subset as well to see if the results are replicated.

Although our approach is entirely standard in the field, we do appreciate the reviewer's point. We repeated our analysis by only considering the neurons from the first recording session of each patient (note that this would also remove any familiarity effect). In the remaining 151 neurons with an overall mean firing rate $>0.2\text{Hz}$, we found 19 emotion-tracking neurons (12.6%, binomial $P=7.09\times 10^{-5}$; 11 neurons increased firing rate with fear levels and 8 neurons decreased firing rate with fear levels) and 22 ambiguity-coding neurons (14.6%, binomial $P=2.01\times 10^{-6}$; 2- neurons increased firing rate with ambiguity levels and 2 neurons decreased firing rate with ambiguity levels). Notably, the percentage of emotion-tracking neurons (χ^2 -test: $P=0.67$), ambiguity-coding neurons ($P=0.81$), and each subclass of neurons (both $P_s>0.68$) was not significantly different from the entire population.

We further confirmed this result by the population regression analysis, where we found that our results could be well replicated with neurons only from the first recording session of each patient (see figure below). Therefore, our results could not be explained by a confound result from "double-counting" neurons across multiple recording sessions.

Figure Legend. Population regression analysis for neurons only from the first recording session. Summary of the effect size across all runs: amygdala neurons as a population encoded both emotion degree (**a**, $P < 0.001$) and ambiguity (**b**, $P < 0.001$). Effect size was computed in a 1.5-second window starting 250 ms after stimulus onset (single fixed window, not a moving window) and was averaged across all neurons for each run. Gray and red vertical lines indicate the chance mean effect size and the observed effect size, respectively. The observed effect size was well above 0 whereas the mean effect size in the permutation test was near 0.

- Overall, there are neurons of each possible type, and I am unconvinced that there is real ambiguity and not some homogenous distribution of responses that vary with the stimulus. The claim for proportions that are beyond expected from chance is not well established in my view. Some better controls for estimating the proportions of different populations are required. E.g. reshuffle the stimuli and re-analyze, to see if similar proportions of neurons would correlate with such imaginary property.

We performed a permutation test in Supplementary Fig. 7 for emotion-tracking neurons. We further performed a permutation test for ambiguity neurons. We have included the following results in Supplemental Results (on p. 12) and the following panels in Supplementary Fig. 7:

“We carried out a permutation test to confirm the ambiguity coding (Supplementary Fig. 7h-j). When we randomly assigned ambiguity levels for each trial, we observed chance selection of 11.72 ± 3.34 (mean \pm SD) neurons (1000 permutations; two-tailed t-test against the number of chance selection (11.7 neurons): $P = 0.86$), among which 6.13 ± 2.51 neurons showed positive correlations (higher response for anchor faces) and 5.58 ± 2.28 neurons showed negative correlations (higher response for more ambiguous faces), in contrast to our observed data with predominantly positive correlations. Together, we confirmed that the ambiguity-coding neurons that we observed were not due to chance.”

Figure Legend. h-j, Quantification of the permutation test. **h**, Cells selected across runs. Each black dot means a particular cell was selected. Cell selection was evenly distributed across cells and runs in the permutation test, showing no consistency or selection bias. **i**, Summary of the number of cells selected. The number of cells selected in the permutation test was near chance (gray bar in the left histogram and black dashed line in the right bar plot). Error bar denotes \pm SEM across permutation runs. **j**, Summary of the likelihood of each cell being selected. Each cell was equally likely to be selected with the predetermined false discovery rate of 0.05.

Reviewers' Comments:

Reviewer #1 (Remarks to the Author):

The authors have continued to be extremely responsive to the concerns expressed by myself and the other remaining reviewer. As requested I've looked over their responses to reviewer 4 as well. I have one remaining suggestion for an analysis and also some fairly general suggestions regarding the Introduction and Discussion. I think the paper had progressed significantly and will make a valuable contribution to the field.

Methods and Results

At this point, my opinion is that the authors have addressed most of specific queries re methods or results that have been raised. I return to one outstanding issue here:

(1) Controlling for Intensity. The authors were extremely responsive in collecting data from Western participants (to match the intracranial participants) and showed in Fig S1 that the patterns of intensity and valence ratings for the facial stimuli were broadly similar across the Western and Eastern control groups. However, there is definitely more of a suggestion of an inverted U of intensity in the Western participants' ratings (e.g. Fig S1p). Given the concern expressed by reviewer 4 that the amygdala's deactivation to ambiguity might actually reflect activation to emotional intensity (regardless of the specific emotion), it would be great if the authors could conduct one final analysis where they add the mean intensity ratings of the Western control participants as a covariate into their regression examining single neuron responses to face ambiguity. If emotional intensity does explain away (in full or in part) the ambiguity effects this is not necessarily uninteresting but it is important that the readers should be clear on whether or not this is the case.

Introduction and Discussion

(2) The authors have also been very responsive in their rewriting of these sections. Here, I think that an ongoing issue for reviewer 4 and, to some extent, for myself has been the framing of the current study's findings as testing the alternate hypotheses that the amygdala encodes fearfulness of expressions versus their ambiguity. The single neuron findings suggest that the actuality is that some neurons encode degree of fear expressed/perceived, others encode degree of happiness expressed/perceived and others respond most strongly to the anchor expressions and least strongly to those faces closest to the perceptual boundary (i.e. most ambiguous in the categorical label that should be applied). If it can be shown that this latter effect is not simply a reflection of expression intensity (see (1) above) then this effect might well reflect a certainty signal / distance from perceptual boundary.

These issues need to be clearly framed in the Introduction (with reference to the amygdala having been shown to respond to various different expressions, though perhaps particularly fear, and also with a clear explanation of what is meant by ambiguity) and addressed in the Discussion. I worry that the incremental nature with which material has been included in response to the review process may hinder the reader in easily coming away with the key take home messages. I think that some restructuring and condensing of both the Introduction and Discussion might help with this. If I were the authors, I might be tempted to flag up the alternate uses of the term 'ambiguity' up front (which they valuably included in the last revision) and to clarify that they are referring to closeness to the perceptual boundary when a categorical decision is being made about face expression. Prior findings of increased amygdala activity as a function of confidence in the social / emotional meaning (e.g. trustworthiness) of a face - i.e certainty regarding information /stimulus content that we know to be represented by the amygdala -can then be presented while also discussing other findings linking heightened amygdala activity to expressions that are inherently ambiguous in meaning (e.g. Paul Whalen's work on surprise). The authors correctly point out that in the decision-making literature, the term ambiguity is typically used to refer to missing information, and that there is no missing information here. This might be worth mentioning but with less of a focus than it is currently given. (The truly relevant decision making literature would be that on perceptual decision-making but I'm not sure if that literature has reported much regarding amygdala activation to perceptual uncertainty or not, as I am less familiar with it.) In summary, I would probably suggest that the authors make it clear what is meant by ambiguity in the current study and then predominantly reference the literature that pertains to the construct that they are addressing.

Smaller points:

Intro, lines 55-58 I would cut these as they relate to hypotheses that have not been introduced as yet. Instead, just include the fact that findings in patients with partial lesions have diverged from those with complete amygdala lesions.

Intro 66-67. 'dimension that admits of' – this doesn't make sense to me, may need rephrasing.

Pg 17. 458-460. "whereas the amygdala responds abruptly when sufficient information for familiar face recognition is accumulated. In our present study, we found similarly parametric effects, not only with neuroimaging but also in direct electrophysiological responses."

This confuses me – it seems you are saying the amygdala unlike other regions has been reported to show a step function but then go on to say here you found 'similarly parametric' effects.

Discussion. It seems like there is some overlap between sections here (e.g. there are various sections dealing with ambiguity in one way or another). I think this adds to the sense of this being very long and the main 'thrust' feeling a little lost. I'd encourage the authors to condense this down a bit, perhaps moving the 'ambiguity and confidence' section to after the 'ambiguity

coding' section and maybe cutting the 'ambiguity versus reward-related' signals section.

Reviewer #4 (Remarks to the Author):

The authors have tried to improve the manuscript further, and i respect that, just as i respect the authors previous excellent work in general.

Yet in the current case, i cannot support publication and i think it belongs in a more specialized journal.

i will not repeat most of the concerns i had, but just emphasize again that the task does not decorrelate intensity and ambiguity. no matter how they spin it, fear-intensity and happiness-intensity can be different, but for interpretation of neural activity - they do not know if a neuron codes for the absolute value of intensity, or for ambiguity - both will have a U-function. this becomes a major confound in the case of the amygdala, that likely codes for attention.

I simply do not think that the rarity and importance (and i do not doubt this) of recording in humans can be used to cover for task and analyses confounds.

i do not wish to be the 'bad guy', and if the other 3 reviewers strongly support publication, then it is an editorial decision.

Reply to comments from Reviewer 1

The authors have continued to be extremely responsive to the concerns expressed by myself and the other remaining reviewer. As requested I've looked over their responses to reviewer 4 as well. I have one remaining suggestion for an analysis and also some fairly general suggestions regarding the Introduction and Discussion. I think the paper had progressed significantly and will make a valuable contribution to the field.

Methods and Results

At this point, my opinion is that the authors have addressed most of specific queries re methods or results that have been raised. I return to one outstanding issue here:

(1) Controlling for Intensity. The authors were extremely responsive in collecting data from Western participants (to match the intracranial participants) and showed in Fig S1 that the patterns of intensity and valence ratings for the facial stimuli were broadly similar across the Western and Eastern control groups. However, there is definitely more of a suggestion of an inverted U of intensity in the Western participants' ratings (e.g. Fig S1p). Given the concern expressed by reviewer 4 that the amygdala's deactivation to ambiguity might actually reflect activation to emotional intensity (regardless of the specific emotion), it would be great if the authors could conduct one final analysis where they add the mean intensity ratings of the Western control participants as a covariate into their regression examining single neuron responses to face ambiguity. If emotional intensity does explain away (in full or in part) the ambiguity effects this is not necessarily uninteresting but it is important that the readers should be clear on whether or not this is the case.

We performed the additional control analysis suggested by the reviewer and added the results to the manuscript. We used the mean ratings of Western Caucasians for each face as a covariate of the regression model used to select neurons. While somewhat fewer neurons were selected as expected (n=36 vs. n=28), we found that the pattern of response of the group of selected ambiguity-coding neurons remained essentially identical to that without this regressor (see reviewer-only Figure below, which shows neurons selected using the new model). Thus, the effect of ambiguity was not 'explained away' by intensity in the majority of ambiguity-coding neurons. In addition, we selected using the new intensity regressor alone and found that the number of selected neurons was no larger than chance. We thus conclude that intensity, as judged by the control subjects, was not a significant confounding factor for our ambiguity neuron result.

Reviewer-only Figure Legend. a,b, Average normalized firing rate of ambiguity-coding neurons that increased (n=23) and decreased (n=5) their firing rate for the least ambiguous faces, respectively. Asterisk indicates a significant difference between the conditions in that bin ($P < 0.05$, one-way ANOVA, Bonferroni-corrected). **c-d,** Mean normalized firing rate. Normalized firing rate for each unit (left) and mean \pm SEM across units (right) are shown at each ambiguity level. Asterisks indicate significant difference between conditions using paired two-tailed t-test. ***: $P < 0.001$.

We have included the following results in Supplementary Results (on p.9):

“In addition, when adding the mean intensity rating from Western Caucasians for each face as a covariate into our regression model, we could still select 28 ambiguity-coding neurons (12.0%; binomial $P = 7.36 \times 10^{-6}$; 5 neurons increased firing rate as a function of ambiguity and 23 neurons decreased firing rate as a function of ambiguity), and these neurons had a similar pattern of response to that shown in Fig. 4. Furthermore, using the same selection as the ambiguity-coding neurons, we found that only 10 neurons had a significant trial-by-trial correlation with intensity ratings (4.27%; binomial $P = 0.63$), and only 1 of these 10 neurons was a ambiguity-

coding neuron. Again, similar results were derived when excluding the neuron that also encoded emotion intensity. Together, our results suggest that the response of ambiguity-coding neurons could not be explained by emotion intensity.”

Introduction and Discussion

(2) The authors have also been very responsive in their rewriting of these sections. Here, I think that an ongoing issue for reviewer 4 and, to some extent, for myself has been the framing of the current study’s findings as testing the alternate hypotheses that the amygdala encodes fearfulness of expressions versus their ambiguity. The single neuron findings suggest that the actuality is that some neurons encode degree of fear expressed/perceived, others encode degree of happiness expressed/perceived and others respond most strongly to the anchor expressions and least strongly to those faces closest to the perceptual boundary (i.e. most ambiguous in the categorical label that should be applied). If it can be shown that this latter effect is not simply a reflection of expression intensity (see (1) above) then this effect might well reflect a certainty signal / distance from perceptual boundary.

These issues need to be clearly framed in the Introduction (with reference to the amygdala having been shown to respond to various different expressions, though perhaps particularly fear, and also with a clear explanation of what is meant by ambiguity) and addressed in the Discussion. I worry that the incremental nature with which material has been included in response to the review process may hinder the reader in easily coming away with the key take home messages. I think that some restructuring and condensing of both the Introduction and Discussion might help with this. If I were the authors, I might be tempted to flag up the alternate uses of the term ‘ambiguity’ up front (which they valuably included in the last revision) and to clarify that they are referring to closeness to the perceptual boundary when a categorical decision is being made about face expression. Prior findings of increased amygdala activity as a function of confidence in the social / emotional meaning (e.g. trustworthiness) of a face - i.e certainty regarding information /stimulus content that we know to be represented by the amygdala -can then be presented while also discussing other findings linking heightened amygdala activity to expressions that are inherently ambiguous in meaning (e.g. Paul Whalen’s work on surprise). The authors correctly point out that in the decision-making literature, the term ambiguity is typically used to refer to missing information, and that there is no missing information here. This might be worth mentioning but with less of a focus than it is currently given. (The truly relevant decision making literature would be that on perceptual decision-making but I’m not sure if that literature has reported much regarding amygdala activation to perceptual uncertainty or not, as I am less familiar with it.) In summary, I would probably suggest that the authors make it clear what is meant by ambiguity in the current study and then predominantly reference the literature that pertains to the construct that they are addressing.

We thank the reviewer for the suggestion. As suggested, we modified our introduction to directly and early on point out the kind of ambiguity (closeness to categorical boundary) we are investigating. We have rewritten the part regarding the ambiguity coding in Introduction (on p.3):

“On the other hand, in addition to encoding facial emotions, an alternative hypothesized function of the amygdala is to identify ambiguous stimuli and modulate vigilance and attention as a function thereof (Whalen, 1998, Adams et al., 2003, Roesch et al., 2010). Here, we tested the hypothesis that the amygdala encodes aspects of perceptual ambiguity when making judgments about facial emotions. Throughout this study, by ambiguity we refer to the closeness to the perceptual boundary during categorical decision between two emotional facial expressions. Note that in studies of decision-making, the term “ambiguity” usually refers to an absence of information about a stimulus above and beyond categorical uncertainty. In contrast, the term ambiguity here refers exclusively to categorical uncertainty, because all information about the stimulus was

always available and the task was deterministic (see **Discussion** for details). Previous neuroimaging work indicates that the amygdala can differentiate stimuli that vary in their perceptual ambiguity. For instance, the amygdala responds strongest to highly trustworthy and untrustworthy faces, but less to faces of intermediate (ambiguous) trustworthiness (Todorov et al., 2008, Said et al., 2009, Freeman et al., 2014), even if the faces are perceived unconsciously (Freeman et al., 2014). Furthermore, for both faces varying along a valence dimension and faces varying along an orthogonal non-valence dimension, the amygdala responds strongest to the anchor faces (Said et al., 2010). Consistent with this idea, it has been found that emotional stimuli of any type lead to greater amygdala activity compared to neutral stimuli, with comparable effect sizes for most negative and positive emotions (Costafreda et al., 2008). Together, these findings argue that the amygdala plays a key role in processing categorical ambiguity of dimensions represented in faces.”

In addition, we included a more detailed explanation of the different types of ambiguity in the Discussion (on p.16):

“The existing literature uses the term ambiguity for two entirely different constructs, and it is important to distinguish the two to properly frame our results. The first definition, which we used throughout, refers to the closeness to categorical boundaries (see Sterzer and Kleinschmidt, 2007 for a classical example of perceptual ambiguity that uses the same meaning of “ambiguity” as we did here). The second definition, which we did not refer to here, is related to missing information about stimuli in economic decision-making. In studies of face processing such as ours, the probability of stimuli belonging to one or the other category (i.e., fear/happiness, trustworthy/untrustworthy) is known. Indeed, increased amygdala responses to the second type of ambiguity have been found in studies on decision-making that do not involve ambiguous choices between two facial attributes (Whalen, 1998, Hsu et al., 2005, Herry et al., 2007, Roesch et al., 2010). In contrast to this finding, other studies find increased amygdala responses to certainty in tasks where an ambiguous choice is made between two options for a face (Costafreda et al., 2008, Todorov et al., 2008, Said et al., 2009, Said et al., 2010, Freeman et al., 2014). Thus, fMRI studies on categorical ambiguity are consistent with our present result by showing that the amygdala tracks the categorical certainty and often shows a minimal response when categorical ambiguity is highest. Therefore, our results fit with a subset of studies on the amygdala’s role in coding perceptual ambiguity/certainty, specifically those studies that investigate the same construct of “ambiguity” as ours (Costafreda et al., 2008, Todorov et al., 2008, Said et al., 2009, Said et al., 2010, Freeman et al., 2014).”

Smaller points:

Intro, lines 55-58 I would cut these as they relate to hypotheses that have not been introduced as yet. Instead, just include the fact that findings in patients with partial lesions have diverged from those with complete amygdala lesions.

We thank the reviewer for the suggestion. We have rephrased as follows (on p.3):

“It is worth noting that the above-mentioned lesion studies encompass damage to both basolateral and centromedial nuclei. In contrast, patients with lesions involving only the basolateral, but sparing the centromedial nuclei, have revealed diverging results from those with such complete lesions, typically showing a hypersensitivity to fear (Terburg et al., 2012, van Honk et al., 2016). In the three amygdala lesion patients we study here, most of the basolateral complex of the amygdala was lesioned but the centromedial nucleus was spared, and we thus hypothesized that these patients would show an increased sensitivity to fear in faces.”

Intro 66-67. ‘dimension that admits of’ – this doesn’t make sense to me, may need rephrasing.

We have rephrased this sentence as shown above.

Pg 17. 458-460. “whereas the amygdala responds abruptly when sufficient information for familiar face recognition is accumulated. In our present study, we found similarly parametric effects, not only with neuroimaging but also in direct electrophysiological responses.”

This confuses me – it seems you are saying the amygdala unlike other regions has been reported to show a step function but then go on to say here you found ‘similarly parametric’ effects.

We thank the reviewer for pointing this out. We have removed the reference to this paper because it is not directly related to our findings.

Discussion. It seems like there is some overlap between sections here (e.g. there are various sections dealing with ambiguity in one way or another). I think this adds to the sense of this being very long and the main ‘thrust’ feeling a little lost. I’d encourage the authors to condense this down a bit, perhaps moving the ‘ambiguity and confidence’ section to after the ‘ambiguity coding’ section and maybe cutting the ‘ambiguity versus reward-related’ signals section.

We thank the reviewer for the suggestion. We have streamlined the Discussion so that it reads more concise. We have further rearranged these sections.

Reply to comments from Reviewer 4

The authors have tried to improve the manuscript further, and i respect that, just as i respect the authors previous excellent work in general.

Yet in the current case, i cannot support publication and i think it belongs in a more specialized journal.

i will not repeat most of the concerns i had, but just emphasize again that the task does not decorrelate intensity and ambiguity. no matter how they spin it, fear-intensity and happiness-intensity can be different, but for interpretation of neural activity - they do not know if a neuron codes for the absolute value of intensity, or for ambiguity - both will have a U-function. this becomes a major confound in the case of the amygdala, that likely codes for attention.

I simply do not think that the rarity and importance (and i do not doubt this) of recording in humans can be used to cover for task and analyses confounds.

i do not wish to be the 'bad guy', and if the other 3 reviewers strongly support publication, then it is an editorial decision.

We once again thank the reviewer for the valuable comments. While we maintain that our task is suitable to make our claims, we at the same time appreciate the comments. Indeed, the reviewer's opinion on how the ambiguity signal is generated represents a valid hypothesis. In response, we modified the discussion to cover this prediction that follows from our work (on p.17):

“Furthermore, future studies will be necessary to investigate whether ambiguity-coding neurons are from a circuit separate from the emotion-tracking neurons. Alternatively, ambiguity-coding neurons might pool the activity of emotion-tracking neurons to generate the ambiguity signal, i.e., ambiguity-coding neurons effectively code for the absolute degree of emotions. The second hypothesis predicts that ambiguity-coding neurons would respond later in time relative to emotion-tracking neurons, a hypothesis that remains to be tested.”

Also, as Reviewer 1 requested, we added the mean intensity ratings of the Western control participants as a covariate into our regression to further examine the sensitivity of ambiguity-neurons to overall intensity. We found only minor differences, discounting this possibility. Please refer to our response to Reviewer 1's Question (1).